# ROBUST GRAPH NEURAL NETWORKS VIA UNBIASED AGGREGATION

## ABSTRACT

The adversarial robustness of Graph Neural Networks (GNNs) has been questioned due to the false sense of security uncovered by strong adaptive attacks despite the existence of numerous defenses. In this work, we delve into the robustness analysis of representative robust GNNs and provide a unified robust estimation point of view to understand their robustness and limitations. Our novel analysis of estimation bias motivates the design of a robust and unbiased graph signal estimator. We then develop an efficient Quasi-Newton iterative reweighted least squares algorithm to solve the estimation problem, which unfolds as robust unbiased aggregation layers in GNNs with a theoretical convergence guarantee. Our comprehensive experiments confirm the strong robustness of our proposed model, and the ablation study provides a deep understanding of its advantages.

## 1 INTRODUCTION

Graph neural networks (GNNs) have gained tremendous popularity in recent years due to their ability to capture topological relationships in graph-structured data (Zhou et al., 2020; Oloulade et al., 2021). However, most GNNs are vulnerable to adversarial attacks, which can lead to a substantial decline in predictive performance (Zhang & Zitnik, 2020; Entezari et al., 2020; Jin et al., 2020; Wu et al., 2019; Geisler et al., 2021). Despite the numerous defense strategies proposed to robustify GNNs, a recent study has revealed that most of these defenses are not as robust as initially claimed (Mujkanovic et al., 2022). Specifically, under adaptive attacks, they easily underperform the multi-layer perceptrons (MLPs) which do not utilize the graph topology information at all (Mujkanovic et al., 2022). Therefore, it is imperative to thoroughly investigate the limitations of existing defenses and develop innovative robust GNNs to securely harness the topology information in the data.

Existing defenses attempt to bolster the resilience of GNNs using diverse approaches. For instance, Jaccard-GCN (Wu et al., 2019) and SVD-GCN (Entezari et al., 2020) aim to denoise the graph by removing potential adversarial edges during the pre-processing procedure, while ProGNN (Jin et al., 2020) learns the clean graph structure during the training process. GRAND (Feng et al., 2020) and robust training (Deng et al., 2019; Chen et al., 2020) also improve the training procedure through data augmentation. Additionally, GNNGuard (Zhang & Zitnik, 2020) and RGCN (Zhu et al., 2019) reinforce their GNN architectures by heuristically reweighting edges in the graph. Although these defenses exhibit decent robustness against transfer attacks, i.e., the attack is generated through surrogate models, they encounter catastrophic performance drops when confronted with adaptive adversarial attacks that directly attack the victim model (Mujkanovic et al., 2022).

Concerned by the false sense of security, we provide a comprehensive study on existing defenses under adaptive attacks. Our preliminary study in Section 2 indicates that SoftMedian (Geisler et al., 2021), TWIRLS (Yang et al., 2021), and ElasticGNN (Liu et al., 2021) exhibit closely aligned performance and notably outperform other defenses under small attack budgets, despite their apparent architectural differences. However, under larger attack budgets, these effective defenses still experience a severe performance decrease and underperform the graph-agnostic MLPs. These observations are intriguing, but the underlying reasons are still unclear.

To unravel the aligned robustness and performance degradation of SoftMedian, TWIRLS, and ElasticGNN, we delve into their theoretical understanding and unveil their inherent connections and limitations in the underlying principles. Specifically, their improved robustness can be understood from a unified view of $\ell_1$-based robust graph smoothing. Moreover, we unearth the problematic

estimation bias of $\ell_1$-based graph smoothing that allows the adversarial impact to accumulate as the attack budget escalates, which provides a plausible explanation of their catastrophic failures. Motivated by these understandings, we propose a robust and unbiased graph signal estimator to reduce the estimation bias in GNNs. We design an efficient Quasi-Newton IRLS algorithm that unrolls as robust unbiased aggregation layers to safeguard GNNs against adversarial attacks. Our contributions can be summarized as follows:

- We provide a unified view of $\ell_1$-based robust graph signal smoothing to justify the improved and closely aligned robustness of representative robust GNNs. Moreover, we reveal their estimation bias, which explains their severe performance degradation under large attack budgets.

- We propose a robust and unbiased graph signal estimator to mitigate the estimation bias in $\ell_1$-based graph signal smoothing and design an efficient Quasi-Newton IRLS algorithm to solve the non-smooth and non-convex estimation problem with a theoretical convergence guarantee.

- The proposed algorithm can be readily unfolded as feature aggregation layers in GNNs, which not only provides clear interpretability but also covers many classic GNNs as special cases.

- Extensive experiments demonstrate that our proposed GNN significantly improves the robustness against large-budget adaptive attacks, e.g., outperforming the best existing model by $16\%$ and $28.6\%$ under local attack of budget $100\%$ and $200\%$ on Cora ML, while maintaining clean accuracy. We also provide comprehensive ablation studies to validate its working mechanism.

## 2 AN ESTIMATION BIAS ANALYSIS OF ROBUST GNNS

In this section, we conduct a preliminary study to evaluate the robustness of representative robust GNNs. Then we establish a unified view to uncover their inherent connections, offering explanations of their improved robustness under small attack budgets and failure under large attack budgets.

**Notation.** Let $\mathcal{G} = \{\mathcal{V}, \mathcal{E}\}$ be a graph with node set $\mathcal{V} = \{v_1, \ldots, v_n\}$ and edge set $\mathcal{E} = \{e_1, \ldots, e_m\}$. The adjacency matrix of $\mathcal{G}$ is denoted as $\boldsymbol{A} \in \{0, 1\}^{n \times n}$ and the graph Laplacian matrix is $\boldsymbol{L} = \boldsymbol{D} - \boldsymbol{A}$. $\boldsymbol{D} = \text{diag}(d_1, \ldots, d_n)$ is the degree matrix where $d_i = |\mathcal{N}(i)|$ and $\mathcal{N}(i)$ is the neighborhood set of $v_i$. The node feature matrix is denoted as $\boldsymbol{F} = [\boldsymbol{f}_1, \ldots, \boldsymbol{f}_n]^\top \in \mathbb{R}^{n \times d}$, and $\boldsymbol{f}^{(0)}$ ($\boldsymbol{F}^{(0)}$) denotes the node feature vector (matrix) before graph smoothing in decoupled GNN models. Let $\Delta \in \{-1, 0, 1\}^{m \times n}$ be the incidence matrix whose $l$-th row denotes the $l$-th edge $e_l = (i, j)$ such that $\Delta_{li} = -1, \Delta_{lj} = 1, \Delta_{lk} = 0 \ \forall k \notin \{i, j\}$. $\tilde{\Delta}$ is its normalized version : $\tilde{\Delta}_{lj} = \Delta_{lj}/\sqrt{d_j}$. For a vector $\boldsymbol{x} \in \mathbb{R}^d$, we use $\ell_1$ penalty to denote either $\|\boldsymbol{x}\|_1 = \sum_i |\boldsymbol{x}_i|$ or $\|\boldsymbol{x}\|_2 = \sqrt{\sum_i \boldsymbol{x}_i^2}$. Note that we use $\ell_2$ penalty to denote $\|\boldsymbol{x}\|_2^2 = \sum_i \boldsymbol{x}_i^2$.

### 2.1 ROBUSTNESS ANALYSIS

To test the robustness of existing GNNs without the false sense of security, we perform a preliminary evaluation of existing robust GNNs against adaptive attacks. We choose various baselines including the undefended MLP, GCN (Kipf & Welling, 2017), some of the most representative defenses in Mujkanovic et al. (2022), and two additional robust models TWIRLS (Yang et al., 2021) and ElasticGNN (Liu et al., 2021). We execute adaptive local evasion topological attacks and test the node classification accuracy on the Cora ML and Citeseer datasets (Mujkanovic et al., 2022). The detailed settings follow Section 4.1. From the results in Figure 1, it can be observed that:

- Among all the selected robust GNNs, only SoftMedian, TWIRLS, and ElasticGNN exhibit notable and closely aligned improvements in robustness whereas other GNNs do not show obvious improvement over undefended GCN.

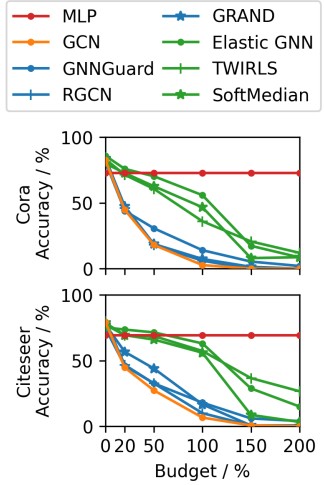

Figure 1: Robustness Analysis.

- SoftMedian, TWIRLS, and ElasticGNN encounter a similar catastrophic performance degradation as the attack budget scales up. At a larger budget, their accuracy easily drops below that of the graph-unaware MLP, indicating their failure in safely exploiting the topology of the data.

## 2.2 A UNIFIED VIEW OF ROBUST ESTIMATION

Our preliminary study provides intriguing observations in Section 2.1, but the underlying reasons behind these phenomena remain obscure. This motivates us to delve into their theoretical understanding and explanation. In this section, we will compare the architectures of those three well-performing GNNs, aiming to reveal their intrinsic connections.

**SoftMedian** (Geisler et al., 2021) substitutes the GCN aggregation for enhanced robustness with the dimension-wise median $\boldsymbol{m}_i \in \mathbb{R}^d$ for all neighbors of each node $i \in \mathcal{V}$. However, the gradient of the median is zero almost everywhere, which is not suitable for the back-propagation training of GNNs. Therefore, the median is approximated as a differentiable weighted sum $\tilde{\boldsymbol{m}}_i = \frac{1}{Z} \sum_{j \in \mathcal{N}(i)} w(\boldsymbol{f}_j, \boldsymbol{m}_i) \boldsymbol{f}_j, \forall i \in \mathcal{V}$, where $\boldsymbol{m}_i$ is the exact non-differentiable dimension-wise median, $\boldsymbol{f}_j$ is the feature vector of the $j$-th neighbor, $w(\boldsymbol{x}, \boldsymbol{y}) = e^{-\beta\|\boldsymbol{x}-\boldsymbol{y}\|_2}$, and $Z = \sum_k w(\boldsymbol{f}_k, \boldsymbol{m}_k)$ is a normalization factor. In this way, the aggregation assigns the largest weights to the neighbors closest to the actual median.

**TWIRLS** (Yang et al., 2021) utilizes the iteratively reweighted least squares (IRLS) algorithm to optimize the objective with parameter $\lambda$, and $\rho(y) = y$ in the default setting:

$$2\lambda \sum_{(i,j)\in\mathcal{E}} \rho(\|\tilde{\boldsymbol{f}}_i - \tilde{\boldsymbol{f}}_j\|_2) + \sum_{i\in\mathcal{V}} \|\tilde{\boldsymbol{f}}_i - \tilde{\boldsymbol{f}}^{(0)}\|_2^2, \ \tilde{\boldsymbol{f}}_i = (1+\lambda d_i)^{-\frac{1}{2}} \boldsymbol{f}_i. \tag{1}$$

**ElasticGNN** (Liu et al., 2021) proposes the elastic message passing which unfolds the proximal alternating predictor-corrector (PAPC) algorithm to minimize the objective with parameter $\lambda_{\{1,2\}}$:

$$\lambda_1 \sum_{(i,j)\in\mathcal{E}} \left\|\frac{\boldsymbol{f}_i}{\sqrt{d_i}} - \frac{\boldsymbol{f}_j}{\sqrt{d_j}}\right\|_p + \lambda_2 \sum_{(i,j)\in\mathcal{E}} \left\|\frac{\boldsymbol{f}_i}{\sqrt{d_i}} - \frac{\boldsymbol{f}_j}{\sqrt{d_j}}\right\|_2^2 + \frac{1}{2} \sum_{i\in\mathcal{V}} \|\boldsymbol{f}_i - \boldsymbol{f}_i^{(0)}\|_2^2, \ p \in \{1, 2\}, \tag{2}$$

**A Unified View of Robust Estimation.** While these three approaches have seemingly different architectures, we provide a unified view of robust estimation to illuminate their inherent connections. First, the objective of TWIRLS in Eq. (1) can be considered as a particular case of ElasticGNN with $\lambda_2 = 0$ and $p = 2$ when neglecting the difference in the node degree normalization. However, TWIRLS and ElasticGNN unroll the iterative minimization into multiple GNN layers. They leverage different optimization solvers, i.e., IRLS and PAPC, which lead to vastly different GNN layers. Second, SoftMedian approximates the computation of medians in a soft way of weighted sums, which can be regarded as approximately solving the dimension-wise median estimation problem (Huber, 2004): $\arg\min_{\boldsymbol{f}_i} \sum_{j\in\mathcal{N}(i)} \|\boldsymbol{f}_i - \boldsymbol{f}_j\|_1$. Therefore, SoftMedian can be regarded as the ElasticGNN with $\lambda_2 = 0$ and $p = 1$. We also note that the SoftMedoid (Geisler et al., 2020) approach also resembles ElasticGNN with $\lambda_2 = 0$ and $p = 2$, and the Total Variation GNN (Hansen & Bianchi, 2023) also utilizes an $\ell_1$ estimator in spectral clustering.

The above analyses reveal that SoftMedian, TWIRLS, and ElasticGNN share the same ideology of $\ell_1$-based robust graph signal estimation, i.e. a similar graph smoothing objective with edge difference penalties $\|\boldsymbol{f}_i - \boldsymbol{f}_j\|_1$ or $\|\boldsymbol{f}_i - \boldsymbol{f}_j\|_2$. However, they adopt different approximation solutions that result in distinct architectural designs. Therefore, this unified view of robust estimation can explain their closely aligned performance despite different specific formulations. Besides, the superiority $\ell_1$-based models over the $\ell_2$-based models such as GCN (Kipf & Welling, 2017), whose graph smoothing objective is essentially $\sum_{(i,j)\in\mathcal{E}} \|\boldsymbol{f}_i/\sqrt{d_i} - \boldsymbol{f}_j/\sqrt{d_j}\|_2^2$ (Ma et al., 2021), can also be understood since $\ell_1$-based graph smoothing mitigates the impact of the outliers (Liu et al., 2021).

## 2.3 BIAS ANALYSIS AND PERFORMANCE DEGRADATION

The unified view of $\ell_1$-based graph smoothing we established in Section 2.2 not only explains their aligned robustness improvement but also provides a perspective to understand their failure under large attack budgets through an estimation bias analysis.

**Bias of $\ell_1$-based Estimation.** In the literature of high-dimensional statistics, it has been well understood that the $\ell_1$ regularization will induce an estimation bias. In the context of de-noising (Donoho, 1995) or variable selection (Tibshirani, 1996), small coefficients $\theta$ are undesirable. To exclude small $\theta$ in the estimation, a soft-thresholding operator can be derived as $\mathbf{S}_\lambda(\theta) = \text{sign}(\theta)\max(|\theta| - \lambda, 0)$. As a result, large $\theta$ are also shrunk by a constant, so the $\ell_1$ estimation is biased towards zero.

A similar bias effect also occurs in graph signal estimation in the presence of adversarial attacks. For example, in TWIRLS (Eq. (1)), edge $e_k = (i,j)$ is reweighted by $w_{ij} = \|\tilde{\boldsymbol{f}}_i - \tilde{\boldsymbol{f}}_j\|_2^{-1}$. After the corresponding graph aggregation $\tilde{\boldsymbol{f}}_i^{(k+1)} = \sum_{j \in \mathcal{N}(i)} w_{ij} \tilde{\boldsymbol{f}}_j^{(k)}$, edge $e_k = (i,j)$ will shrink the edge difference $\tilde{\boldsymbol{f}}_i - \tilde{\boldsymbol{f}}_j$ by the unit vector $\boldsymbol{u}_{\tilde{\boldsymbol{f}}_i - \tilde{\boldsymbol{f}}_j}$. Consequently, every heterophilic edge added will induce a constant bias that can be accumulated and amplified when the attack budget scales up.

**Numerical Simulation.** To provide a more intuitive illustration of the estimation bias of $\ell_1$-based models, we simulate a mean estimation problem on synthetic data since most message passing schemes in GNNs essentially estimate the mean of neighboring node features. In the context of mean estimation, the bias is measured as the distances between different mean estimators and the true mean. We firstly generated clean samples $\{\boldsymbol{x}_i\}_{i=1}^n$ (blue dots) and the outlier samples $\{\boldsymbol{x}_i\}_{i=n+1}^{n+m}$ (red dots) from 2-dimensional Gaussian distributions, $\mathcal{N}((0,0),1)$ and $\mathcal{N}((8,8),0.5)$, respectively. We calculate the mean of clean samples $\frac{1}{n}\sum_{i=1}^n \boldsymbol{x}_i$ as the ground truth of the mean estimator. Then we estimate the mean of all the samples by solving $\arg\min_{\boldsymbol{z}} \sum_{i=1}^{n+m} \eta(\boldsymbol{z} - \boldsymbol{x}_i)$ using the Weiszfeld method (Candès et al., 2008; Beck & Sabach, 2015), where $\eta(\cdot)$ can take different penalties such as $\ell_2$ penalty $\|\cdot\|_2^2$ and $\ell_1$ penalty $\|\cdot\|_2$.

In Figure 2, we visualize the generated clean samples and outliers, as well as the ground truth means and the mean estimators with $\eta(\cdot) = \|\cdot\|_2^2$ or $\|\cdot\|_2$ under different outlier ratios (10%, 25%, 40%). The results show that the $\ell_2$-based estimator deviates far from the true mean, while the $\ell_1$-based estimator is more resistant to outliers, which explains why $\ell_1$-based methods exhibit stronger robustness. However, as the ratio of outliers escalates, the $\ell_1$-based estimator encounters a greater shift from the true mean due to the accumulated bias caused by outliers. This observation explains why $\ell_1$-based graph smoothing models suffer from catastrophic degradation under large attack budgets.

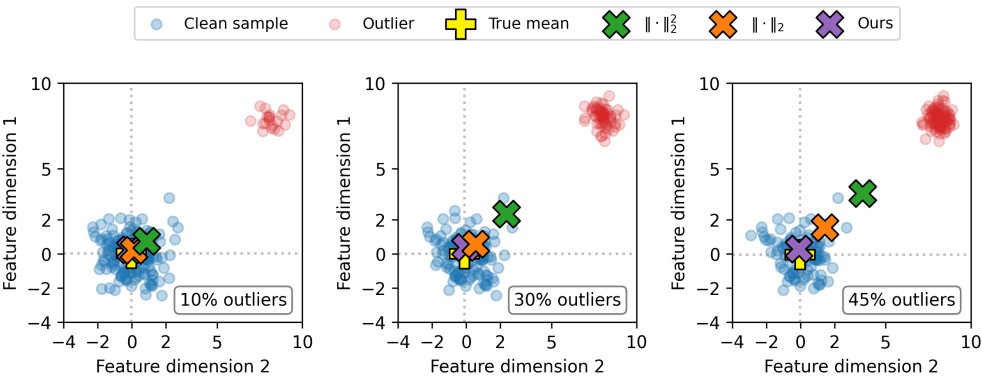

Figure 2: Different mean estimators in the presence of outliers.

# 3 ROBUST UNBIASED AGGREGATION

In this section, we design a robust unbiased estimator to reduce the bias in graph signal estimation and propose an efficient second-order IRLS algorithm to be unrolled as the robust unbiased aggregation in GNNs with a theoretical convergence guarantee.

## 3.1 ROBUST AND UNBIASED GRAPH SIGNAL ESTIMATOR

Our study and analysis in Section 2 have shown that while $\ell_1$-based methods outperform $\ell_2$-based methods in robustness, they still suffer from the accumulated estimation bias, leading to severe performance degradation under large perturbation budgets. This motivates us to design a robust and unbiased graph signal estimator that derives unbiased robust aggregation for GNNs with stronger resilience to attacks.

Theoretically, the estimation bias in Lasso regression has been discovered and analyzed in high-dimensional statistics (Zou, 2006). Statisticians have proposed adaptive Lasso (Zou, 2006) and many non-convex penalties such as Smoothly Clipped Absolute Deviation (SCAD) (Fan & Li, 2001)

and Minimax Concave Penalty (MCP) (Zhang, 2010) to alleviate this bias. Motivated by these advancements, we propose a Robust and Unbiased Graph signal Estimator (RUGE) as follows:

$$\arg\min_{\boldsymbol{F}} \mathcal{H}(\boldsymbol{F}) = \sum_{(i,j)\in\mathcal{E}} \rho_\gamma \left( \left\| \frac{\boldsymbol{f}_i}{\sqrt{d_i}} - \frac{\boldsymbol{f}_j}{\sqrt{d_j}} \right\|_2 \right) + \lambda \sum_{i\in\mathcal{V}} \|\boldsymbol{f}_i - \boldsymbol{f}_i^{(0)}\|_2^2, \tag{3}$$

where $\rho_\gamma(y)$ denotes the function that penalizes the feature differences on edges by the non-convex MCP:

$$\rho_\gamma(y) = \begin{cases} y - \frac{y^2}{2\gamma} & \text{if } y < \gamma \\ \frac{\gamma}{2} & \text{if } y \geq \gamma \end{cases} \tag{4}$$

As shown in Figure 3, MCP closely approximates the $\ell_1$ penalty when $y$ is small since the quadratic term $\frac{y^2}{2\gamma}$ is negligible, and it becomes a constant value when $y$ is large. This transition can be adjusted by the thresholding parameter $\gamma$. When $\gamma$ approaches infinity, the penalty $\rho_\gamma(y)$ reduces to the $\ell_1$ penalty. Conversely, when $\gamma$ is very small, the "valley" of $\rho_\gamma$ near zero is exceptionally sharp, so $\rho_\gamma(y)$ approaches the $\ell_0$ penalty and becomes a constant for a slightly larger $y$. This enables RUGE to promote smoothing through reliable edges connecting homophilic nodes and suppress smoothing on edges the node differences exceeding the threshold $\gamma$. This not only mitigates the estimation bias against outliers but also main-

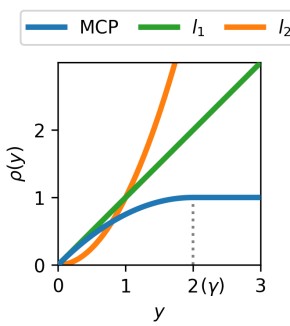

Figure 3: Different penalties.

tains the estimation accuracy in the absence of outliers. The simulation in Figure 2 verifies that our proposed estimator ($\eta(\boldsymbol{x}) \coloneqq \rho_\gamma(\|\boldsymbol{x}\|_2)$) can recover the true mean despite the increasing outlier ratio when the outlier ratio is below the theoretical optimal breakdown point.

### 3.2 QUASI-NEWTON IRLS

Despite the advantages discussed above, the proposed RUGE in Eq. (3) is non-smooth and non-convex, which results in challenges for deriving efficient numerical solutions that can be readily unfolded as neural network layers. In the literature, researchers have developed optimization algorithms for MCP-related problems, such as the Alternating Direction Multiplier Method (ADMM) and Newton-type algorithms (Fan & Li, 2001; Zhang, 2010; Varma et al., 2019). However, due to their excessive computation and memory requirements as well as the incompatibility with back-propagation training, these algorithms are not well-suited for the construction of feature aggregation layers and employment in GNNs. To solve these challenges, we derive an efficient Quasi-Newton Iteratively Reweighted Least Squares algorithm (QN-IRLS) to solve the estimation problem in Eq. (3).

**IRLS.** Before stepping into our QN-IRLS, we first introduce the main idea of iteratively reweighted least squares (IRLS) (Holland & Welsch, 1977) and analyze its weakness in convergence. IRLS aims to circumvent the non-smooth $\mathcal{H}(\boldsymbol{F})$ in Eq. (3) by computing its quadratic upper bound $\hat{\mathcal{H}}^{(k)}$ based on $\boldsymbol{F}^{(k)}$ in each iteration $k$ and optimizing $\hat{\mathcal{H}}^{(k)}$ which follows

$$\hat{\mathcal{H}}^{(k)}(\boldsymbol{F}) = \sum_{(i,j)\in\mathcal{E}, i\neq j} \boldsymbol{W}_{ij}^{(k)} \left\| \frac{\boldsymbol{f}_i}{\sqrt{d_i}} - \frac{\boldsymbol{f}_j}{\sqrt{d_j}} \right\|_2^2 + \lambda \sum_{i\in\mathcal{V}} \|\boldsymbol{f}_i - \boldsymbol{f}_i^{(0)}\|_2^2, \tag{5}$$

where $\boldsymbol{W}_{ij}^{(k)} = \boldsymbol{1}_{i\neq j} \frac{d\rho_\gamma(y_{ij})}{dy_{ij}^2}\big|_{y_{ij}=y_{ij}^{(k)}}$[1], $\rho_\gamma(\cdot)$ is the MCP function and $y_{ij}^{(k)} = \left\| \frac{\boldsymbol{f}_i^{(k)}}{\sqrt{d_i}} - \frac{\boldsymbol{f}_j^{(k)}}{\sqrt{d_j}} \right\|_2$. For the detailed proof of the upper bound, please refer to Lemma 1 in Appendix B. Then, each iterative step of IRLS can be formulated as the first-order gradient descent iteration for $\hat{\mathcal{H}}^{(k)}(\boldsymbol{F})$:

$$\boldsymbol{F}^{(k+1)} = \boldsymbol{F}^{(k)} - \eta\nabla\hat{\mathcal{H}}^{(k)}(\boldsymbol{F}^{(k)}) = \boldsymbol{F}^{(k)} - \eta\left(2(\hat{\boldsymbol{Q}}^{(k)} - \boldsymbol{W}^{(k)}\odot\tilde{\boldsymbol{A}})\boldsymbol{F}^{(k)} - 2\lambda\boldsymbol{F}^{(0)}\right), \tag{6}$$

where $\hat{\boldsymbol{Q}}^{(k)} = 2(\text{diag}(\boldsymbol{q}^{(k)}) + \lambda\boldsymbol{I})$, $\boldsymbol{q}_m^{(k)} = \sum_j \boldsymbol{W}_{mj}^{(k)}\boldsymbol{A}_{mj}/d_m$, and $\eta$ is the update step size. The convergence condition of Eq. (6) is given in Theorem 1, whose proof is presented in Appendix B.

---

[1] $\boldsymbol{W}_{ij}$ is defined as $\frac{d\rho(y)}{dy^2}\big|_{y=y_{ij}^{(k)}}$ so that the quadratic upper bound $\hat{\mathcal{H}}^{(k)}$ is tight at $\boldsymbol{F}^{(k)}$ according to Lemma 3. The diagonal terms of $\boldsymbol{W}$ are set to zero to avoid undefined derivative of $\frac{d\rho(y)}{dy^2}\big|_{y=0}$ as discussed in Remark 2.

**Theorem 1.** *If $\boldsymbol{F}^{(k)}$ follows the update rule in Eq. (6) where $\rho$ defining $\boldsymbol{W}$ satisfies that $\frac{d\rho(y)}{dy^2}$ is non-decreasing $\forall y \in (0, \infty)$, then a sufficient condition for $\mathcal{H}(\boldsymbol{F}^{(k+1)}) \leq \mathcal{H}(\boldsymbol{F}^{(k)})$ is that the step size $\eta$ satisfies $0 < \eta \leq \|diag(\boldsymbol{q}^{(k)}) - \boldsymbol{W}^{(k)} \odot \tilde{\boldsymbol{A}} + \lambda \boldsymbol{I}\|_2^{-1}$.*

**Quasi-Newton IRLS.** Theorem 1 suggests the difficulty in the proper selection of stepsize for (first-order) IRLS due to its non-trivial dependency on the graph ($\tilde{\boldsymbol{A}}$) and the dynamic terms ($\boldsymbol{q}^{(k)}$ and $\boldsymbol{W}^{(k)}$) [2]. The dilemma is that a small stepsize will lead to slow convergence but a large step easily causes divergence and instability as verified by our experiments in Section 4.3 (Figrue 5), which reveals its critical shortcoming for the construction of reliable GNN aggregation layers.

To overcome this limitation, we aim to propose a second-order Newton method, $\boldsymbol{F}^{(k+1)} = \boldsymbol{F}^{(k)} - (\nabla^2 \hat{\mathcal{H}}^{(k)}(\boldsymbol{F}^{(k)}))^{-1} \nabla \hat{\mathcal{H}}^{(k)}(\boldsymbol{F}^{(k)})$, to achieve faster convergence and stepsize-free hyperparameter tuning by better capturing the geometry of the optimization landscape. However, obtaining the analytic expression for the inverse Hessian matrix $(\nabla^2 \hat{\mathcal{H}}^{(k)}(\boldsymbol{F}^{(k)}))^{-1} \in \mathbb{R}^{n \times n}$ is intractable and the numerical solution requires expensive computation for large graphs. To resolve this challenge, we propose a novel Quasi-Newton IRLS algorithm (QN-IRLS) that approximates the Hessian matrix $\nabla^2 \hat{\mathcal{H}}^{(k)}(\boldsymbol{F}^{(k)}) = 2(diag(\boldsymbol{q}^{(k)}) - \boldsymbol{W}^{(k)} \odot \tilde{\boldsymbol{A}} + \lambda \boldsymbol{I})$ by the diagonal matrix $\hat{\boldsymbol{Q}}^{(k)} = 2(diag(\boldsymbol{q}^{(k)}) + \lambda \boldsymbol{I})$ such that the inverse is trivial. The proposed QN-IRLS iterates as follows:

$$\boldsymbol{F}^{(k+1)} = 2(\hat{\boldsymbol{Q}}^{(k)})^{-1} \left( (\boldsymbol{W}^{(k)} \odot \tilde{\boldsymbol{A}})\boldsymbol{F}^{(k)} + \lambda \boldsymbol{F}^{(0)} \right), \tag{7}$$

where $2(\hat{\boldsymbol{Q}}^{(k)})^{-1} = (diag(\boldsymbol{q}^{(k)}) + \lambda \boldsymbol{I})^{-1}$ automatically adjusts the per-coordinate stepsize according to the local geometry of the optimization landscape, $\boldsymbol{q}^{(k)}$ and $\boldsymbol{W}^{(k)}$ are defined as in Eq. (5) and (6). In this way, QN-IRLS provides faster convergence without needing to select a stepsize. The convergence is guaranteed by Theorem 2 with the proof in Appendix B.

**Theorem 2.** *If $\boldsymbol{F}^{(k+1)}$ follows update rule in Eq. (7), where $\rho$ satisfies that $\frac{d\rho(y)}{dy^2}$ is non-decreasing $\forall y \in (0, \infty)$, it is guaranteed that $\mathcal{H}(\boldsymbol{F}^{(k+1)}) \leq \mathcal{H}(\boldsymbol{F}^{(k)})$.*

### 3.3 GNN with Robust Unbiased Aggregation

The proposed QN-IRLS provides an efficient algorithm to optimize the RUGE in Eq. (3) with a theoretical convergence guarantee. Instantiated with $\rho = \rho_\gamma$, each iteration in QN-IRLS (Eq. (7)) can be used as one layer in robust GNNs, which yields the Robust Unbiased Aggregation (RUNG):

$$\boldsymbol{F}^{(k+1)} = (diag(\boldsymbol{q}^{(k)}) + \lambda \boldsymbol{I})^{-1} \left( (\boldsymbol{W}^{(k)} \odot \tilde{\boldsymbol{A}})\boldsymbol{F}^{(k)} + \lambda \boldsymbol{F}^{(0)} \right), \tag{8}$$

where $\boldsymbol{q}_m^{(k)} = \sum_j \boldsymbol{W}_{mj}^{(k)} \boldsymbol{A}_{mj}/d_m$, $\boldsymbol{W}_{ij}^{(k)} = \mathbf{1}_{i \neq j} \max(0, \frac{1}{2y_{ij}^{(k)}} - \frac{1}{2\gamma})$ and $y_{ij}^{(k)} = \left\| \frac{\boldsymbol{f}_i^{(k)}}{\sqrt{d_i}} - \frac{\boldsymbol{f}_j^{(k)}}{\sqrt{d_j}} \right\|_2$.

**Interpretability.** The proposed RUNG can be interpreted intuitively with edge reweighting. In Eq. (8), the normalized adjacency matrix $\tilde{\boldsymbol{A}}$ is reweighted by $\boldsymbol{W}^{(k)}$, where $\boldsymbol{W}_{ij}^{(k)} = \frac{d\rho(y)}{dy^2}\big|_{y=y_{ij}^{(k)}}$. It is shown in Figure 4 that $\boldsymbol{W}_{ij}$ becomes zero for any edge $e_k = (i, j)$ with a node difference $y_{ij}^{(k)} \geq \gamma$, thus pruning suspicious edges. This implies RUNG's strong robustness under large-budget adversarial attacks. With the inclusion of the skip connection $\boldsymbol{F}^{(0)}$, $diag(\boldsymbol{q}^{(k)}) + \lambda \boldsymbol{I}$ can be seen as a normalizer of the layer output.

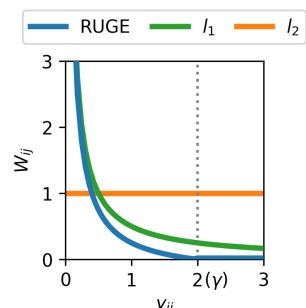

Figure 4: $\frac{d\rho(y)}{dy^2}$ of different penalties. RUNG uses RUGE.

**Relations with Existing GNNs.** RUNG can adopt different $\rho$ allowed by Theorem 2, thus connecting many classic GNNs as special cases. When $\rho(y) = y$, the objective of RUGE is equivalent to ElasticGNN with $p = 2$, which further relates to SoftMedian and TWIRLS due to their inherent resemblance as $\ell_1$-based graph smoothing. When $\rho(y) = y^2$, RUNG in Eq. (8) reduces to APPNP (Gasteiger et al., 2018) ($\boldsymbol{F}^{(k+1)} = \frac{1}{1+\lambda} \tilde{\boldsymbol{A}} \boldsymbol{F}^{(k)} + \frac{\lambda}{1+\lambda} \boldsymbol{F}^{(0)}$) and GCN ($\boldsymbol{F}^{(k+1)} = \tilde{\boldsymbol{A}} \boldsymbol{F}^{(k)}$) if chosing $\lambda = 0$.

---

[2] A related work (Yang et al., 2021) adopts IRLS algorithm to optimize the problem in Eq. (1). A preconditioned version is proposed to handle the unnormalized graph Laplacian, but its step size needs to satisfy $\eta \leq \|\Delta^\top \boldsymbol{\Gamma}^{(k)} \Delta + \lambda \boldsymbol{I}\|_2^{-1}$ as shown in Lemma 3.3 of Yang et al. (2021), which is also expensive to estimate.

## 4 EXPERIMENT

In this section, we perform comprehensive experiments to validate the robustness of the proposed RUNG. Besides, ablation studies show the convergence and the defense mechanism of RUNG.

### 4.1 EXPERIMENT SETTING

**Datasets.** We test our RUNG with the node classification task on two widely used real-world citation networks, Cora ML and Citeseer (Sen et al., 2008). We adopt the data split of $80\%$ training, $10\%$ validation, and $10\%$ testing, and report the classification accuracy of the attacked nodes following Mujkanovic et al. (2022). Each experiment is averaged over $5$ different random splits.

**Baselines.** To evaluate the performance of RUNG, we compare it to 12 other representative baselines. Among them, MLP, GCN (Kipf & Welling, 2017), APPNP (Gasteiger et al., 2018), and GAT (Veličković et al., 2017) are undefended vanilla models. GNNGuard (Zhang & Zitnik, 2020), RGCN (Zhu et al., 2019), GRAND (Feng et al., 2020), ProGNN (Jin et al., 2020), Jaccard-GCN (Wu et al., 2019), and SVD-GCN (Entezari et al., 2020) are representative robust GNNs. Besides, Soft-Median and TWIRLS are representative approaches with $\ell_1$-based graph smoothing [3]. We also evaluate a variant of TWIRLS with a special thresholding attention (TWIRLS-T). For RUNG, we test two variants: default RUNG (Eq. (8)) and RUNG-$\ell_1$ with $\ell_1$ penalty ($\rho(y) = y$).

**Hyperparameters.** The model hyperparameters including learning rate, weight decay, and dropout rate are tuned as in Mujkanovic et al. (2022). Other hyperparameters follow the settings in the original papers. RUNG uses an MLP connected to 10 graph aggregation layers following the decoupled GNN architecture of APPNP, $\hat{\lambda} = \frac{1}{1+\lambda}$ is tuned in $\{0.7, 0.8, 0.9\}$, and $\gamma$ tuned in $\{0.5, 1, 2, 3, 5\}$ and the hyperparameter combination that yields the best robustness without a notable impact (smaller than $1\%$) on the clean accuracy is chosen, following the setting in Bojchevski & Günnemann (2019).

**Attack setting.** We use the PGD attack (Xu et al., 2019) to execute the adaptive evasion topology attack since it delivers the strongest attack in most settings (Mujkanovic et al., 2022). The adversarial attacks aim to misclassify specific target nodes (*local attack*) or the entire set of test nodes (*global attack*). For *local attack*, we randomly select 20 target nodes for each data split. To avoid a false sense of robustness, our adaptive attacks directly target the victim model instead of the surrogate model. In particular, we execute the adaptive attack after all the hyperparameters are fixed.

### 4.2 ADVERSARIAL ROBUSTNESS

Here we evaluate the the performance of RUNG against the baselines under different settings. The results of local and global adaptive attacks on Cora ML are presented in Table 1 and Table 2, while those on Citeseer are presented in Table 4 and Table 3 in Appendix D due to space limits. We summarize the following analysis from Cora ML, noting that the same observations apply to Citeseer.

- Under adaptive attacks, many existing defenses are not significantly more robust than undefended models. The $\ell_1$-based models such as TWIRLS, SoftMedian, and RUNG-$\ell_1$ demonstrate considerable and closely aligned robustness under both local and global attacks, which supports our unified $\ell_1$-based robust view analysis in Section 2.2.

- When the attack budget is large, all baselines encounter a serious performance drop and underperform MLPs by significant margins. For instance, under local attack with budgets of 100%, 150%, and 200%, the best GNN baseline underperforms MLP by 17.9%, 28.6%, and 31.9%.

- Our RUNG exhibits significant improvements over existing approaches when the attack budget is large. Under local attacks, RUNG outperforms the best GNN baseline by 16.0%, 25.3%, and 28.6% with attack budgets of 100%, 150%, and 200%. Note that RUNG exhibits stable performance with the increase of attack budget. Under global attacks, RUNG outperforms the best GNN baseline by 2.3%, 4.0%, and 4.3% with budgets of 20%, 30%, and 40%.

- When there is no attack, RUNG largely preserves an excellent clean performance. RUNG also achieves state-of-the-art performance under small attack budgets.

- Local attacks are stronger than global attacks since local attacks concentrate on targeted nodes. The robustness improvement of RUNG appears to be more remarkable in local attacks.

---

[3]We do not include ElasticGNN because it is still unclear how to attack it adaptively due to its special incident matrix formulation (Liu et al., 2021). In the preliminary study (Section 2.1), we evaluate the robustness of ElasticGNN following the unit test setting proposed in Mujkanovic et al. (2022).

Table 1: Adaptive local attack on Cora ML. The **best** and second are marked.

| Model | 0% | 20% | 50% | 100% | 150% | 200% |
|---|---|---|---|---|---|---|
| MLP | $72.6 \pm 6.4$ | $72.6 \pm 6.4$ | $72.6 \pm 6.4$ | $\mathbf{72.6 \pm 6.4}$ | $\mathbf{72.6 \pm 6.4}$ | $\mathbf{72.6 \pm 6.4}$ |
| GCN | $82.7 \pm 4.9$ | $40.7 \pm 10.2$ | $12.0 \pm 6.2$ | $2.7 \pm 2.5$ | $0.0 \pm 0.0$ | $0.0 \pm 0.0$ |
| APPNP | $\mathbf{84.7 \pm 6.8}$ | $50.0 \pm 13.0$ | $27.3 \pm 6.5$ | $14.0 \pm 5.3$ | $3.3 \pm 3.0$ | $0.7 \pm 1.3$ |
| GAT | $80.7 \pm 10.0$ | $30.7 \pm 16.1$ | $16.0 \pm 12.2$ | $11.3 \pm 4.5$ | $1.3 \pm 1.6$ | $2.0 \pm 1.6$ |
| GNNGuard | $82.7 \pm 6.7$ | $44.0 \pm 11.6$ | $30.7 \pm 11.6$ | $14.0 \pm 6.8$ | $5.3 \pm 3.4$ | $2.0 \pm 2.7$ |
| RGCN | $84.6 \pm 4.0$ | $46.0 \pm 9.3$ | $18.0 \pm 8.1$ | $6.0 \pm 3.9$ | $0.0 \pm 0.0$ | $0.0 \pm 0.0$ |
| GRAND | $84.0 \pm 6.8$ | $47.3 \pm 9.0$ | $18.7 \pm 9.1$ | $7.3 \pm 4.9$ | $1.3 \pm 1.6$ | $0.0 \pm 0.0$ |
| ProGNN | $\mathbf{84.7 \pm 6.2}$ | $47.3 \pm 10.4$ | $21.3 \pm 7.8$ | $4.0 \pm 2.5$ | $0.0 \pm 0.0$ | $0.0 \pm 0.0$ |
| Jaccard-GCN | $81.3 \pm 5.0$ | $46.0 \pm 6.8$ | $17.3 \pm 4.9$ | $4.7 \pm 3.4$ | $0.7 \pm 1.3$ | $0.7 \pm 1.3$ |
| SoftMedian | $80.0 \pm 10.2$ | $72.7 \pm 13.7$ | $62.7 \pm 12.7$ | $46.7 \pm 11.0$ | $8.0 \pm 4.5$ | $8.7 \pm 3.4$ |
| TWIRLS | $83.3 \pm 7.3$ | $71.3 \pm 8.6$ | $60.7 \pm 11.0$ | $36.0 \pm 8.8$ | $20.7 \pm 10.4$ | $12.0 \pm 6.9$ |
| TWIRLS-T | $82.0 \pm 4.5$ | $70.7 \pm 4.4$ | $62.7 \pm 7.4$ | $54.7 \pm 6.2$ | $44.0 \pm 11.2$ | $40.7 \pm 11.8$ |
| RUNG-$\ell_1$ (ours) | $84.0 \pm 6.8$ | $72.7 \pm 7.1$ | $62.7 \pm 11.2$ | $53.3 \pm 8.2$ | $22.0 \pm 9.3$ | $14.0 \pm 7.4$ |
| RUNG (ours) | $84.0 \pm 5.3$ | $\mathbf{75.3 \pm 6.9}$ | $\mathbf{72.7 \pm 8.5}$ | $70.7 \pm 10.6$ | $\mathbf{69.3 \pm 9.8}$ | $69.3 \pm 9.0$ |

Table 2: Adaptive global attack on Cora ML. The **best** and second are marked.

| Model | Clean | 5% | 10% | 20% | 30% | 40% |
|---|---|---|---|---|---|---|
| MLP | $65.0 \pm 1.0$ | $65.0 \pm 1.0$ | $65.0 \pm 1.0$ | $65.0 \pm 1.0$ | $65.0 \pm 1.0$ | $65.0 \pm 1.0$ |
| GCN | $85.0 \pm 0.4$ | $75.3 \pm 0.5$ | $69.6 \pm 0.5$ | $60.9 \pm 0.7$ | $54.2 \pm 0.6$ | $48.4 \pm 0.5$ |
| APPNP | $\mathbf{86.3 \pm 0.4}$ | $75.8 \pm 0.5$ | $69.7 \pm 0.7$ | $60.3 \pm 0.9$ | $53.8 \pm 1.2$ | $49.0 \pm 1.6$ |
| GAT | $83.5 \pm 0.5$ | $75.8 \pm 0.8$ | $71.2 \pm 1.2$ | $65.0 \pm 0.9$ | $60.5 \pm 0.9$ | $56.7 \pm 0.9$ |
| GNNGuard | $83.1 \pm 0.7$ | $74.6 \pm 0.7$ | $70.2 \pm 1.0$ | $63.1 \pm 1.1$ | $57.5 \pm 1.6$ | $51.0 \pm 1.2$ |
| RGCN | $85.7 \pm 0.4$ | $75.0 \pm 0.8$ | $69.1 \pm 0.4$ | $59.8 \pm 0.7$ | $52.8 \pm 0.7$ | $46.1 \pm 0.7$ |
| GRAND | $86.1 \pm 0.7$ | $76.2 \pm 0.8$ | $70.7 \pm 0.7$ | $61.6 \pm 0.7$ | $56.7 \pm 0.8$ | $51.9 \pm 0.9$ |
| ProGNN | $85.6 \pm 0.5$ | $76.5 \pm 0.7$ | $71.0 \pm 0.5$ | $63.0 \pm 0.7$ | $56.8 \pm 0.7$ | $51.3 \pm 0.6$ |
| Jaccard-GCN | $83.7 \pm 0.7$ | $73.9 \pm 0.5$ | $68.3 \pm 0.7$ | $60.0 \pm 1.1$ | $54.0 \pm 1.7$ | $49.1 \pm 2.4$ |
| SoftMedian | $85.0 \pm 0.7$ | $78.6 \pm 0.3$ | $75.5 \pm 0.9$ | $69.5 \pm 0.5$ | $62.8 \pm 0.8$ | $58.1 \pm 0.7$ |
| TWIRLS | $84.2 \pm 0.6$ | $77.3 \pm 0.8$ | $72.9 \pm 0.3$ | $66.9 \pm 0.2$ | $62.4 \pm 0.6$ | $58.7 \pm 1.1$ |
| TWIRLS-T | $82.8 \pm 0.5$ | $76.8 \pm 0.6$ | $73.2 \pm 0.4$ | $67.7 \pm 0.4$ | $63.8 \pm 0.2$ | $60.8 \pm 0.3$ |
| RUNG-$\ell_1$ (ours) | $85.8 \pm 0.5$ | $78.4 \pm 0.4$ | $74.3 \pm 0.3$ | $68.1 \pm 0.6$ | $63.5 \pm 0.7$ | $59.8 \pm 0.8$ |
| RUNG (ours) | $84.6 \pm 0.5$ | $\mathbf{78.9 \pm 0.4}$ | $\mathbf{75.7 \pm 0.2}$ | $\mathbf{71.8 \pm 0.4}$ | $\mathbf{67.8 \pm 1.3}$ | $\mathbf{65.1 \pm 1.2}$ |

## 4.3 ABLATION STUDY

**Convergence.** To verify the advantage of our QN-IRLS method (Eq (7)) over the first-order IRLS (Eq (6)), we show the objective $\mathcal{H}$ on each layer in Figure 5. It can be observed that our stepsize-free QN-IRLS captures the landscape well, demonstrating the best convergence as discussed in Section 3.

**Estimation Bias.** The bias effect in $\ell_1$-based GNNs and the unbiasedness of our RUNG can be empirically verified. We quantify the bias with $\sum_{i \in \mathcal{V}} \|\boldsymbol{f}_i - \boldsymbol{f}_i^\star\|_2^2$, where $\boldsymbol{f}_i^\star$ and $\boldsymbol{f}_i$ denote the aggregated feature on the clean graph and attacked graph, respectively. As shown in Figure 6, when the budget scales up, $\ell_1$ GNNs exhibit a notable bias, whereas RUNG has nearly zero bias.

**Defense Mechanism** To further investigate how our defense takes effect, we analyze the edges added under adaptive attacks. The distribution of the node feature differences $\left\|\boldsymbol{f}_i/\sqrt{d_i} - \boldsymbol{f}_j/\sqrt{d_j}\right\|_2$ of attacked edges is shown in Figure 7 for different graph signal estimators. It can be observed that our RUNG forces the attacker to focus on the edges with a small feature difference because otherwise the attacked edges will be pruned. Therefore, the attacks become less influential, which explains why RUNG demonstrates outstanding robustness.

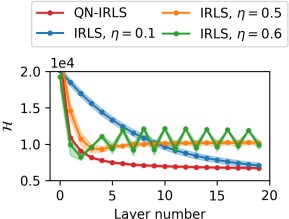

Figure 5: Convergence of our QN-IRLS compared to IRLS.

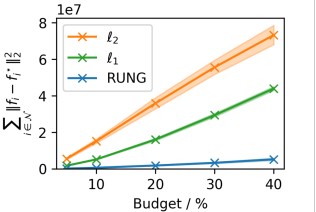

Figure 6: Bias induced by different attack budgets.

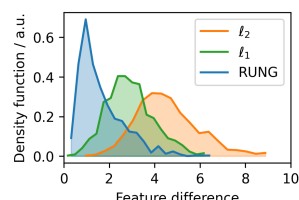

Figure 7: Distribution of feature difference on attacked edges.

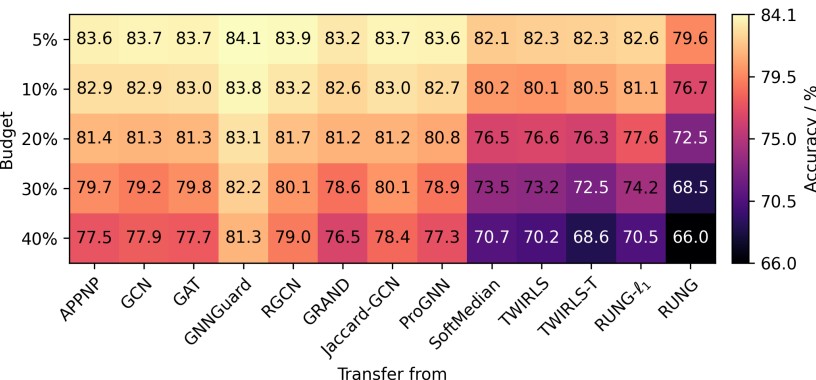

Figure 8: Transfer global attack from different surrogate models to our RUNG on Cora ML.

**Transfer Attacks.** In addition to the adaptive attack, we also conduct a set of transfer attacks that take every baseline GNN as the surrogate model to comprehensively test the robustness of RUNG, following the unit test attack protocol proposed in Mujkanovic et al. (2022). We summarize the results on Cora ML in Figure 8 and leave the results on Citeseer in Figure 10 in Appendix D due to space limits. All transfer attacks are weaker than the adaptive attack in Section 4.2, which validates that our adaptive attack is executed correctly and verifies that RUNG is not falsely robust. Note that the attack transferred from RUNG model is slightly weaker than the adaptive attack since the surrogate and victim RUNG models have different model parameters in the transfer attack setting.

## 5 RELATED WORK

To the best of our knowledge, although there are works unifying existing GNNs from a graph signal denoising perspective (Ma et al., 2021), no work has been dedicated to uniformly understand the robustness and limitations of robust GNNs such as SoftMedian (Geisler et al., 2021), Soft-Medoid (Geisler et al., 2020), TWIRLS (Yang et al., 2021), ElasticGNN (Liu et al., 2021), and TVGNN (Hansen & Bianchi, 2023) from the $\ell_1$ robust statistics and bias analysis perspectives. To mitigate the estimation bias, MCP penalty is promising since it is well known for its near unbiasedness property (Zhang, 2010) and has been applied to the graph trend filtering problem (Varma et al., 2019) to promote piecewise signal modeling, but their robustness is unexplored. Nevertheless, other robust GNNs have utilized alternative penalties that might alleviate the bias effect. For example, GNNGuard (Zhang & Zitnik, 2020) prunes the edges whose cosine similarity is too small. Another example is that TWIRLS (Yang et al., 2021) with a thresholding penalty can also exclude edges using graph attention. However, the design of their edge weighting or graph attention is heuristic-based and exhibits suboptimal performance compared to the RUNG we propose.

## 6 CONCLUSION

In this work, we propose a unified view of $\ell_1$ robust graph smoothing to uniformly understand the robustness and limitations of representative robust GNNs. The established view not only justifies their improved and closely aligned robustness but also explains their severe performance degradation under large attack budgets by an accumulated estimation bias analysis. To mitigate the estimation bias, we propose a robust and unbiased graph signal estimator. To solve this non-trivial estimation problem, we design a novel and efficient Quasi-Newton IRLS algorithm that can better capture the landscape of the optimization problem and converge stably with a theoretical guarantee. This algorithm is unfolded into an interpretable GNN with Robust Unbiased Aggregation (RUNG). As verified by our experiments, RUNG provides the best performance under strong adaptive attacks and overcomes performance degradation under large attack budgets. Furthermore, RUNG also covers many classic GNNs as special cases. Most importantly, this work provides a deeper understanding of existing approaches and reveals a principled direction for designing robust GNNs.

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

# A  BIAS ACCUMULATION OF $\ell_1$ MODELS

## A.1  DETAILS OF THE NUMERICAL SIMULATION SETTINGS

In section 2, we conducted a numerical simulation of mean estimation on synthetic data $\boldsymbol{x}_i$. The mean estimators are formulated as minimization operators

$$\bar{\boldsymbol{z}} = \arg\min_{\boldsymbol{z}} \sum_i^{n+m} \eta(\boldsymbol{z} - \boldsymbol{x}_i), \tag{9}$$

where $n$ is the number of clean samples and $m$ is the number of adversarial samples.

$\ell_1$ **estimator.** The $\ell_1$ estimator ($\eta(\boldsymbol{y}) := \|\boldsymbol{y}\|_2$), essentially is the geometric median. We adopted the Weiszfeld method to iteratively reweight $\boldsymbol{z}$ to minimize the objective, following

$$\boldsymbol{z}^{(k+1)} = \frac{\sum_i w_i^{(k)} \boldsymbol{x}_i}{\sum_i w_i^{(k)}}, \tag{10}$$

where $w_i^{(k)} = \frac{1}{\|\boldsymbol{z}^{(k)} - \boldsymbol{x}_i\|_2}$. This can be seen as a gradient descent step of $\boldsymbol{z}^{(k+1)} = \boldsymbol{z}^{(k)} - \alpha\nabla_{\boldsymbol{z}} \sum_i \|\boldsymbol{z} - \boldsymbol{x}_i\|_2 = \boldsymbol{z}^{(k+1)} - \alpha \sum_i \frac{\boldsymbol{z}^{(k)} - \boldsymbol{x}_i}{\|\boldsymbol{z}^{(k)} - \boldsymbol{x}_i\|_2}$. Taking $\alpha = \frac{1}{\sum_i w_i^{(k)}}$ instantly yields Eq. (10).

**MCP-based estimator.** We therefore adopt a similar approach for the MCP-based estimator ("Ours" in Fig. Figure 2), where $\eta(\boldsymbol{y}) := \rho_\gamma(\boldsymbol{y})$:

$$\boldsymbol{z}^{(k+1)} = \boldsymbol{z}^{(k)} - \alpha\nabla_{\boldsymbol{z}} \sum_i \rho_\gamma(\|\boldsymbol{z} - \boldsymbol{x}_i\|_2) \tag{11}$$

$$= \boldsymbol{z}^{(k)} - \alpha \sum_i \max(0, \frac{1}{\|\boldsymbol{z}^{(k)} - \boldsymbol{x}_i\|_2} - \frac{1}{\gamma})(\boldsymbol{z}^{(k)} - \boldsymbol{x}_i). \tag{12}$$

Denoting $\max(0, \|\boldsymbol{z}^{(k)} - \boldsymbol{x}_i\|_2^{-1} - \frac{1}{\gamma})$ as $w_i$, and then $\alpha = \frac{1}{\sum_i w_i}$ yields a similar reweighting iteration $\boldsymbol{z}^{(k+1)} = \frac{\sum_i w_i^{(k)} \boldsymbol{x}_i}{\sum_i w_i^{(k)}}$.

$\ell_2$ **estimator.** It is worth noting that the same technique can be applied to the $\ell_2$ estimator with $\rho(\boldsymbol{z}) := \|\boldsymbol{z}\|_2^2$. The iteration becomes

$$\boldsymbol{z}^{(k+1)} = \boldsymbol{z}^{(k)} - \alpha\nabla_{\boldsymbol{z}} \sum_i \|\boldsymbol{z} - \boldsymbol{x}_i\|_2^2 \tag{13}$$

$$= \boldsymbol{z}^{(k)} - \alpha \sum_i (\boldsymbol{z}^{(k)} - \boldsymbol{x}_i), \tag{14}$$

and $\alpha = \frac{1}{n+m}$ yields $\boldsymbol{z}^{(k+1)} = \frac{1}{n+m} \sum_i \boldsymbol{x}_i$, which gives the mean of all samples in one single iteration.

Similarities between this mean estimation scenario and our QN-IRLS in graph smoothing can be observed, both of which involve iterative reweighting to estimate similar objectives. The approximated Hessian in our QN-IRLS resembles the Weiszfeld method, canceling the $\boldsymbol{z}^{(k)}$ by tuning the stepsize.

## A.2  ADDITIONAL SIMULATION RESULTS AND DISCUSSIONS

Here, we complement Fig. Figure 2 with the settings of higher attack budgets. As the outlier ratio exceeds the breakdown point $50\%$, we observe that our MCP-based mean estimator can correctly recover the majority of the samples, i.e. converge to the center of "outliers".

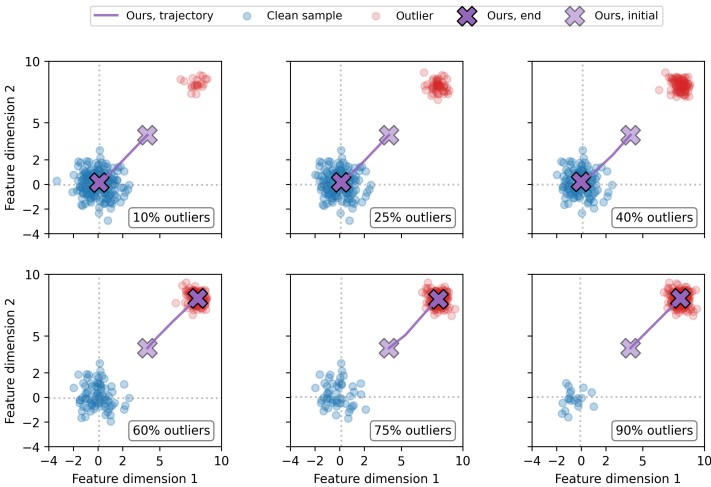

Figure 9: The trajectory of our MCP-based mean estimator.

## B CONVERGENCE ANALYSIS

To begin with, we will provide an overview of our proof, followed by a detailed presentation of the formal proof for the convergence analysis.

**Overview of proof.** First, for both IRLS and QN-IRLS, we construct, for $\boldsymbol{F}^{(k)}$ in every iteration $k$, a quadratic upper bound $\hat{\mathcal{H}}$ that satisfies $\hat{\mathcal{H}} + C \geq \mathcal{H}$ where the equality is reached at $\boldsymbol{F}^{(k)}$. Then we can minimize $\hat{\mathcal{H}}$ to guarantee the iterative descent of $\mathcal{H}$ since $\mathcal{H}(\boldsymbol{F}^{(k+1)}) \leq \hat{\mathcal{H}}(\boldsymbol{F}^{(k+1)}) + C \leq \hat{\mathcal{H}}(\boldsymbol{F}^{(k)}) + C = \mathcal{H}(\boldsymbol{F}^{(k)})$.

To find the $\boldsymbol{F}^{(k+1)}$ such that $\hat{\mathcal{H}}(\boldsymbol{F}^{(k+1)}) \leq \hat{\mathcal{H}}(\boldsymbol{F}^{(k)})$, IRLS simply adopts the plain gradient descent $\boldsymbol{F}^{(k+1)} = \boldsymbol{F}^{(k)} - \eta\nabla\hat{\mathcal{H}}(\boldsymbol{F}^{(k)})$ whose convergence condition can be analyzed with the $\beta$-smoothness of the quadratic $\hat{\mathcal{H}}$ (Theorem 1). To address the problems of IRLS as motivated in Section 3.2, our Quasi-Newton IRLS utilizes the diagonal approximate Hessian $\hat{\boldsymbol{Q}}$ to scale the update step size in different dimensions respectively as $\boldsymbol{F}^{(k+1)} = \boldsymbol{F}^{(k)} - \hat{\boldsymbol{Q}}^{-1}\nabla\hat{\mathcal{H}}(\boldsymbol{F}^{(k)})$. Thereafter, by bounding the Hessian with $2\hat{\boldsymbol{Q}}$, the descent condition of $\hat{\mathcal{H}}$ is simplified (Theorem 2).

**Lemma 1.** *For any $\rho(y)$ satisfying $\frac{d\rho(y)}{dy^2}$ is non-increasing, denote $y_{ij} := \left\|\frac{\boldsymbol{f}_i}{\sqrt{d_i}} - \frac{\boldsymbol{f}_j}{\sqrt{d_j}}\right\|_2$, then*

$\mathcal{H}(\boldsymbol{F}) = \sum_{(i,j)\in\mathcal{E},i\neq j}\rho(y_{ij}) + \lambda\sum_{i\in\mathcal{V}}\|\boldsymbol{f}_i - \boldsymbol{f}_i^{(0)}\|_2^2$ *has the following upper bound:*

$$\mathcal{H}(\boldsymbol{F}) \leq \hat{\mathcal{H}}(\boldsymbol{F}) + C = \sum_{(i,j)\in\mathcal{E},i\neq j}\boldsymbol{W}_{ij}^{(k)}y_{ij}^2 + \lambda\sum_{i\in\mathcal{V}}\|\boldsymbol{f}_i - \boldsymbol{f}_i^{(0)}\|_2^2 + C, \tag{15}$$

*where $\boldsymbol{W}_{ij}^{(k)} = \frac{\partial\rho(y)}{\partial y^2}\big|_{y=y_{ij}^{(k)}}$ and $y_{ij}^{(k)} = \left\|\frac{\boldsymbol{f}_i^{(k)}}{\sqrt{d_i}} - \frac{\boldsymbol{f}_j^{(k)}}{\sqrt{d_j}}\right\|_2$ and $C = \mathcal{H}(\boldsymbol{F}^{(k)}) - \hat{\mathcal{H}}(\boldsymbol{F}^{(k)})$ is a constant. The equality in Eq. (15) is achieved when $\boldsymbol{F} = \boldsymbol{F}^{(k)}$.*

*Proof.* Let $v = y^2$ and define $\psi(v) := \rho(y) = \rho(\sqrt{v})$. Then $\psi$ is concave since $\frac{d\psi(v)}{dv} = \frac{d\rho(y)}{dy^2}$ is non-increasing. According to the concavity property, we have $\psi(v) \leq \psi(v_0) + \psi'(\nu)\big|_{\nu=v_0}(v - v_0)$. Substitute $v = y^2, v_0 = y_0^2$, we obtain:

$$\rho(y) \leq y^2\frac{\partial\rho(y)}{\partial y^2}\big|_{y=y_0} - y_0^2\frac{\partial\rho(y)}{\partial y^2}\big|_{y=y_0} + \rho(y_0) = y^2\frac{\partial\rho(y)}{\partial y^2}\big|_{y=y_0} + C(y_0) \tag{16}$$

where the inequality is reached when $y = y_0$. Next, substitute $y = y_{ij}$ and $y_0 = y_{ij}^{(k)}$, we can get $\rho(y_{ij}) \leq \boldsymbol{W}_{ij}^{(k)}y_{ij}^2 + C(y_{ij}^{(k)})$ which takes the equality at $y_{ij} = y_{ij}^{(k)}$. Finally, by summing up both sides and add a regularization term, we can prove the Eq. (15). □

**Remark 1.** *It can be seen that the definition of $\hat{\mathcal{H}}$ depends on $\boldsymbol{F}^{(k)}$, which ensures that the bound is tight when $\boldsymbol{F} = \boldsymbol{F}^{(k)}$. This tight bound condition is essential in the majorization-minimization algorithm as seen in Theorem 1.*

**Lemma 2.** *For $\hat{\mathcal{H}} = \sum_{(i,j)\in\mathcal{E}, i\neq j} \boldsymbol{W}_{ij} y_{ij}^2 + \lambda \sum_{i\in\mathcal{V}} \|\boldsymbol{f}_i - \boldsymbol{f}_i^{(0)}\|_2^2$, the gradient and Hessian w.r.t. $\boldsymbol{F}$[4] satisfy*

$$\nabla_{\boldsymbol{F}_{mn}} \hat{\mathcal{H}}(\boldsymbol{F}) = 2\left( (diag(\boldsymbol{q}) - \boldsymbol{W} \odot \tilde{\boldsymbol{A}} + \lambda \boldsymbol{I})\boldsymbol{F} - \lambda \boldsymbol{F}^{(0)} \right)_{mn}, \tag{17}$$

*and*

$$\nabla_{\boldsymbol{F}_{mn}} \nabla_{\boldsymbol{F}_{kl}} \hat{\mathcal{H}}(\boldsymbol{F}) = 2\left( diag(\boldsymbol{q}) - \boldsymbol{W} \odot \tilde{\boldsymbol{A}} + \lambda \boldsymbol{I} \right)_{mk}, \tag{18}$$

*where $\boldsymbol{q}_m = \sum_j \boldsymbol{W}_{mj} \boldsymbol{A}_{mj}/d_m$ and $\tilde{\boldsymbol{A}}_{ij} = \frac{\boldsymbol{A}_{ij}}{\sqrt{d_i d_j}}$ is the symmetrically normalized adjacency matrix.*

*Proof.* Follow $\boldsymbol{A} = \boldsymbol{A}^\top$ and define $y_{ij}^2 := \left\| \frac{\boldsymbol{f}_i}{\sqrt{d_i}} - \frac{\boldsymbol{f}_j}{\sqrt{d_j}} \right\|_2^2$, then the first-order gradient of $\sum_{(i,j)\in\mathcal{E}, i\neq j} \boldsymbol{W}_{ij} y_{ij}^2$ will be

$$\nabla_{\boldsymbol{F}_{mn}} \left( \sum_{(i,j)\in\mathcal{E}, i\neq j} \boldsymbol{W}_{ij} y_{ij}^2 \right) \tag{19}$$

$$= \sum_{(i,j)\in\mathcal{E}, i\neq j} \boldsymbol{W}_{ij} \frac{\partial y_{ij}^2}{\partial \boldsymbol{F}_{mn}} \tag{20}$$

$$= \sum_{(m,j)\in\mathcal{E}} \boldsymbol{W}_{mj} \frac{\partial y_{mj}^2}{\partial \boldsymbol{F}_{mn}} \tag{21}$$

$$= \sum_{(m,j)\in\mathcal{E}} \boldsymbol{W}_{mj} \frac{\partial \left( \frac{\boldsymbol{F}_{mn}}{\sqrt{d_m}} - \frac{\boldsymbol{F}_{jn}}{\sqrt{d_j}} \right)^2}{\partial \boldsymbol{F}_{mn}} \tag{22}$$

$$= \sum_{j\in\mathcal{N}(m)} 2\boldsymbol{W}_{mj} \left( \frac{\boldsymbol{F}_{mn}}{d_m} - \frac{\boldsymbol{F}_{jn}}{\sqrt{d_m d_j}} \right) \tag{23}$$

$$= 2 \sum_j \boldsymbol{W}_{mj} \left( \frac{\boldsymbol{A}_{mj}}{d_m} \boldsymbol{F}_{mn} - \frac{\boldsymbol{A}_{mj}}{\sqrt{d_m d_j}} \boldsymbol{F}_{jn} \right) \tag{24}$$

$$= 2 \left( \frac{\sum_j \boldsymbol{W}_{mj} \boldsymbol{A}_{mj}}{d_m} \right) \boldsymbol{F}_{mn} - 2((\boldsymbol{W} \odot \tilde{\boldsymbol{A}})\boldsymbol{F})_{mn} \tag{25}$$

$$= \left( 2(\mathrm{diag}(\boldsymbol{q}) - \boldsymbol{W} \odot \tilde{\boldsymbol{A}})\boldsymbol{F} \right)_{mn}, \tag{26}$$

---

[4]Here are some explanations on the tensor 'Hessian' $\nabla^2 \hat{\mathcal{H}}$. Since $\hat{\mathcal{H}}(\boldsymbol{F})$ is dependent on a matrix, there are some difficulties in defining the Hessian. However, as can be observed in Eq. (26) and Eq. (31), the feature dimension can be accounted for by the following. Initially, we treat the feature dimension as an irrelevant dimension that is excluded from the matrix operations. E.g., $\boldsymbol{F} \nabla^2 \hat{\mathcal{H}} \boldsymbol{F} = \sum_{ik} \boldsymbol{F}_{ij} \nabla^2 \hat{\mathcal{H}}_{ijkl} \boldsymbol{F}_{kl}$ where the feature dimensions $j$ and $l$ remain free indices while the node indices $i$ and $k$ are eliminated as dummy indices. Finally, we take the trace of the resulting #feature×#feature matrix to get the desired value.

and the second-order hessian will be:

$$\nabla^2_{\boldsymbol{F}_{mn}\boldsymbol{F}_{kl}} \left( \sum_{(i,j)\in\mathcal{E}, i\neq j} \boldsymbol{W}_{ij} y_{ij}^2 \right) \tag{27}$$

$$= \sum_{(i,j)\in\mathcal{E}, i\neq j} \boldsymbol{W}_{ij} \frac{\partial y_{ij}^2}{\partial \boldsymbol{F}_{mn} \partial \boldsymbol{F}_{kl}} \tag{28}$$

$$= 2\frac{\partial}{\partial \boldsymbol{F}_{kl}} \left( \sum_j \boldsymbol{W}_{mj} \left( \frac{\boldsymbol{A}_{mj}}{d_m} \boldsymbol{F}_{mn} - \frac{\boldsymbol{A}_{mj}}{\sqrt{d_m d_j}} \boldsymbol{F}_{jn} \right) \right) \tag{29}$$

$$= 2\left( \boldsymbol{q}_m \delta_{mk} - \sum_j \boldsymbol{W}_{mj} \frac{\boldsymbol{A}_{mj}}{\sqrt{d_m d_j}} \delta_{jk} \right) \delta_{nl} \tag{30}$$

$$= 2(\mathrm{diag}(\boldsymbol{q}) - \boldsymbol{W} \odot \tilde{\boldsymbol{A}})_{mk} \delta_{nl}. \tag{31}$$

$\square$

**Remark 2.** *From Eq.* (20) *to Eq.* (23)*, one can assume* $m \notin \mathcal{N}(m)$*, and thus* $\boldsymbol{W}_{mm} = 0$*. However, as we know, a self-loop is often added to* $\boldsymbol{A}$ *to facilitate stability by avoiding zero-degree nodes that cannot be normalized. This is not as problematic as it seems, though. Because* $\sum_{(i,j)\in\mathcal{E}, i\neq j}$ *intrinsically excludes the diagonal terms, we can simply assign zero to the diagonal terms of* $\boldsymbol{W}$ *so that the term of* $j = m$ *is still zero in Eq.* (23)*, as defined in Eq.* (5)*.*

To minimize $\hat{\mathcal{H}}$, the gradient descent update rule takes the form of Eq. (6). One may assume that when $\eta$ is chosen to be small enough, $\hat{\mathcal{H}}(\boldsymbol{F}^{(k+1)}) \leq \hat{\mathcal{H}}(\boldsymbol{F}^{(k)})$. For a formal justification, we have Theorem 1 to determine the convergence condition of $\eta$.

**Theorem 1.** *If* $\boldsymbol{F}^{(k)}$ *follows the update rule in Eq.* (6)*, where the* $\rho$ *satisfies that* $\frac{d\rho(y)}{dy^2}$ *is non-decreasing for* $y \in (0, \infty)$*, then a sufficient condition for* $\mathcal{H}(\boldsymbol{F}^{(k+1)}) \leq \mathcal{H}(\boldsymbol{F}^{(k)})$ *is that the step size* $\eta$ *satisfies* $0 < \eta \leq \|diag(\boldsymbol{q}^{(k)}) - \boldsymbol{W}^{(k)} \odot \tilde{\boldsymbol{A}} + \lambda \boldsymbol{I}\|_2^{-1}$*.*

*Proof.* The descent of $\hat{\mathcal{H}}(\boldsymbol{F})$ can ensure the descent of $\mathcal{H}(\boldsymbol{F})$ since $\mathcal{H}(\boldsymbol{F}^{(k+1)}) \leq \hat{\mathcal{H}}(\boldsymbol{F}^{(k+1)}) \leq \hat{\mathcal{H}}(\boldsymbol{F}^{(k)}) = \mathcal{H}(\boldsymbol{F}^{(k)})$. Therefore, we only need to prove $\hat{\mathcal{H}}(\boldsymbol{F}^{(k+1)}) \leq \hat{\mathcal{H}}(\boldsymbol{F}^{(k)})$.

Noting that $\hat{\mathcal{H}}$ is a quadratic function and $\boldsymbol{F}^{(k+1)} - \boldsymbol{F}^{(k)} = -\eta\nabla\hat{\mathcal{H}}(\boldsymbol{F}^{(k)})$, then $\hat{\mathcal{H}}(\boldsymbol{F}^{(k+1)})$ and $\hat{\mathcal{H}}(\boldsymbol{F}^{(k)})$ can be connected using Taylor expansion[4], where $\nabla\hat{\mathcal{H}}$ and $\nabla^2\hat{\mathcal{H}}$ is given in Lemma 2:

$$\hat{\mathcal{H}}(\boldsymbol{F}^{(k+1)}) - \hat{\mathcal{H}}(\boldsymbol{F}^{(k)}) \tag{32}$$

$$= \mathrm{tr}\left( \nabla\hat{\mathcal{H}}(\boldsymbol{F}^{(k)})^\top (\boldsymbol{F}^{(k+1)} - \boldsymbol{F}^{(k)}) \right) + \frac{1}{2}\mathrm{tr}\left( (\boldsymbol{F}^{(k+1)} - \boldsymbol{F}^{(k)})^\top \nabla^2\hat{\mathcal{H}}(\boldsymbol{F}^{(k)})(\boldsymbol{F}^{(k+1)} - \boldsymbol{F}^{(k)}) \right) \tag{33}$$

$$= \mathrm{tr}\left( -\eta\nabla\hat{\mathcal{H}}(\boldsymbol{F}^{(k)})^\top \nabla\hat{\mathcal{H}}(\boldsymbol{F}^{(k)}) + \frac{\eta^2}{2}\nabla\hat{\mathcal{H}}(\boldsymbol{F}^{(k)})^\top \nabla^2\hat{\mathcal{H}}(\boldsymbol{F}^{(k)})\nabla\hat{\mathcal{H}}(\boldsymbol{F}^{(k)}) \right). \tag{34}$$

Insert $\nabla^2\hat{\mathcal{H}}(\boldsymbol{F}^{(k)}) = 2(\mathrm{diag}(\boldsymbol{q}) - \boldsymbol{W}^{(k)} \odot \tilde{\boldsymbol{A}} + \lambda\boldsymbol{I})$ from Lemma 2 into the above equation and we can find a sufficient condition for $\hat{\mathcal{H}}(\boldsymbol{F}^{(k+1)}) \leq \hat{\mathcal{H}}(\boldsymbol{F}^{(k)})$ to be

$$-\eta + \|\mathrm{diag}(\boldsymbol{q}) - \boldsymbol{W}^{(k)} \odot \tilde{\boldsymbol{A}} + \hat{\lambda}\boldsymbol{I}\|_2 \eta^2 \leq 0, \tag{35}$$

or

$$\eta \leq \|\mathrm{diag}(\boldsymbol{q}) - \boldsymbol{W}^{(k)} \odot \tilde{\boldsymbol{A}} + \hat{\lambda}\boldsymbol{I}\|_2^{-1}. \tag{36}$$

$\square$

Now we prove that when taking the Quasi-Newton-IRLS step as in Eq. (7), the objective $\hat{\mathcal{H}}$ is guaranteed to descend. Since the features in different dimensions are irrelevant, we simplify our notations as if feature dimension was 1. One may easily recover the general scenario by taking the trace.

**Lemma 3.** $2\hat{Q}-\nabla^2\hat{\mathcal{H}}(y)$ is positive semi-definite, where $\hat{\mathcal{H}} = \sum_{(i,j)\in\mathcal{E},i\neq j} \boldsymbol{W}_{ij}y_{ij}^2 + \lambda\sum_{i\in\mathcal{V}}\|\boldsymbol{f}_i - \boldsymbol{f}_i^{(0)}\|_2^2$, $\hat{\boldsymbol{Q}} = 2(diag(\boldsymbol{q}) + \lambda\boldsymbol{I})$, and $\boldsymbol{q}_m = \sum_j \boldsymbol{W}_{mj}\boldsymbol{A}_{mj}/d_m$.

*Proof.* In Lemma 2, we have $\nabla^2\hat{\mathcal{H}}(\boldsymbol{y}) = 2(\text{diag}(\boldsymbol{q}) + \lambda\boldsymbol{I} - \boldsymbol{W}\odot\tilde{\boldsymbol{A}})$, then

$$2\hat{\boldsymbol{Q}} - \nabla^2\hat{\mathcal{H}}(\boldsymbol{y}) = 2(\text{diag}(\boldsymbol{q}) + \lambda\boldsymbol{I} + \boldsymbol{W}\odot\tilde{\boldsymbol{A}}). \tag{37}$$

Recall how we derived Eq. (26) from Eq. (15), where we proved that

$$\sum_{(i,j)\in\mathcal{E},i\neq j} \boldsymbol{W}_{ij}\|\frac{\boldsymbol{f}_i}{\sqrt{d_i}} - \frac{\boldsymbol{f}_j}{\sqrt{d_j}}\|_2^2 = \text{tr}(\boldsymbol{F}^\top(\text{diag}(\boldsymbol{q}) - \boldsymbol{W}\odot\tilde{\boldsymbol{A}})\boldsymbol{F}), \tag{38}$$

which holds for all $\boldsymbol{F}$. Similarly, the equation still holds after flipping the sign before $\boldsymbol{f}_j/\sqrt{d_j}$ and $\boldsymbol{W}\odot\tilde{\boldsymbol{A}}$. We then have this inequality: $\forall\boldsymbol{F},\forall\lambda\geq 0$

$$0\leq \sum_{(i,j)\in\mathcal{E},i\neq j} \boldsymbol{W}_{ij}\|\frac{\boldsymbol{f}_i}{\sqrt{d_i}} + \frac{\boldsymbol{f}_j}{\sqrt{d_j}}\|_2^2 = \text{tr}(\boldsymbol{F}^\top(\text{diag}(\boldsymbol{q}) + \boldsymbol{W}\odot\tilde{\boldsymbol{A}})\boldsymbol{F}) \tag{39}$$

$$\leq\text{tr}(\boldsymbol{F}^\top(\text{diag}(\boldsymbol{q}) + \boldsymbol{W}\odot\tilde{\boldsymbol{A}} + \lambda\boldsymbol{I})\boldsymbol{F}). \tag{40}$$

Thus, $(\text{diag}(\boldsymbol{q}) + \boldsymbol{W}\odot\tilde{\boldsymbol{A}} + \lambda\boldsymbol{I})\succeq 0$, and thus $2\hat{\boldsymbol{Q}} - \nabla^2\hat{\mathcal{H}}(\boldsymbol{y})\succeq 0$.

$\square$

Using Lemma 1 and Lemma 3 we can prove Theorem 2. Note that we continue to assume #feature= 1 for simplicity but without loss of generality[4].

**Theorem 2.** If $\boldsymbol{F}^{(k+1)}$ follows update rule in Eq. (7), where $\rho$ satisfies that $\frac{d\rho(y)}{dy^2}$ is non-decreasing $\forall y\in(0,\infty)$, it is guaranteed that $\mathcal{H}(\boldsymbol{F}^{(k+1)})\leq\mathcal{H}(\boldsymbol{F}^{(k)})$.

*Proof.* Following the discussions in Theorem 1, we only need to prove $\hat{\mathcal{H}}(\boldsymbol{F}^{(k+1)})\leq\hat{\mathcal{H}}(\boldsymbol{F}^{(k)})$. For the quadratic $\hat{\mathcal{H}}$, we have:

$$\hat{\mathcal{H}}(\boldsymbol{x}) = \hat{\mathcal{H}}(\boldsymbol{y}) + \nabla\hat{\mathcal{H}}(\boldsymbol{y})^\top(\boldsymbol{x} - \boldsymbol{y}) + \frac{1}{2}(\boldsymbol{x} - \boldsymbol{y})^\top\nabla^2\hat{\mathcal{H}}(\boldsymbol{y})(\boldsymbol{x} - \boldsymbol{y}). \tag{41}$$

We can define $\mathcal{Q}(\boldsymbol{y}) = 2\hat{\boldsymbol{Q}}(\boldsymbol{y})$ in Lemma 3 such that $\mathcal{Q}(\boldsymbol{y}) - \nabla^2\hat{\mathcal{H}}(\boldsymbol{y})\succeq 0$, then

$$\forall\boldsymbol{z}, \boldsymbol{z}^\top\mathcal{Q}(\boldsymbol{y})\boldsymbol{z}\geq \boldsymbol{z}^\top\nabla^2\hat{\mathcal{H}}(\boldsymbol{y})\boldsymbol{z}. \tag{42}$$

Then an upper bound of $\hat{\mathcal{H}}(\boldsymbol{x})$ can be found by inserting Eq. (42) into Eq. (41).

$$\hat{\mathcal{H}}(\boldsymbol{x})\leq\hat{\mathcal{H}}(\boldsymbol{y}) + \nabla\hat{\mathcal{H}}(\boldsymbol{y})^\top(\boldsymbol{x} - \boldsymbol{y}) + \frac{1}{2}(\boldsymbol{x} - \boldsymbol{y})^\top\mathcal{Q}(\boldsymbol{y})(\boldsymbol{x} - \boldsymbol{y}). \tag{43}$$

Then, insert $\mathcal{Q} = 2\hat{\boldsymbol{Q}}$ into Eq. (43). Note that $\hat{\boldsymbol{Q}} := 2(\text{diag}(\boldsymbol{q}) + \hat{\lambda}\boldsymbol{I})$, so $\hat{\boldsymbol{Q}}\succeq 0$ and $\hat{\boldsymbol{Q}}^\top = \hat{\boldsymbol{Q}}$. Thereafter, the update rule $\boldsymbol{x} = \boldsymbol{y} - \hat{\boldsymbol{Q}}^{-1}\nabla\hat{\mathcal{H}}(\boldsymbol{y})$ in Eq. (7) gives

$$\hat{\mathcal{H}}(\boldsymbol{x}) - \hat{\mathcal{H}}(\boldsymbol{y})\leq\nabla\hat{\mathcal{H}}(\boldsymbol{y})^\top(\boldsymbol{x} - \boldsymbol{y}) + \frac{1}{2}(\boldsymbol{x} - \boldsymbol{y})^\top\mathcal{Q}(\boldsymbol{y})(\boldsymbol{x} - \boldsymbol{y}) \tag{44}$$

$$=\nabla\hat{\mathcal{H}}(\boldsymbol{y})^\top(\boldsymbol{x} - \boldsymbol{y}) + 2\left(\hat{\boldsymbol{Q}}^{\frac{1}{2}}(\boldsymbol{x} - \boldsymbol{y})\right)^\top\left(\hat{\boldsymbol{Q}}^{\frac{1}{2}}(\boldsymbol{x} - \boldsymbol{y})\right) \tag{45}$$

$$=2\nabla\hat{\mathcal{H}}(\boldsymbol{y})^\top\hat{\boldsymbol{Q}}^{-1}\nabla\hat{\mathcal{H}}(\boldsymbol{y}) - 2\nabla\hat{\mathcal{H}}(\boldsymbol{y})^\top(\hat{\boldsymbol{Q}}^{-\frac{1}{2}})^\top\hat{\boldsymbol{Q}}^{-\frac{1}{2}}\nabla\hat{\mathcal{H}}(\boldsymbol{y}) \tag{46}$$

$$=0. \tag{47}$$

Therefore, our QN-IRLS in Eq. (7) is guaranteed to descend. $\square$

# C  COMPUTATION EFFICIENCY

Our RUNG model preserves advantageous efficiency even adopting the quasi-Newton IRLS algorithm.

## C.1  TIME COMPLEXITY ANALYSIS

Each RUNG layer involves computing $W$, $q$, and the subsequent aggregations. We elaborate on them one by one. We denote the number of feature dimensions $d$, the number of nodes $n$, and the number of edges $m$, which are assumed to satisfy $m \gg 1$, $n \gg 1$ and $d \gg 1$. The number of layers is denoted as $k$. The asymptotic computation complexity is denoted as $\mathcal{O}(\cdot)$.

**Computation of $W \odot A$ and $W \odot \tilde{A}$.**  $W := \mathbf{1}_{i \neq j} \frac{d\rho_\gamma(y_{ij})}{dy_{ij}^2}$ is the edge weighting matrix dependent on the node feature matrix $F$. The computation of $y_{ij} = \|\frac{f_i}{\sqrt{d_i}} - \frac{f_j}{\sqrt{d_j}}\|_2$ is $\mathcal{O}(d)$ and that of $\frac{d\rho_\gamma(y_{ij})}{dy_{ij}^2}$ is $\mathcal{O}(1)$. $W_{ij}$ only needs computing when $(i, j) \in \mathcal{E}$, because $\forall (i, j) \notin \mathcal{E}$, $W_{ij}$ will be masked out by $A$ or $\tilde{A}$ anyways. Each element of $W$ involves computation time of $\mathcal{O}(d)$ and $m$ elements are needed. In total, $W$ costs $\mathcal{O}(md)$, and $W \odot A$ and $W \odot \tilde{A}$ cost $\mathcal{O}(md + m) = \mathcal{O}(md)$.

**Computation of $\hat{Q}^{-1}$.**  $\hat{Q}^{-1} := \frac{1}{2}(\text{diag}(q^{(k)}) + \lambda I)^{-1}$ is the inverse Hessian in our quasi-Newton IRLS. Because $\hat{Q}$ is designed to be a diagonal matrix, its inverse can be evaluated as element-wise reciprocal which is efficient. As for $q := \sum_j W_{mj}^{(k)} A_{mj}/d_m$, only existing edges $(i, j) \in \mathcal{E}$ need evaluation in the summation. Therefore, this computation costs $\mathcal{O}(m)$. Thus, $\hat{Q}^{-1}$ costs $\mathcal{O}(m)$ in total.

**Computation of aggregation.**  An RUNG layer follows

$$F^{(k+1)} = 2\hat{Q}^{-1}\left((W^{(k)} \odot \tilde{A})F^{(k)} + \lambda F^{(0)}\right), \tag{48}$$

which combines the quantities calculated above. An extra graph aggregation realized by the matrix multiplication between $W \odot \tilde{A}$ and $F$ is required, costing $\mathcal{O}(md)$. The subsequent addition to $F^{(0)}$ and the multiplication to the diagonal $\hat{Q}^{-1}$ both cost $\mathcal{O}(nd)$.

**Stacking layers.**  RUNG unrolls the QN-IRLS optimization procedure, which has multiple iterations. Therefore, the convergence increase that QN-IRLS introduces allows a RUNG with fewer layers and increases the overall complexity. It is worth noting that the QN-IRLS utilizes a diagonal approximated Hessian, and thus the computation per iteration is also kept efficient as discussed above.

Summing up all the costs, we have the total computational complexity of our RUNG, $\mathcal{O}((m + n)kd)$. Our RUNG thus scales well to larger graph datasets such as ogbn-arxiv.

**Space Complexity Analysis**  The only notable extra storage cost is $W$ whose sparse layout takes up $\mathcal{O}(m)$. This is the same order of size as the adjacency matrix itself, thus not impacting the total asymptotic complexity.

## C.2  ALTERNATIVE PERSPECTIVE

In fact, the above analysis can be simplified when we look at the local aggregation behavior of RUNG. For node $i$, it's updated via aggregation $f_i = \frac{2}{\hat{Q}_{ii}^{-1}}((\sum_{j \in \mathcal{N}(i)} W_{ij}f_j) + \lambda f_i^{(0)})$. The summation over neighbors' $f_j$ will give $\mathcal{O}(m)$ in the total time complexity in each feature dimension, and $W_{ij}$ involves $\mathcal{O}(d)$ computations for each neighbor. This sums up to $\mathcal{O}(md)$ as well. Essentially, the high efficiency of RUNG originates from that every edge weighting in our model involves only the 2 nodes on this edge.

# D ADDITIONAL EXPERIMENT RESULTS

In this section, we present the experiment results that are not shown in the main paper due to space limits. Table 3 and Table 4 are the results of adaptive local and global attacks on Citeseer, referred to in Section 4.2. Figure 10 is the experiment result of our RUNG attacked by transfer attacks generated on different surrogate models as mentioned in Section 4.3.

Table 3: Adaptive local attack on Citeseer. The **best** and second are marked.

| Model | 0% | 20% | 50% | 100% | 150% | 200% |
|---|---|---|---|---|---|---|
| MLP | $69.3 \pm 2.5$ | $69.3 \pm 2.5$ | $69.3 \pm 2.5$ | **$69.3 \pm 2.5$** | **$69.3 \pm 2.5$** | **$69.3 \pm 2.5$** |
| GCN | $79.3 \pm 3.3$ | $44.7 \pm 8.8$ | $27.3 \pm 7.7$ | $6.7 \pm 3.7$ | $0.7 \pm 1.3$ | $0.0 \pm 0.0$ |
| APPNP | **$80.7 \pm 4.4$** | $50.0 \pm 6.7$ | $39.3 \pm 6.5$ | $16.7 \pm 12.3$ | $16.0 \pm 8.0$ | $0.0 \pm 0.0$ |
| GAT | $74.7 \pm 5.0$ | $15.3 \pm 17.5$ | $13.3 \pm 13.5$ | $12.0 \pm 11.3$ | $12.7 \pm 6.5$ | $9.3 \pm 5.3$ |
| GNNGuard | $74.7 \pm 4.5$ | $46.0 \pm 10.4$ | $32.7 \pm 11.0$ | $18.0 \pm 7.5$ | $6.0 \pm 3.9$ | $4.0 \pm 3.9$ |
| RGCN | $80.0 \pm 2.1$ | $46.7 \pm 9.4$ | $32.7 \pm 8.8$ | $10.0 \pm 5.2$ | $0.7 \pm 1.3$ | $0.7 \pm 1.3$ |
| GRAND | $77.3 \pm 2.5$ | $56.7 \pm 4.2$ | $44.0 \pm 3.9$ | $16.7 \pm 6.3$ | $0.7 \pm 1.3$ | $0.0 \pm 0.0$ |
| ProGNN | $80.0 \pm 2.1$ | $42.7 \pm 7.4$ | $26.0 \pm 5.3$ | $10.0 \pm 4.7$ | $0.7 \pm 1.3$ | $0.0 \pm 0.0$ |
| Jaccard-GCN | $78.7 \pm 3.4$ | $46.7 \pm 7.3$ | $28.0 \pm 7.5$ | $6.7 \pm 4.7$ | $0.7 \pm 1.3$ | $0.0 \pm 0.0$ |
| SoftMedian | $78.7 \pm 3.4$ | $69.3 \pm 6.5$ | $66.0 \pm 7.1$ | $56.0 \pm 4.4$ | $8.7 \pm 6.9$ | $3.3 \pm 3.0$ |
| TWIRLS | $77.3 \pm 2.5$ | $69.3 \pm 1.3$ | $68.7 \pm 1.6$ | $57.3 \pm 2.5$ | $36.7 \pm 4.7$ | $26.7 \pm 4.7$ |
| TWIRLS-T | $76.0 \pm 3.3$ | $70.7 \pm 2.5$ | $68.7 \pm 2.7$ | $62.0 \pm 3.4$ | $52.7 \pm 5.3$ | $47.3 \pm 8.3$ |
| RUNG-$\ell_1$ (ours) | $80.0 \pm 3.7$ | **$75.3 \pm 4.5$** | **$73.3 \pm 3.0$** | $67.3 \pm 3.3$ | $36.0 \pm 9.3$ | $26.0 \pm 8.3$ |
| RUNG (ours) | $77.3 \pm 1.3$ | $70.7 \pm 5.7$ | $69.3 \pm 6.8$ | $67.3 \pm 7.1$ | $64.0 \pm 5.7$ | $61.3 \pm 5.8$ |

Table 4: Adaptive global attack on Citeseer. The **best** and second are marked.

| Model | Clean | 5% | 10% | 20% | 30% | 40% |
|---|---|---|---|---|---|---|
| MLP | $67.7 \pm 0.3$ | $67.7 \pm 0.3$ | $67.7 \pm 0.3$ | **$67.7 \pm 0.3$** | **$67.7 \pm 0.3$** | **$67.7 \pm 0.3$** |
| GCN | $74.8 \pm 1.2$ | $66.1 \pm 1.0$ | $60.9 \pm 0.8$ | $53.0 \pm 1.0$ | $47.0 \pm 0.8$ | $41.2 \pm 1.1$ |
| APPNP | **$75.3 \pm 1.1$** | $65.8 \pm 0.9$ | $60.7 \pm 1.6$ | $52.3 \pm 1.6$ | $46.0 \pm 2.0$ | $41.2 \pm 2.2$ |
| GAT | $73.4 \pm 1.2$ | $65.4 \pm 1.3$ | $60.4 \pm 1.4$ | $52.6 \pm 2.5$ | $47.2 \pm 3.4$ | $41.2 \pm 4.8$ |
| GNNGuard | $72.4 \pm 1.1$ | $65.6 \pm 0.9$ | $61.8 \pm 1.4$ | $55.6 \pm 1.4$ | $51.0 \pm 1.3$ | $47.3 \pm 1.3$ |
| RGCN | $74.4 \pm 1.0$ | $66.0 \pm 0.8$ | $60.6 \pm 0.9$ | $52.5 \pm 0.8$ | $46.1 \pm 0.9$ | $40.2 \pm 1.0$ |
| GRAND | $74.8 \pm 0.6$ | $66.6 \pm 0.7$ | $61.8 \pm 0.7$ | $53.6 \pm 1.1$ | $47.4 \pm 1.2$ | $42.2 \pm 0.9$ |
| ProGNN | $74.2 \pm 1.3$ | $65.6 \pm 1.1$ | $60.3 \pm 1.1$ | $52.7 \pm 1.4$ | $46.2 \pm 0.9$ | $40.8 \pm 0.6$ |
| Jaccard-GCN | $74.8 \pm 1.2$ | $66.3 \pm 1.2$ | $60.9 \pm 1.2$ | $53.3 \pm 0.9$ | $46.5 \pm 0.9$ | $41.1 \pm 1.0$ |
| SoftMedian | $74.6 \pm 0.7$ | $68.0 \pm 0.7$ | $64.4 \pm 0.9$ | $59.3 \pm 1.1$ | $55.2 \pm 2.0$ | $51.9 \pm 2.1$ |
| TWIRLS | $74.2 \pm 0.8$ | $69.2 \pm 0.8$ | $66.4 \pm 0.7$ | $61.6 \pm 0.9$ | $58.1 \pm 1.2$ | $51.8 \pm 1.5$ |
| TWIRLS-T | $73.7 \pm 1.1$ | $69.1 \pm 1.2$ | $66.4 \pm 1.0$ | $62.8 \pm 1.5$ | $60.0 \pm 1.4$ | $57.4 \pm 1.5$ |
| RUNG-$\ell_1$ (ours) | **$75.5 \pm 1.1$** | $69.3 \pm 1.2$ | $65.9 \pm 1.2$ | $61.1 \pm 1.1$ | $57.2 \pm 1.4$ | $53.9 \pm 1.3$ |
| RUNG (ours) | $74.3 \pm 0.7$ | **$71.4 \pm 1.0$** | **$69.8 \pm 1.3$** | $67.6 \pm 1.2$ | $66.5 \pm 1.3$ | $65.3 \pm 1.5$ |

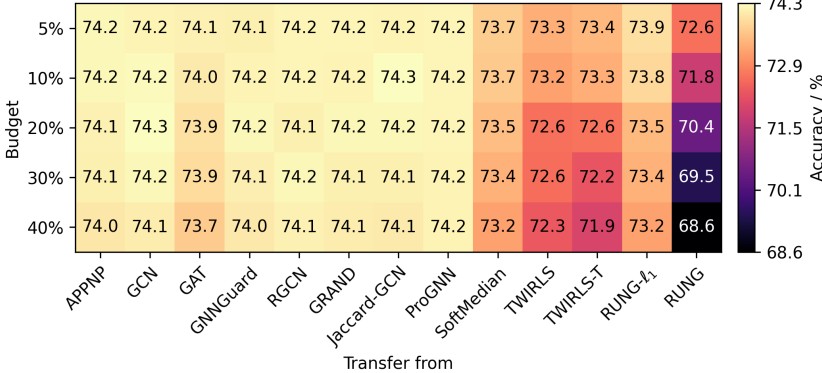

Figure 10: Transfer global attack from different surrogate models to our RUNG on Citeseer.

# E  ADAPTIVE ATTACK SETTINGS

For the adaptive PGD attack we adpoted in the experiments, we majorly followed the algorithm in Mujkanovic et al. (2022) in the adaptive evasion attack. For the sake of completeness, we describe it below:

In summary, we consider the topology attack setting where the adjacency matrix $\boldsymbol{A}$ is perturbed by $\delta\boldsymbol{A}$ whose element $\delta\boldsymbol{A}_{ij} \in \{0, 1\}$. The budget $B$ is defined as $B \geq \|\delta\boldsymbol{A}\|_0$. The PGD attack involves first relaxing $\boldsymbol{A}$ from binary to continuous so that a gradient ascent attack can be conducted on the relaxed graph.

During the attack, the minimization problem below is solved:

$$\delta\boldsymbol{A}_\star = \arg\min_{\delta\boldsymbol{A}} \mathcal{L}_{\text{attack}}(\text{GNN}_\theta(\boldsymbol{A} + (\boldsymbol{I} - 2\boldsymbol{A}) \odot \delta\boldsymbol{A}, \boldsymbol{F}), y_{\text{target}}), \tag{49}$$

where $\mathcal{L}$ is carefully designed attack loss function (Geisler et al., 2021; Mujkanovic et al., 2022), $\boldsymbol{A}$, $\boldsymbol{F}$ and $y_{\text{target}}$ are respectively the graph, node feature matrix and ground truth labels in the dataset, $\theta$ is the parameters of the GNN under attack which are not altered in the evasion attack setting. $(\boldsymbol{I} - 2\boldsymbol{A}) \odot \delta\boldsymbol{A}$ is the calculated perturbation that "flips" the adjacency matrix between 0 and 1 when it is perturbed. The gradient of $\frac{\mathcal{L}_{\text{attack}}}{\delta\boldsymbol{A}}$ is computed and utilized to update the perturbation matrix $\delta\boldsymbol{A}$.

After the optimization problem is solved, $\delta\boldsymbol{A}$ is projected back to the feasible domain of $\delta\boldsymbol{A}_{ij} \in \{1\}$. The adjacency matrix serves as a probability matrix allowing a Bernoulli sampling of the binary adjacency matrix $\boldsymbol{A}'$. The sampling is executed repeatedly so that an $\boldsymbol{A}'$ producing the strongest perturbation is finally generated.

# F  TRANSFER ATTACKS

Figure 11 shows results of global evasion transfer attacks between different models on Cora. Our observations are summarized below:

- The attacks generated by RUNG are stronger when applied to more robust models like SoftMedian, while are not strong against undefended or weakly defended models.

- For $\ell_1$ GNNs, the attacks are the strongest when transferred from $\ell_1$ GNNs. This supports again our unified view on $\ell_1$ GNNs. An exception is TWIRLS because it only has one attention layer and does not always converge to the actual $\ell_1$ objective.

# G  ADDITIONAL ABLATION STUDY OF RUNG

## G.1  HYPERPARAMETERS

The choice of the hyperparameters $\gamma$ and $\lambda$ is crucial to the performance of RUNG. We therefore experimented with their different combinations and conducted adaptive attacks on Cora as shown in Fig. 12.

Recall the formulation of RUNG in Eq.(8):

$$\boldsymbol{F}^{(k+1)} = (\text{diag}(\boldsymbol{q}^{(k)}) + \lambda\boldsymbol{I})^{-1} \left( (\boldsymbol{W}^{(k)} \odot \tilde{\boldsymbol{A}})\boldsymbol{F}^{(k)} + \lambda\boldsymbol{F}^{(0)} \right), \tag{50}$$

where $\boldsymbol{q}_m^{(k)} = \sum_j \boldsymbol{W}_{mj}^{(k)}\boldsymbol{A}_{mj}/d_m$, $\boldsymbol{W}_{ij}^{(k)} = \mathbf{1}_{i\neq j} \max(0, \frac{1}{2y_{ij}^{(k)}} - \frac{1}{2\gamma})$ and $y_{ij}^{(k)} = \big\|\frac{\boldsymbol{f}_i^{(k)}}{\sqrt{d_i}} - \frac{\boldsymbol{f}_j^{(k)}}{\sqrt{d_j}}\big\|_2$.

In the formulation, $\lambda$ controls the intensity of the regularization in the graph smoothing. In our experiments, we tune $\hat{\lambda} := \frac{1}{1+\lambda}$ which is normalized into $(0, 1)$. In Figure 12, the optimal value of $\hat{\lambda}$ can be found almost always near 0.9 regardless of the attack budget. This indicates that our penalty function $\rho_\gamma$ is decoupled from $\gamma$ which makes the tuning easier, contrary to the commonly used formulation of MCP Zhang (2010).

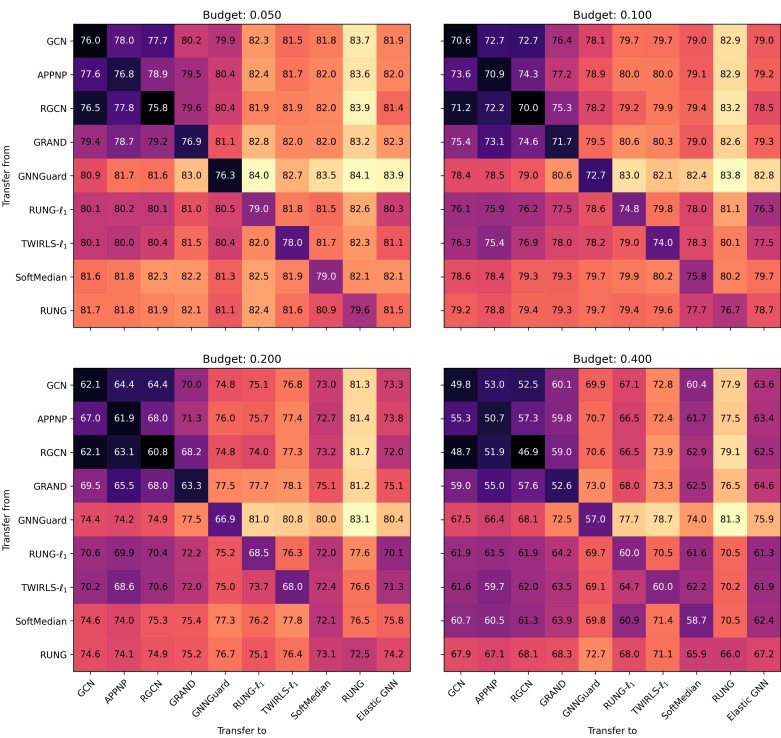

Figure 11: Transfer global attack between different model pairs on Cora.

On the other hand, $\gamma$ has a more intricate impact on the performance of RUNG. Generally speaking, the smaller $\gamma$ is, the more edges get pruned, which leads to higher robustness and a lower clean accuracy. We begin our discussion in three cases:

**Small attack budget** ($0\%, 5\%, 10\%$). The performance is largely dependent on clean accuracy. Besides, when $\gamma \to \infty$, RUNG becomes a state-of-the-art robust $\ell_1$ model. Therefore, a small $\gamma$ likely introduces more harm to the clean performance than robustness increments over $\ell_1$ models. The optimal $\gamma$ thus at least recovers the performance of $\ell_1$ models.

**Large attack budget** ($20\%, 30\%, 40\%$). In these cases, $\gamma \to \infty$ is no longer a good choice because $\ell_1$ models are beginning to suffer from the accumulated bias effect. The optimal $\gamma$ is thus smaller (near $0.5$). However, for fairness, we chose the same $\gamma$ under different budgets in our experiments, so the reported RUNG fixes $\gamma = 3$. In reality, however, when we know the possible attack budgets in advance, we can tune $\gamma$ for an even better performance.

**Very large attack budget** ($50\%, 60\%$). We did not include these scenarios because almost all GNNs perform poorly in this region. However, we believe it can provide some insights into robust graph learning. Under these budgets, more than half of the edges are perturbed. In the context of robust statistics (e.g. mean estimation), the estimator will definitely break down. However, in our problem of graph estimation, the input node features offer extra information allowing us to exploit the graph information even beyond the breakdown point. In the "peak" near $(0.9, 0.5)$, RUNG achieves $> 70\%$ accuracy which is higher than MLP. This indicates that the edge weighting of RUNG is capable of securely harnessing the graph information even in the existence of strong adversarial attacks. The "ridge" near a $\hat{\lambda} = 0.2$, on the other hand, emerges because of MLP. When the regularization dominates, $\lambda \to \infty$, and $\hat{\lambda} \to 0$. A small $\lambda$ is then connected to a larger emphasis on the input node feature prior. Under large attack budgets, MLP delivers relatively good estimation, so a small $\hat{\lambda}$ is beneficial.

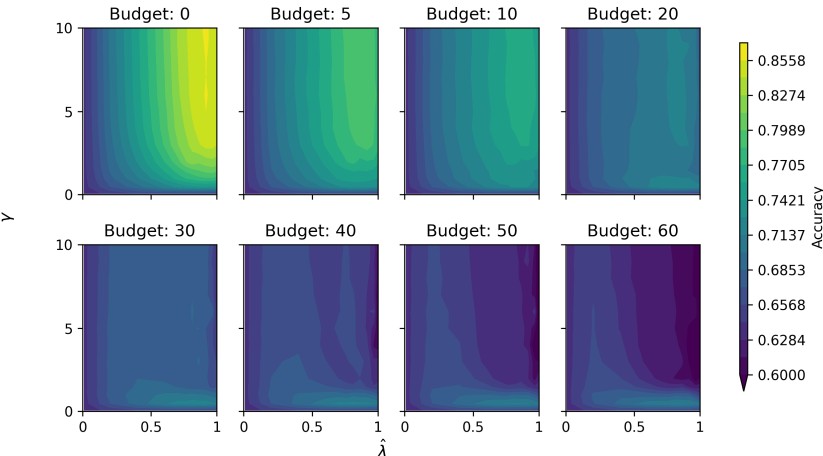

Figure 12: The performance dependence of RUNG with different hyperparameters $\gamma$ and $\lambda$. The performance is evaluated under different attack budgets. The attack setting is the global evasion attack and the dataset is Cora. Note the x-axis is $\hat{\lambda} := \frac{1}{1+\lambda}$ instead of $\lambda$.

## G.2 GNN LAYERS

In RUNG, QN-IRLS is unrolled into GNN layers. We would naturally expect RUNG to have enough number of layers so that the estimator converges as desired. We conducted an ablation study on the performance (clean and adversarial) of RUNG with different layer numbers and the results are shown in Fig. Figure 13. We make the following observations:

- As the layer number increases, RUNG exhibits better performance. This verifies the effectiveness of our proposed RUGE, as well as the stably converging QN-IRLS.
- The performance of RUNG can achieve a reasonably good level even with a small layer number (3-5 layers) with accelerated convergence powered by QN-IRLS. This can further reduce the computation complexity of RUNG.

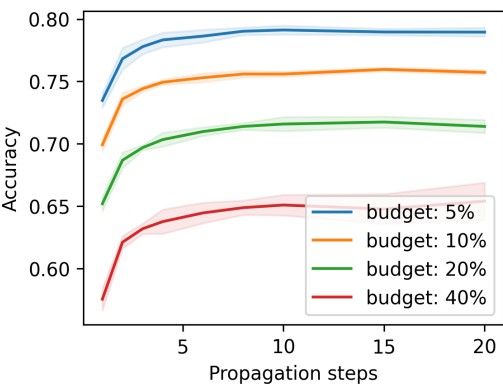

Figure 13: The performance dependence of RUNG on the number of aggregation layers.

