# OpenReview forum: "Robust Graph Neural Networks via Unbiased Aggregation"
_ICLR.cc/2024/Conference — Submitted to ICLR 2024_

### Official Review · Reviewer_YmW7 · 2023-10-19

**Soundness:** 4 excellent
**Presentation:** 2 fair
**Contribution:** 2 fair
**Rating:** 5
**Confidence:** 4

**Summary:**

The paper reviews the estimation bias of several representative GNNs. It proposes a quasi-Newton iterative reweighted LS algorithm to optimize a GNN model, whose loss is based on the minimax concave penalty. This penalty alleviates the bias in typical GNN models and is also more robust.

**Strengths:**

The paper proposes a quasi-Newton iterative reweighted LS algorithm to optimize a GNN model, whose loss is based on the minimax concave penalty. This penalty alleviates the bias in typical GNN models and is also more robust.

**Weaknesses:**

1. The paper is not well-written, with notations that are not always defined. E.g., the superscript (0) first appears in eq. (1) but has never been defined. In addition, the paper uses a strange definition of $\ell_1$ norm to mean either the usual 1-norm or the 2-norm. The $\ell_2$ "norm" in this paper is defined to be the square of the usual 2-norm. This is technically not a norm!

1. The contributions are weak as the formulation is essentially based on TWIRLS with the MCP penalty by Zhang, 2010. Robustness issues in Lasso regression have also been well-studied.

1. Most of the cited works are from 2021 or before. The authors are missing recent GNN robustness works like “On the robustness of graph neural diffusion to topology perturbations", NeurIPS 2022 and “Graph-coupled oscillator networks", ICLR 2022.

1. The experiments are limited with tests only on Cora and CiteSeer. Attacks like TDGIA and MetaGIA are not considered.

**Questions:**

1. A quasi-Newton optimization approach is proposed. Is this related to standard quasi-Newton optimization procedures like BFGS, Broyden, etc.?

1. RUGE is claimed to be unbiased. Is this true and is it proven? An analytical robustness measure is also not provided.

---

> ### Author Response · Authors · 2023-11-22
> **Official Response to Reviewer YmW7 (1/5)**
>
> Dear reviewer, thank you for your valuable comments and your recognition of our contribution in designing QN-IRLS algorithm and RUNG. We are happy to fully address all of your concerns as follows.
>
> We have updated our manuscript, and the changes are highlighted in blue. For newly added appendix sections, the section title is set to blue.
>
>
> ### Q1
>
> > *1. The paper is not well-written, with notations that are not always defined. E.g., the superscript (0) first appears in eq. (1) but has never been defined. In addition, the paper uses a strange definition of $\ell_1$ norm to mean either the usual 1-norm or the 2-norm. The $\ell_2$ "norm" in this paper is defined to be the square of the usual 2-norm. This is technically not a norm!*
>
> Thank you for your comments. We have provided detailed definitions and notations at the beginning of Section 2. We are happy to make them clear here and in our revision.
>
> (1) The superscript $(0)$ denotes the input node feature (at the $0$-th layer). We will make its definition more visible in the paper.
>
> (2) For the notation of $\ell_1$ and $\ell_2$ penalty, we give the precise definitions of them in Section 2, which closely follows ElasticGNN [1].
>
> To avoid confusion, we differentiate them from the classic "norm" and change them to "penalty". We define the $\ell_1$ and $\ell_2$ penalty based on the penalty increase rate. Specifically, as shown in the Figure 3, $\ell_1$ penalty ($\|\cdot\|_1$ and $\|\cdot\|_2$) increases linearly while $\ell_2$ penalty ( and $\|\cdot\|^2_2$) increases quadraticly.
>
> [1] Liu, Xiaorui, et al. "Elastic Graph Neural Networks.", ICML'21
>
>
> ### Q2
>
> > *2. The contributions are weak as the formulation is essentially based on TWIRLS with the MCP penalty by Zhang, 2010. Robustness issues in Lasso regression have also been well-studied.*
>
> We respectfully disagree with this comment. Our contributions are significant from many perspectives.
>
> (1) First of all, the graph signal denoising perspective [1] is generally discovered as a powerful and principled paradigm for encoding prior knowledge about data. TWIRLS is just one such example. Our RUNG method differs from TWILRS from two key perspectives: different formulation and optimization solver. Therefore, our graph convolution layer in Eq. (7) derived from the proposed Quasi-Newton IRLS algorithm is totally different from the TWIRLS. Moreover, the robustness of our RUNG model is significantly stronger than TWIRLS in our experiments, especially under large attacking budgets.
>
> (2) While the robustness issue in Lasso regression is well-studied, there is no work revealing the negative impact of these issues on the robustness of GNNs. In this paper, we provide the first study and unified view to discover that the robustness degradation of a class of state-of-the-art $\ell_1$-based GNNs including SoftMedian, ElasticGNN, and TWIRLS is caused by the accumulation of estimation bias. Note that such accumulation is not observed in Lasso regression. From this novel discovery, we design a novel RUNG model to significantly avoid such strong robustness degradation, which clearly advances the state-of-the-art in GNN defense.
>
> (3) We would also like to advocate that it is imperative to design deep learning architectures with well-understood statistical principles (such as the MCP penalty we study in this paper). The recent work [2] already revealed a bitter lesson: many heuristically designed robust GNNs are not truly robust, majorly because of the lack of principled design and understanding of their robustness. On the contrary, the principally designed $\ell_1$-based GNNs exhibit the best robustness, and this work further pushes this research boundary and makes a solid step forward.
>
> Due to the above reasons, we believe the contributions and impact of our work are significant.
>
> [1] Ma et al., "A Unified View on Graph Neural Networks as Graph Signal Denoising", CIKM'21
>
> [2] Mujkanovic et al., "Are Defenses for Graph Neural Networks Robust?", NIPS 2022

---

> ### Author Response · Authors · 2023-11-22
> **Official Response to Reviewer YmW7 (2/5)**
>
> ### Q3
>
> > *3. Most of the cited works are from 2021 or before. The authors are missing recent GNN robustness works like “On the robustness of graph neural diffusion to topology perturbations", NeurIPS 2022 and “Graph-coupled oscillator networks", ICLR 2022.*
>
> Thanks for your suggestions. We have evaluated the robustness of more than 8 robust GNN baselines such as SoftMedian [1] and GNNGuard [3], which are widely recognized as state-of-the-art defenses [2] and are still competitive with or superior to more recent models. Nevertheless, we have also included two more baselines published recently, EvenNet and GARNET as shown below.
>
>
> *Global adaptive evasion attack on Cora. **Best** and <u>second</u> highlighted. Prediction accuracy in percentage:*
> | Budget                 | Clean          | 5\%                        | 10\%                       | 20\%                       | 30\%                       | 40\%                     |
> | ---------------------- | -------------- | -------------------------- | -------------------------- | -------------------------- | -------------------------- | ------------------------ |
> | GCN                    | $85.0 \pm 0.4$ | $75.3 \pm 0.5$             | $69.6 \pm 0.5$             | $60.9 \pm 0.7$             | $54.2 \pm 0.6$             | $48.4\pm0.5$             |
> | EvenNet                | $84.8 \pm 0.9$ | $75.8 \pm 0.9$             | $70.7 \pm 0.6$             | $63.9 \pm 0.5$             | $58.7 \pm 0.7$             | $54.7 \pm 0.8$           |
> | GARNET                 | $84.0 \pm 0.5$ | $76.5 \pm 0.4$             | $72.1 \pm 0.1$             | $66.4 \pm 0.7$             | $62.1 \pm 1.3$             | $58.7 \pm 1.5$           |
> | SoftMedian             | $85.0 \pm 0.7$ | $\underline{78.6 \pm 0.3}$ | $\underline{75.5 \pm 0.9}$ | $\underline{69.5 \pm 0.5}$ | $62.8 \pm 0.8$             | $58.1\pm0.7$             |
> | RUNG-$\ell_{1}$ (ours) | $85.8 \pm 0.5$ | $78.4 \pm 0.4$             | $74.3 \pm 0.3$             | $68.1 \pm 0.6$             | $\underline{63.5 \pm 0.7}$ | $\underline{59.8\pm0.8}$ |
> | RUNG (ours)            | $84.6 \pm 0.5$ | $\mathbf{78.9 \pm 0.4}$    | $\mathbf{75.7 \pm 0.2}$    | $\mathbf{71.8 \pm 0.4}$    | $\mathbf{67.8 \pm 1.3}$    | $\mathbf{65.1 \pm 1.2}$  |
>
> *Global adaptive evasion attack on Citeseer. **Best** and <u>second</u> highlighted. Prediction accuracy in percentage:*
> | Budget                 | Clean          | 5\%                        | 10\%                       | 20\%                       | 30\%                       | 40\%                     |
> | ---------------------- | -------------- | -------------------------- | -------------------------- | -------------------------- | -------------------------- | ------------------------ |
> | GCN                    | $74.8 \pm 1.2$ | $66.1 \pm 1.0$             | $60.9 \pm 0.8$             | $53.0 \pm 1.0$             | $47.0 \pm 0.8$             | $41.2\pm1.1$             |
> | EvenNet                | $74.6 \pm 0.5$ | $66.8 \pm 0.5$             | $62.0 \pm 0.6$             | $55.9 \pm 0.4$             | $51.1 \pm 0.4$             | $47.4 \pm 0.8$           |
> | GARNET                 | $74.8 \pm 1.3$ | $68.0 \pm 0.9$             | $64.0 \pm 1.1$             | $58.2 \pm 0.7$             | $53.9 \pm 0.8$             | $51.0 \pm 0.9$           |
> | SoftMedian             | $74.6 \pm 0.7$ | $68.0 \pm 0.7$             | $64.4 \pm 0.9$             | $59.3 \pm 1.1$             | $55.2 \pm 2.0$             | $51.9\pm2.1$             |
> | RUNG-$\ell_{1}$ (ours) | $75.5 \pm 1.1$ | $\underline{69.3 \pm 1.2}$ | $\underline{65.9 \pm 1.2}$ | $\underline{61.1 \pm 1.1}$ | $\underline{57.2 \pm 1.4}$ | $\underline{53.9\pm1.3}$ |
> | RUNG (ours)            | $74.3 \pm 0.7$ | $\mathbf{71.4 \pm 1.0}$    | $\mathbf{69.8 \pm 1.3}$    | $\mathbf{67.6 \pm 1.2}$    | $\mathbf{66.5 \pm 1.3}$    | $\mathbf{65.3 \pm 1.5}$  |
>
> From these results, we found that the more recent baselines are still less robust than $\ell_1$ baselines.
>
> For the two models you mentioned:
> (1) The proposed method in [6] is reported to underperform GNNGuard which has been included in the baselines of our submission; (2) The Graph Coupled Oscillator in [7] is not a GNN defense method and is certainly not robust.
>
> Therefore, we believe our evaluation has demonstrated our state-of-the-art robustness performance.
>
>
>
> [1] Geisler, Simon, et al. "Robustness of Graph Neural Networks at Scale.", NIPS'21
>
> [2] Mujkanovic et al., "Are Defenses for Graph Neural Networks Robust?", NIPS 2022
>
> [3] Zhang et al., "GNNGuard: Defending Graph Neural Networks against Adversarial Attacks", NIPS'20
>
> [4] Lei et al., "EvenNet: Ignoring Odd-Hop Neighbors Improves Robustness of Graph Neural Networks", NIPS'22
>
> [5] Song et al., "On the robustness of graph neural diffusion to topology perturbations", NIPS'22
>
> [6] Rusch et al., "Graph-coupled oscillator networks", ICML'22

---

> ### Author Response · Authors · 2023-11-22
> **Official Response to Reviewer YmW7 (3/5)**
>
> ### Q4
> > *4. The experiments are limited with tests only on Cora and CiteSeer. Attacks like TDGIA and MetaGIA are not considered.*
>
> We did experiments on the widely used Cora and Citeseer datasets since the strong adaptive attack on these datasets is well-studied. We believe they serve as a credible verification of the robustness of RUNG which avoids the false sense of robustness when evaluating our model. Nevertheless, we have done new experiments that provide stronger evidence of RUNG's superior robustness.
>
> **ogbn-arxiv**
>
> To demonstrate the scalability of RUNG, we have also conducted experiments on ogbn-arxiv.
>
> *Global evasion attack on ogbn-arxiv. Prediction accuracy in percentage:*
> | Budget               | Clean  | 1\%                   | 5\%                   | 10\%                  |
> | -------------------- | ------ | --------------------- | --------------------- | --------------------- |
> | GCN                  | $71.9$ | $63.1\pm0.4$          | $48.9\pm 3.2$         | $41.8\pm 0.5$         |
> | RUNG-$\ell_1$ (ours) | $71.6$ | $\mathbf{65.5\pm0.4}$ | $55.0\pm 1.1$         | $49.6\pm1.1$          |
> | RUNG (ours)          | $70.2$ | $65.2\pm0.2$          | $\mathbf{64.0\pm0.8}$ | $\mathbf{61.2\pm0.7}$ |
>
> The results verified the superior robustness of RUNG model especially under larger attack budgets.
>
> RUNG's robustness outperforms its $\ell_1$ variant which delivers similar performance as SoftMedian and Elastic GNN due to their shared $\ell_1$ graph smoothing, which is also the strongest baseline as shown in our experiments. We will add other baselines (if scalable) in the final version of our paper.
>
>
> **TDGIA**
>
> In order to more comprehensively evaluate the robustness of RUNG, we also conducted injection attack following your suggestion of considering TDGIA [1]. In the experiment, we set the budget on the number of injected nodes as 100 and the budget on degree as 200.
>
> *Prediction accuracy in percentage, Targeted Injection Attack on Citeseer:*
> | Budget        | Clean   | Attacked |
> | ------------- | ------- | -------- |
> | GCN           | $75.4$  | $28.14$  |
> | RUNG-$\ell_1$ | $75.01$ | $39.22 $ |
> | RUNG (ours)   | $75.65$ | $51.13$  |
>
> The results show that our RUNG significantly outperforms the baseline models. Its improvement over RUNG-$\ell_1$ again demonstrates RUNG's capability of addressing the bias accumulation problem of $\ell_1$ models.

---

> ### Author Response · Authors · 2023-11-22
> **Official Response to Reviewer YmW7 (4/5)**
>
> **PGD poisoning**
>
> In addition to the above scenarios, we have also adopted the strong PGD attack in the poisoning setting following [2].
>
>
> *Prediction accuracy in percentage, global adaptive poisoning on Cora:*
> | Budget        | 5\%                     | 10\%                    | 20\%                    | 30\%                    | 40\%                    |
> | ------------- | ----------------------- | ----------------------- | ----------------------- | ----------------------- | ----------------------- |
> | GCN           | $74.9 \pm 0.4$          | $69.7 \pm 0.7$          | $60.7 \pm 0.7$          | $54.0 \pm 1.0$          | $48.7 \pm 1.0$          |
> | APPNP         | $76.3 \pm 0.9$          | $71.1 \pm 1.2$          | $63.0 \pm 1.3$          | $57.1 \pm 0.6$          | $53.2 \pm 1.1$          |
> | SoftMedian    | $79.2 \pm 0.7$          | $75.6 \pm 0.3$          | $67.8 \pm 0.6$          | $62.9 \pm 1.0$          | $58.6 \pm 0.7$          |
> | RUNG-$\ell_1$ | $\mathbf{79.7 \pm 0.6}$ | $\mathbf{76.4 \pm 0.6}$ | $68.1 \pm 0.6$          | $63.8 \pm 0.5$          | $60.1 \pm 0.9$          |
> | RUNG (ours)   | $78.5 \pm 0.5$          | $75.5 \pm 0.3$          | $\mathbf{71.5 \pm 0.4}$ | $\mathbf{67.1 \pm 1.6}$ | $\mathbf{64.6 \pm 1.3}$ |
>
> *Prediction accuracy in percentage, global adaptive poisoning on Citeseer:*
> | Budget        | 5\%                     | 10\%                    | 20\%                    | 30\%                    | 40\%                    |
> | ------------- | ----------------------- | ----------------------- | ----------------------- | ----------------------- | ----------------------- |
> | GCN           | $65.5 \pm 1.1$          | $59.8 \pm 1.0$          | $51.0 \pm 1.0$          | $44.0 \pm 1.2$          | $37.9 \pm 1.0$          |
> | APPNP         | $64.2 \pm 1.8$          | $58.1 \pm 2.6$          | $49.8 \pm 2.5$          | $43.4 \pm 2.3$          | $40.6 \pm 2.7$          |
> | SoftMedian    | $67.1 \pm 1.0$          | $63.8 \pm 1.0$          | $58.5 \pm 1.1$          | $54.3 \pm 1.9$          | $51.2 \pm 2.4$          |
> | RUNG-$\ell_1$ | $68.9 \pm 1.0$          | $65.9 \pm 1.1$          | $61.0 \pm 1.0$          | $57.2 \pm 1.1$          | $53.9 \pm 1.3$          |
> | RUNG (ours)   | $\mathbf{72.4 \pm 0.9}$ | $\mathbf{72.1 \pm 1.2}$ | $\mathbf{71.3 \pm 1.4}$ | $\mathbf{70.8 \pm 1.3}$ | $\mathbf{69.7 \pm 1.4}$ |
>
> As can be seen from the poisoning results, the bias accumulation effect of $\ell_1$ models (SoftMedian and RUNG-$\ell_1$) also emerges. Our RUNG, however, alleviates this problem and shows a substantial improvement over the baselines, again.
>
>
>
> [1] Zou et al., “TDGIA: Effective Injection Attacks on Graph Neural Networks.”, KDD'21
>
> [2] Mujkanovic et al., "Are Defenses for Graph Neural Networks Robust?", NIPS'22

---

> ### Author Response · Authors · 2023-11-22
> **Official Response to Reviewer YmW7 (5/5)**
>
> ### Q5
>
> > *1. A quasi-Newton optimization approach is proposed. Is this related to standard quasi-Newton optimization procedures like BFGS, Broyden, etc.?*
>
> Thanks for your question. Our Quasi-Newton IRLS in RUNG model is very different from standard quasi-Newtom methods such as BFGS and Broyden. These two methods are not compatible with the efficiency and simplicity requirements of GNNs. Therefore, we design a novel quasi-Newton method that is customized for our problem. The approximate Hessian of our method is based on the upper bound of the optimization objective and is diagonal, which delivers great computation efficiency, preserving RUNG's scalability and meanwhile offering a convergence guarantee for the RUGE objective as we proved in Appendix B.
>
>
> ### Q6
>
> > *2. RUGE is claimed to be unbiased. Is this true and is it proven? An analytical robustness measure is also not provided.*
>
>
> Thanks for your comments. Actually, the unbiasedness of RUGE has both the theoretical guarantee and the empirical evidence. In our paper, we provide multiple evidence from various perspectives to clarify this:
>
> (1) From the theoretical perspective, different from the $\ell_1$ regularization which induces an estimation bias, the MCP-based problem has a nearly unbiased solution as proved in [1,2]. Roughly speaking, as shown in Figure 3, if the edge difference is larger than some threshold, the bias is totally removed; if the difference is small, the bias is
> also limited.
>
> (2) From the algorithm perspective, the firm thresholding operator $F_{\lambda,\gamma}(\theta)$ induced by the MCP-based problem is $\frac{\gamma}{\gamma-1}S_\lambda( \theta)$ when $ |\theta|<\gamma\lambda$ and keep the identity $\theta$ when $|\theta|>\gamma\lambda$. When $\theta$ is larger than the threshold $\gamma\lambda$, the estimator keeps unbiased instead of being biased towards 0 as soft-thresholding.
>
> (4) From the performance perspective, our RUNG outperforms the $\ell_1$-based GNNs such as SoftMedian, TWIRLS, and RUNG-$\ell_1$ (the $\ell_1$ variant of our model). It can tolerate severe feature differences when the attack budget increases, avoiding suffering from significant performance degradation.
>
> (5) From our ablation study in Figure 6, we quantify the estimation bias of the aggregated feature $f_i^\star$ on the attacked graph from the feature $f_i$ on the clean graph: $\sum_{i\in\mathcal{V}} \|f_i-f_i^*\|_2^2$. The results demonstrate that our proposed RUNG method exhibits a nearly zero estimation bias under adversarial attacks.
>
> We believe all of this evidence can convincingly support our claim that our RUNG estimator is nearly unbiased, which validates the motivation of our robust GNN model. We will incorporate these discussions in our revision to further clarify the unbiasedness of RUNG.
>
>
> [1] Varma, Rohan, et al. “Vector-Valued Graph Trend Filtering with Non-Convex Penalties.” IEEE Transactions on Signal and Information Processing over Networks, vol. 6, 2020, pp. 48–62.
>
> [2] Zhang, Cun-Hui. “Nearly Unbiased Variable Selection under Minimax Concave Penalty.” The Annals of Statistics, vol. 38, no. 2, Apr. 2010.
>
> To conclude, we hope we have fully addressed all of your concerns. Please kindly let us know about any further concerns you have.

---

> > ### Comment · Reviewer_YmW7 · 2023-11-23
> >
> > I thank the authors for the detailed response. I have also read the comments of other reviewers and share some of their concerns. The paper can be improved. Novelty can also be enhanced or more clearly articulated. I have increased my score.

---

### Official Review · Reviewer_WLQa · 2023-10-29

**Soundness:** 2 fair
**Presentation:** 3 good
**Contribution:** 2 fair
**Rating:** 5
**Confidence:** 4

**Summary:**

This paper proposes a robust model (RUNG) based on unbiased aggregation. Specifically, the authors first unify a few robust GNNs through the lens of $l_1$-based graph smoothing, while they still suffer from accuracy degradation when subjected to large attack budgets. To build a robust model against attacks with a large budget, authors introduce a robust and unbiased estimator by minimizing an objective function with a minimax concave penalty. Subsequently, an efficient Quasi-Newton iterative algorithm is used to optimize the objective function, from which RUNG is derived. Experimental results demonstrate that RUNG outperforms the baseline (defense) GNNs on CoraML and Citeseer, under most attack settings.

**Strengths:**

- Overall, the paper is well-written.
- The unified view of robust estimation on prior GNNs is interesting.
- Authors have considered adaptive attacks.

**Weaknesses:**

- The proposed method relies on a strong assumption of graph homophily, raising concerns about the performance of RUNG on heterophilic datasets. Thus, conducting additional experiments on heterophilic graphs is necessary. Otherwise, authors should explicitly discuss this limitation in the manuscript.
- Authors leverage QN-IRLS to approximate the inverse Hessian matrix to address the scalability issue. However, the experiments are only conducted on two small graphs, making it unclear whether the proposed approach would still work on large graphs.
- The accuracy improvement under small attack budgets is less convincing. For instance, the accuracy gap between RUNG and the runner-up model is often smaller than the standard deviation, as observed under local attacks with a 20% budget on CoraML. My concern is that RUNG might underperform the baselines with a new random seed, considering that the authors obtained averaged accuracy using only 5 different random splits.
- Authors have not performed sensitivity analyses on critical hyperparameters such as $\lambda$ and $\gamma$. It's unclear how these hyperparameters affect the performance of RUNG.
- Some recent defense models (e.g., [1, 2, 3]) are not compared in this work.
- Typo: the runner-up model is SoftMedian rather than RUNG-$l_1$ under the 5% attack budget in Table 2.

[1]: Li et al., "Reliable Representations Make A Stronger Defender: Unsupervised Structure Refinement for Robust GNN", KDD'22. \
[2]: Deng et al., “GARNET: Reduced-Rank Topology Learning for Robust and Scalable Graph Neural Networks”, LoG'22. \
[3]: Lei et al., "EvenNet: Ignoring Odd-Hop Neighbors Improves Robustness of Graph Neural Networks", NeurIPS'22.

**Questions:**

- What is $\beta$ in Section 2.3? I would suggest authors to use another notation, since $\beta$ has been used for SoftMedian.
- How do authors perform adaptive PGD attack on Jaccard-GCN, which preprocesses the adjacency matrix?
- Authors mention SVD-GCN as one of the baselines. Where are the results of SVD-GCN?
- It's unclear to me whether the defense under large attack budgets is practical. Is there any realistic application/scenario where an attacker can largely perturb the graph structure?

---

> ### Author Response · Authors · 2023-11-22
> **Official Response to Reviewer WLQa (1/4)**
>
> Dear reviewer, we appreciate your valuable comments and your recognition of our presentation, unified view on robust $\ell_1$ GNNs, and the soundness of the robustness evaluation we proposed which naturally unrolls into RUNG. We are happy to fully address all of your concerns as follows.
>
> We have updated our manuscript, and the changes are highlighted in blue. For newly added appendix sections, the section title is set to blue.
>
> ### Q1
> > *The proposed method relies on a strong assumption of graph homophily, raising concerns about the performance of RUNG on heterophilic datasets. Thus, conducting additional experiments on heterophilic graphs is necessary. Otherwise, authors should explicitly discuss this limitation in the manuscript.*
>
>
> It is true that our defense is best suited for homophilic graphs, and we will make this point clear in our revision. We would also like to point out that the majority of GNN defenses[1][2][3] rely on homophily assumption. The defense of GNNs on heterophilic graphs is largely unexplored to the best of our knowledge. The major challenge is the ambiguity between adversarial attacks and heterophic edges. The adversarial robustness in heterophilic graphs will be an interesting future direction.
>
> [1] Zhu, Dingyuan, et al. “Robust Graph Convolutional Networks Against Adversarial Attacks.”, KDD'2019
>
> [2] Wu, Huijun, et al. "Adversarial Examples on Graph Data: Deep Insights into Attack and Defense.", IJCAI'19
>
> [3] Jin, Wei, et al., "Graph Structure Learning for Robust Graph Neural Networks.", KDD'20
>
> [4] Entezari, Negin, et al. “All You Need Is Low (Rank): Defending Against Adversarial Attacks on Graphs.”, WSDM'20
>
> [5] Zhang, Xiang, and Marinka Zitnik. "GNNGuard: Defending Graph Neural Networks against Adversarial Attacks.", NIPS'20
>
> [6] Liu, Xiaorui, et al. "Elastic Graph Neural Networks.", ICML'21
>
> [7] Yang, Yongyi, et al. "Graph Neural Networks Inspired by Classical Iterative Algorithms.", ICML'21
>
> [8] Geisler, Simon, et al. "Robustness of Graph Neural Networks at Scale.", NIPS'21
>
> [9] Li, Kuan, et al. “Reliable Representations Make A Stronger Defender: Unsupervised Structure Refinement for Robust GNN.”, KDD'22
>
> [10] Song, Yang, et al. "On the Robustness of Graph Neural Diffusion to Topology Perturbations.", NIPS'22
>
>
> ### Q2
> > *Authors leverage QN-IRLS to approximate the inverse Hessian matrix to address the scalability issue. However, the experiments are only conducted on two small graphs, making it unclear whether the proposed approach would still work on large graphs.*
>
> We understand your concern. We have added an analysis of our space and time complexity in Appendix C. Let us address your concern about the scalability of our RUNG here:
>
> (1)  RUNG is scalable with time complexity of $O(k(m + n)d)$ and space complexity $O(m+nd)$, where $m$ is the number of edges, $d$ is the number of features, $n$ is the number of nodes, and $k$ is the number of GNN layers. Therefore, the complexity of our RUNG is comparable to normal GCN (with a constant difference) and it is feasible to implement.
>
> (2)  To further mitigate your concern with empirical evidence, we also provide a comparison between the runtime of GCN and RUNG with the same layer number (3 layers) on ogbn-arxiv. The result verifies the efficiency of RUNG.
>
> *Training speed on ogbn-arxiv:*
> | Model          | GCN         | RUNG        |
> | -------------- | ----------- | ----------- |
> | Training speed | $23.3$ it/s | $10.9$ it/s |
>
>
> To verify the robustness of RUNG when scaling up, we conducted experiments on the larger dataset of ogbn-arxiv.
>
> *Global evasion attack on ogbn-arxiv. Prediction accuracy in percentage:*
> | Budget        | Clean           | 1%                    | 5%                    | 10%                   |
> | ------------- | --------------- | --------------------- | --------------------- | --------------------- |
> | GCN           | $\mathbf{71.9}$ | $63.1\pm0.4$          | $48.9\pm 3.2$         | $41.8\pm 0.5$         |
> | RUNG-$\ell_1$ | $71.6$          | $\mathbf{65.5\pm0.4}$ | $55.0\pm 1.1$         | $49.6\pm1.1$          |
> | RUNG (ours)   | $70.2$          | $65.2\pm0.2$          | $\mathbf{64.0\pm0.8}$ | $\mathbf{61.2\pm0.7}$ |
>
> The results verified the superior robustness of the RUNG model especially under larger attack budgets.
>
> RUNG's robustness outperforms its $\ell_1$ variant (RUNG-$\ell_1$) which delivers similar performance as SoftMedian and Elastic GNN due to their shared $\ell_1$ graph smoothing, which is also the strongest baseline as shown in our experiments. We will add other baselines (if scalable) in the final version of our paper.

---

> ### Author Response · Authors · 2023-11-22
> **Official Response to Reviewer WLQa (2/4)**
>
> ### Q3
>
> > *The accuracy improvement under small attack budgets is less convincing. For instance, the accuracy gap between RUNG and the runner-up model is often smaller than the standard deviation, as observed under local attacks with a 20% budget on CoraML. My concern is that RUNG might underperform the baselines with a new random seed, considering that the authors obtained averaged accuracy using only 5 different random splits.*
>
> We understand your concern and let's address it as follows.
>
> (1) In most cases where the performance gap is close, the runner-up model is an $\ell_1$ model (SoftMedian or RUNG-$\ell_1$). However, the $\ell_1$ model is actually a special case of RUNG, with $\gamma\rightarrow\infty$. Therefore, RUNG can always achieve at least the performance of the $\ell_1$ model.
>
> (2) When the attack budget is small, $\ell_1$ models do not suffer from a strong accumulation of bias so they are comparable to RUNG. However, when the attack budget escalates, the bias effect will become dominant and RUNG will demonstrate a significant improvement. In fact, the large-budget performance is our emphasis when designing RUNG. An example shown below, in addition to the experiments in the paper, supports our claim.
>
> | Budget        | 50\%           | 80\%           | 100\%          |
> | ------------- | -------------- | -------------- | -------------- |
> | MLP           | $65.0 \pm 1.0$ | $65.0 \pm 1.0$ | $65.0 \pm 1.0$ |
> | GCN           | $43.2 \pm 1.2$ | $31.7 \pm 1.6$ | $25.7 \pm 2.8$ |
> | RUNG-$\ell_1$ | $52.8 \pm 1.4$ | $46.5 \pm 1.8$ | $44.5 \pm 2.5$ |
> | RUNG (ours)   | $68.4 \pm 1.2$ | $66.6 \pm 1.4$ | $65.4 \pm 1.7$ |
>
>
> ### Q4
>
> > *Authors have not performed sensitivity analyses on critical hyperparameters such as $\lambda$ and $\gamma$. It's unclear how these hyperparameters affect the performance of RUNG.*
>
>
> Thank you for the comments. We followed your suggestions to conduct a study on the accuracy of RUNG with different combinations of $\gamma$ and $\lambda$ under different attack intensities in Appendix G.1.
>
> The tuning of $\lambda$ is similar to other graph-smoothing based models [1-3] and has been well studied. Moreover, its optimal value is stable across budgets, so $\lambda$ is omitted from the result below. Now, we show the accuracy of RUNG under different attack budgets.
>
> *Global adaptive evasion attack on Cora. Prediction accuracy in percentage:*
> | $\downarrow\gamma$ \ Budget $\rightarrow$ | $5\%$            | $10\%$           | $20\%$           | $40\%$           |
> | ----------------------------------------- | ---------------- | ---------------- | ---------------- | ---------------- |
> | $0.5$                                     | $ 73.1 \pm  0.8$ | $ 72.8 \pm  0.7$ | $ 72.2 \pm  0.6$ | $ 71.7 \pm  0.6$ |
> | $1$                                       | $ 74.8 \pm  0.5$ | $ 73.5 \pm  0.4$ | $ 71.6 \pm  0.5$ | $ 69.4 \pm  0.8$ |
> | $3$                                       | $ 79.1 \pm  0.4$ | $ 75.7 \pm  0.3$ | $ 71.4 \pm  0.5$ | $ 65.4 \pm  1.6$ |
> | $5$                                       | $ 79.4 \pm  0.3$ | $ 75.9 \pm  0.2$ | $ 71.0 \pm  0.8$ | $ 64.0 \pm  2.3$ |
>
>
> The result above verifies that RUNG's performance is not very sensitive to the value of $\gamma$ and it varies in a simple way: under small budgets, a larger $\gamma$ is better, while when the budget is larger, a smaller $\gamma$ is more favorable. $\gamma$ can thus be easily tuned.
>
> We would like to emphasize that in our experiments, we keep $\gamma$ and $\lambda$ the same under different budgets for the sake of fairness. However, if the attack budget is known in advance, RUNG can be easily tuned to achieve even better performance than reported, which is also a significant advantage of our model.

---

> ### Author Response · Authors · 2023-11-22
> **Official Response to Reviewer WLQa (3/4)**
>
> ### Q5
> > *Some recent defense models (e.g., [1, 2, 3]) are not compared in this work.*
>
> Thanks for your comments. We would like to point out that STABLE and GARNET are the purification methods as ProGNN which is already included in our submission and has been verified to be weak in robustness under the adaptive attack.
>
> Following your suggestions, we added these baselines to our experiments to make a comparison, except STABLE [1] on which how to conduct adaptive attack remains unknown. Nevertheless, the experiment results in [4] show that the robustness of STABLE is worse than RGCN (included in our experiments) across several attacks (e.g., Nettack, TDGIA, G-NIA, etc.).
>
> The experimental results with the new baselines are shown below.
>
> *Global adaptive evasion attack on Cora. **Best** and <u>second</u> highlighted. Prediction accuracy in percentage:*
> | Budget                 | Clean          | 5\%                        | 10\%                       | 20\%                       | 30\%                       | 40\%                     |
> | ---------------------- | -------------- | -------------------------- | -------------------------- | -------------------------- | -------------------------- | ------------------------ |
> | GCN                    | $85.0 \pm 0.4$ | $75.3 \pm 0.5$             | $69.6 \pm 0.5$             | $60.9 \pm 0.7$             | $54.2 \pm 0.6$             | $48.4\pm0.5$             |
> | EvenNet                | $84.8 \pm 0.9$ | $75.8 \pm 0.9$             | $70.7 \pm 0.6$             | $63.9 \pm 0.5$             | $58.7 \pm 0.7$             | $54.7 \pm 0.8$           |
> | GARNET                 | $84.0 \pm 0.5$ | $76.5 \pm 0.4$             | $72.1 \pm 0.1$             | $66.4 \pm 0.7$             | $62.1 \pm 1.3$             | $58.7 \pm 1.5$           |
> | SoftMedian             | $85.0 \pm 0.7$ | $\underline{78.6 \pm 0.3}$ | $\underline{75.5 \pm 0.9}$ | $\underline{69.5 \pm 0.5}$ | $62.8 \pm 0.8$             | $58.1\pm0.7$             |
> | RUNG-$\ell_{1}$ (ours) | $85.8 \pm 0.5$ | $78.4 \pm 0.4$             | $74.3 \pm 0.3$             | $68.1 \pm 0.6$             | $\underline{63.5 \pm 0.7}$ | $\underline{59.8\pm0.8}$ |
> | RUNG (ours)            | $84.6 \pm 0.5$ | $\mathbf{78.9 \pm 0.4}$    | $\mathbf{75.7 \pm 0.2}$    | $\mathbf{71.8 \pm 0.4}$    | $\mathbf{67.8 \pm 1.3}$    | $\mathbf{65.1 \pm 1.2}$  |
>
> *Global adaptive evasion attack on Citeseer. **Best** and <u>second</u> highlighted. Prediction accuracy in percentage:*
> | Budget                 | Clean          | 5\%                        | 10\%                       | 20\%                       | 30\%                       | 40\%                     |
> | ---------------------- | -------------- | -------------------------- | -------------------------- | -------------------------- | -------------------------- | ------------------------ |
> | GCN                    | $74.8 \pm 1.2$ | $66.1 \pm 1.0$             | $60.9 \pm 0.8$             | $53.0 \pm 1.0$             | $47.0 \pm 0.8$             | $41.2\pm1.1$             |
> | EvenNet                | $74.6 \pm 0.5$ | $66.8 \pm 0.5$             | $62.0 \pm 0.6$             | $55.9 \pm 0.4$             | $51.1 \pm 0.4$             | $47.4 \pm 0.8$           |
> | GARNET                 | $74.8 \pm 1.3$ | $68.0 \pm 0.9$             | $64.0 \pm 1.1$             | $58.2 \pm 0.7$             | $53.9 \pm 0.8$             | $51.0 \pm 0.9$           |
> | SoftMedian             | $74.6 \pm 0.7$ | $68.0 \pm 0.7$             | $64.4 \pm 0.9$             | $59.3 \pm 1.1$             | $55.2 \pm 2.0$             | $51.9\pm2.1$             |
> | RUNG-$\ell_{1}$ (ours) | $75.5 \pm 1.1$ | $\underline{69.3 \pm 1.2}$ | $\underline{65.9 \pm 1.2}$ | $\underline{61.1 \pm 1.1}$ | $\underline{57.2 \pm 1.4}$ | $\underline{53.9\pm1.3}$ |
> | RUNG (ours)            | $74.3 \pm 0.7$ | $\mathbf{71.4 \pm 1.0}$    | $\mathbf{69.8 \pm 1.3}$    | $\mathbf{67.6 \pm 1.2}$    | $\mathbf{66.5 \pm 1.3}$    | $\mathbf{65.3 \pm 1.5}$  |
>
> As can be observed, these new baselines are close to or weaker than $\ell_1$ models, which validates our claims on $\ell_1$ models as well as the superior performance of RUNG.
>
>
> [1] Li et al., “Reliable Representations Make A Stronger Defender: Unsupervised Structure Refinement for Robust GNN.”, KDD'22
> [2] Deng, Chenhui, et al. "GARNET: Reduced-Rank Topology Learning for Robust and Scalable Graph Neural Networks.", LoG'22
> [3] Lei et al., "EvenNet: Ignoring Odd-Hop Neighbors Improves Robustness of Graph Neural Networks", NIPS'22.
> [4] Tao et al., "IDEA: Invariant Causal Defense for Graph Adversarial Robustness", arXiv:2305.15792

---

> ### Author Response · Authors · 2023-11-22
> **Official Response to Reviewer WLQa (4/4)**
>
> ### Q6
>
> > *Typo: the runner-up model is SoftMedian rather than RUNG-$\ell_1$ under the 5% attack budget in Table 2.*
>
> Thank you for pointing out the typo. We have amended that.
>
>
> ### Q7
>
> > *What is $\beta$ in Section 2.3? I would suggest authors to use another notation, since $\beta$ has been used for SoftMedian.*
>
> Thank you for the suggestion. $\beta$ in section 2.3 denotes the parameters in the regression model. We have changed it to another commonly used notation $\theta$.
>
> ### Q8
>
> > *How do authors perform adaptive PGD attack on Jaccard-GCN, which preprocesses the adjacency matrix?*
>
> Jaccard-GCN drops edges based on the node features in a preprocessing manner. By doing the Hadamard product between the original adjacency matrix and the edge-dropping mask matrix, there will be no problem in the attack gradient evaluation. As our results showed, the adaptive attack setting successfully broke the defense of Jaccard-GCN and is thus effective. This adaptive attack setting follows that in [1].
>
> [1] Mujkanovic et al., "Are Defenses for Graph Neural Networks Robust?", NIPS'22
>
> ### Q9
>
> > *Authors mention SVD-GCN as one of the baselines. Where are the results of SVD-GCN?*
>
> We apologize for our oversight. We missed SVD-GCN by mistake in our table and will include it in the revision. It does not show notable robustness improvement over base models under adaptive attack, which is consistent with the results reported in [1].
>
> *Global adaptive global evasion on Cora. Prediction accuracy in percentage:*
> | Budget      | 5\%                     | 10\%                    | 20\%                    | 30\%                    | 40\%                    |
> | ----------- | ----------------------- | ----------------------- | ----------------------- | ----------------------- | ----------------------- |
> | GCN         | $75.3 \pm 0.5$          | $69.6 \pm 0.5$          | $60.9 \pm 0.7$          | $54.2 \pm 0.6$          | $48.4\pm0.5$            |
> | SVD-GCN     | $68.7 \pm 0.6$          | $62.2 \pm 0.9$          | $55.5 \pm 1.4$          | $50.4 \pm 2.3$          | $46.6 \pm 1.5$          |
> | RUNG (ours) | $\mathbf{78.9 \pm 0.4}$ | $\mathbf{75.7 \pm 0.2}$ | $\mathbf{71.8 \pm 0.4}$ | $\mathbf{67.8 \pm 1.3}$ | $\mathbf{65.1 \pm 1.2}$ |
>
> *Global adaptive global evasion on Citeseer. Prediction accuracy in percentage:*
> | Budget      | 5\%                     | 10\%                    | 20\%                    | 30\%                    | 40\%                    |
> | ----------- | ----------------------- | ----------------------- | ----------------------- | ----------------------- | ----------------------- |
> | GCN         | $66.1 \pm 1.0$          | $60.9 \pm 0.8$          | $53.0 \pm 1.0$          | $47.0 \pm 0.8$          | $41.2\pm1.1$            |
> | SVD-GCN     | $60.4 \pm 0.8$          | $54.7 \pm 1.6$          | $41.2 \pm 13.2$         | $43.9 \pm 1.3$          | $40.8 \pm 3.9$          |
> | RUNG (ours) | $\mathbf{71.4 \pm 1.0}$ | $\mathbf{69.8 \pm 1.3}$ | $\mathbf{67.6 \pm 1.2}$ | $\mathbf{66.5 \pm 1.3}$ | $\mathbf{65.3 \pm 1.5}$ |
>
>
>
> [1] Mujkanovic et al., "Are Defenses for Graph Neural Networks Robust?", NIPS'22
>
>
> ### Q10
>
> > *It's unclear to me whether the defense under large attack budgets is practical. Is there any realistic application/scenario where an attacker can largely perturb the graph structure?*
>
> Yes, there are many real-world applications where the attacker can largely perturb the graph structure. For instance, Web spam on the internet [1][2] and fake accounts in social media [3] can be easily generated at scale, which could lead to a large number of adversarial edges. In financial systems like blockchain, the accounts are anonymous so it is easy to generate lots of fake accounts and transactions [4][5]. In addition, the robustness under large budgets can be seen as the worst-case scenario, which prevents stronger attacks that may not happen in the past but could happen in the future.
>
>
> [1] Gyongyi et al., "Combating Web Spam with TrustRank", VLDB'04
>
> [2] Spirin et al., "Survey on web spam detection: principles and algorithms", KDD'12
>
> [3] Khaled et al., "Detecting Fake Accounts on Social Media", IEEE BigData'18
>
> [4] Xia et al., "Characterizing cryptocurrency exchange scams", Computers & Security, vol. 98, 2020
>
> [5] Chen et al., "Do cryptocurrency exchanges fake trading volumes? An empirical analysis of wash trading based on data mining", Physica A: Statistical Mechanics and its Applications, vol 586, 2022
>
> To conclude, we hope we have fully addressed all of your concerns. Please kindly let us know about any further concerns you have.

---

> > ### Comment · Reviewer_WLQa · 2023-11-23
> > **Follow-up**
> >
> > Thanks for the detailed response and additional experiments. Here are my major concerns:
> >
> > > The defense of GNNs on heterophilic graphs is largely unexplored to the best of our knowledge.
> >
> > There have been several prior studies focusing on improving GNN robustness on heterophilic graphs, such as GNNGuard and GARNET. I would encourage authors to conduct a thorough literature review before making such a strong claim.
> >
> > > We would like to point out that STABLE and GARNET are the purification methods as ProGNN which is already included in our submission.
> >
> > Both STABLE and GARNET have demonstrated better results compared to ProGNN in their respective papers. Therefore, authors should consider comparing their work with stronger baselines. It would be more convincing if authors could comprehensively compare RUNG against these recent defense methods in the revision (and mark them in blue).

---

### Official Review · Reviewer_Vg8s · 2023-10-30

**Soundness:** 2 fair
**Presentation:** 3 good
**Contribution:** 2 fair
**Rating:** 5
**Confidence:** 4

**Summary:**

This paper analyzes the robust GNN models including SoftMedian, TWIRLS, and  ElasticGNN from a unified view of robust estimation. Building on this analysis, the authors introduce a robust unbiased aggregation method, further developing an efficient Quasi-Newton iterative reweighted least squares algorithm. However, the empirical validation seems somewhat limited, confined to two small graphs and a single attack setting.

**Strengths:**

The paper provides a unified view of $l_1$-based robust graph signal smoothing for three robust GNNs.

It introduces an unbiased graph signal estimator, which is unfolded into feature aggregation layers, aiming to enhance the robustness of GNNs.

**Weaknesses:**

1. The definition of $l_1$-based graph smoothing is not clear. The MCP function $\rho_\gamma$ used in Equation 4 and Equation 7 is not defined.

2. The paper does not provide an analysis of the computational complexity of RUNG, leaving its efficiency and applicability to larger datasets, such as Ogbn[1], unclear. Additionally, numerical experiments are limited, being only applied to Cora-ML and Citeseer datasets. It would be beneficial to see how RUNG performs on larger-scale datasets.

3. The description of the attack setting in Section 4.1 lacks clarity. Could you provide more details on how the adaptive evasion attack is designed based [2] you referenced? Specifically, what type of perturbations are performed during the attack - are they feature perturbations, graph topology perturbations, or both?

4. The robustness validation of RUNG in the paper is notably insufficient, solely relying on the PGD attack.
A comprehensive assessment using various attacks, as detailed in references [3][4][5][6], is necessary for a credible demonstration of RUNG's robustness.

5. Implementation code was not provided.

[1]. Hu W, Fey M, Zitnik M, et al. Open graph benchmark: Datasets for machine learning on graphs[J]. Advances in neural information processing systems, 2020, 33: 22118-22133.

[2]. Mujkanovic F, Geisler S, Günnemann S, et al. Are Defenses for Graph Neural Networks Robust?[J]. Advances in Neural Information Processing Systems, 2022, 35: 8954-8968.

[3]. Zheng Q, Zou X, Dong Y, et al. Graph robustness benchmark: Benchmarking the adversarial robustness of graph machine learning[J]. arXiv preprint arXiv:2111.04314, 2021.

[4]. Chen Y, Yang H, Zhang Y, et al. Understanding and improving graph injection attack by promoting unnoticeability[J]. arXiv preprint arXiv:2202.08057, 2022.

[5].D. Zügner and S. Günnemann, “Adversarial attacks on graph neural networks via meta learning,” in Proc. Int. Conf. Learn.Representations, 2019.

[6].D. Zügner, A. Akbarnejad, and S. Günnemann, “Adversarial attacks on neural networks for graph data,” in Proc. Int. Conf. Knowl. Discovery Data Mining, 2018.

**Questions:**

How to calculate the derivative in the $W_{ij}^{(k)}$ of Equation (7)?

The hyperparameter of $\lambda$ and $\gamma$  appears crucial in the model. Could you provide an ablation study to demonstrate the impact of varying these values? It would be insightful to understand how sensitive the performance of RUNG is to the selection of  $\lambda$ and $\gamma$.

---

> ### Author Response · Authors · 2023-11-22
> **Official Response to Reviewer Vg8s (1/4)**
>
> Dear reviewer, thank you so much for your thoughtful comments and your recognition of our unified view on robust $\ell_1$ GNNs and the robust unbiased graph estimator (RUGE) we proposed which naturally unrolls into RUNG. We are happy to fully address all of your concerns as follows.
>
> We have updated our manuscript, and the changes are highlighted in blue. For newly added appendix sections, the section title is set to blue.
>
> ### Q1
> > *1. The definition of $\ell_1$-based graph smoothing is not clear. The MCP function $\rho_\gamma$ used in Equation 4 and Equation 7 is not defined*
>
> We have defined the MCP in Equation 4 and showed the difference between MCP and other penalty functions in Figure 3. For the $\ell_1$-based graph smoothing models, we show three examples in Section 2.2 and summarize them in the last sentence in "A Unified View of Robust Estimation". Specifically, $\ell_1$-based GNNs share a similar graph smoothing objective with edge difference penalties $\lVert f_i-f_j\rVert_1$ or $\lVert f_i-f_j\rVert_2$., following the definitions in ElasticGNN [1].  We will make this clear in our revision.
>
> [1] Liu et al., "Elastic Graph Neural Networks.", ICML'21
>
>
> ### Q2
> > *2. The paper does not provide an analysis of the computational complexity of RUNG, leaving its efficiency and applicability to larger datasets, such as Ogbn[1], unclear. Additionally, numerical experiments are limited, being only applied to Cora-ML and Citeseer datasets. It would be beneficial to see how RUNG performs on larger-scale datasets.*
>
>
> We understand your concern. We have added an analysis of our space and time complexity in Appendix C. Let us address your concern about the scalability of our RUNG here:
>
> (1)  RUNG is scalable with time complexity of $O(k(m + n)d)$ and space complexity $O(m+nd)$, where $m$ is the number of edges, $d$ is the number of features, $n$ is the number of nodes, and $k$ is the number of GNN layers. Therefore, the complexity of our RUNG is comparable to normal GCN (with a constant difference) and it is feasible to implement.
>
> (2)  To further mitigate your concern with empirical evidence, we also provide a comparison between the runtime of GCN and RUNG with the same layer number (3 layers) on ogbn-arxiv. The result verifies the efficiency of RUNG.
>
> *Training speed on ogbn-arxiv:*
> | Model          | GCN         | RUNG        |
> | -------------- | ----------- | ----------- |
> | Training speed | $23.3$ it/s | $10.9$ it/s |
>
>
> To verify the robustness of RUNG when scaling up, we conducted experiments on the larger dataset of ogbn-arxiv.
>
> *Global evasion attack on ogbn-arxiv. Prediction accuracy in percentage:*
> | Budget        | Clean           | 1%                    | 5%                    | 10%                   |
> | ------------- | --------------- | --------------------- | --------------------- | --------------------- |
> | GCN           | $\mathbf{71.9}$ | $63.1\pm0.4$          | $48.9\pm 3.2$         | $41.8\pm 0.5$         |
> | RUNG-$\ell_1$ | $71.6$          | $\mathbf{65.5\pm0.4}$ | $55.0\pm 1.1$         | $49.6\pm1.1$          |
> | RUNG (ours)   | $70.2$          | $65.2\pm0.2$          | $\mathbf{64.0\pm0.8}$ | $\mathbf{61.2\pm0.7}$ |
>
> The results verified the superior robustness of RUNG model especially under larger attack budgets.
>
> RUNG's robustness outperforms its $\ell_1$ variant (RUNG-$\ell_1$) which delivers similar performance as SoftMedian and Elastic GNN due to their shared $\ell_1$ graph smoothing, which is also the strongest baseline as shown in our experiments. We will add other baselines (if scalable) in the final version of our paper.

---

> ### Author Response · Authors · 2023-11-22
> **Official Response to Reviewer Vg8s (2/4)**
>
> ### Q3
> > *3. The description of the attack setting in Section 4.1 lacks clarity. Could you provide more details on how the adaptive evasion attack is designed based [2] you referenced? Specifically, what type of perturbations are performed during the attack - are they feature perturbations, graph topology perturbations, or both?*
>
>
>
> The adaptive evasion attack we employ uses the graph topology PGD attack that perturbs the adjacency matrix, which has been explicitly mentioned in the revision. Additionally, we are happy to clarify attack settings we adopted here. Evasion attack means the data fed into a trained GNN model is perturbed rather than the data used to train a GNN, while adaptive means the attack is tailored for different GNN model respectively instead of using the same perturbation across different models. Specifically, the adaptive attack is realized through PGD, which leverages gradient information of the attacked GNN to precisely identify the adversarial edges, so as to modify the graph topology and mislead the GNN predictions. We didn't describe the details of PGD because it is a well-known attack [1] and it's not our contribution, but for completeness, we added some details of it in Appendix E. PGD evasion attack is claimed to be the strongest adaptive attack so far in [2], which can better reflect the true robustness of the GNNs.
>
>
> [1] Xu, Kaidi, et al., "Topology Attack and Defense for Graph Neural Networks: An Optimization Perspective.", IJCAI'19
>
> [2] Mujkanovic, Felix, et al., "Are Defenses for Graph Neural Networks Robust?", NIPS'22
>
>
> ### Q4
>
> > *4. The robustness validation of RUNG in the paper is notably insufficient, solely relying on the PGD attack. A comprehensive assessment using various attacks, as detailed in references [3][4][5][6], is necessary for a credible demonstration of RUNG's robustness.*
>
>
> Thank you for your advice. We would like to address your concerns as follows.
>
>
> (1) First of all, we would like to point out that one notable strength of our work is to avoid the false sense of security as much as we can since the recent work [1] reveals that most existing GNN defenses are not robust although they have been shown to be robust under various settings such as black-box attack and poison attack. Therefore, to obtain a reliable robustness estimation, we mainly use the strongest attack setting we have found so far: white-box adaptive attack in the evasion setting, following the work [1].
>
> (2) Since PGD generates attacks on all edges simultaneously with the relaxation of edge width instead of flipping edges greedily one by one, it can generate a stronger attack. Specifically, PGD attack is reported to be the strongest in the recent work [1], almost always outperforming Nettack [6]. Besides, the PGD-based poisoning attack named Meta-PGD [1] is also reported to outperform Metattack in most cases. Therefore, our choice of PGD attack should reflect the true robustness of our RUNG.
>
> (3) The effect of injection [4] and modification attacks are similar in terms of producing estimation bias by introducing outliers. The same applies to the poisoning [5] attack and the evasion topology attack. Therefore, a comprehensive evaluation of the modification evasion setting should be able to demonstrate the robustness of our RUNG.
>
> (4) To further test the robustness of RUNG in different scenarios, we still provide new experimental evidence including poisoning attacks and injection attacks following your suggestions. The results are shown below:

---

> > ### Comment · Reviewer_Vg8s · 2023-11-23
> >
> > > while adaptive means the attack is tailored for different GNN model respectively instead of using the same perturbation across different models
> >
> > Yes, this aligns with my concerns. How exactly is the adaptive attack tailored to your model? The description in Section E suggests it's a standard PGD applicable to any framework. What aspects are specifically designed for your model? Additionally, how does your adaptive attack relate to Section 4 in [2], particularly in addressing potential gradient obfuscation issues?
> >
> > > PGD evasion attack is claimed to be the strongest adaptive attack so far in [2], which can better reflect the true robustness of the GNNs.
> >
> > I challenge this claim. Could you specify which sentence in [2] leads to this conclusion? You have only applied a topology attack. However,  a stronger attack strategy would encompass both feature and topology attacks together.

---

> ### Author Response · Authors · 2023-11-22
> **Official Response to Reviewer Vg8s (3/4)**
>
> *Prediction accuracy in percentage, global adaptive poisoning on Cora:*
> | Budget        | 5\%                     | 10\%                    | 20\%                    | 30\%                    | 40\%                    |
> | ------------- | ----------------------- | ----------------------- | ----------------------- | ----------------------- | ----------------------- |
> | GCN           | $74.9 \pm 0.4$          | $69.7 \pm 0.7$          | $60.7 \pm 0.7$          | $54.0 \pm 1.0$          | $48.7 \pm 1.0$          |
> | APPNP         | $76.3 \pm 0.9$          | $71.1 \pm 1.2$          | $63.0 \pm 1.3$          | $57.1 \pm 0.6$          | $53.2 \pm 1.1$          |
> | SoftMedian    | $79.2 \pm 0.7$          | $75.6 \pm 0.3$          | $67.8 \pm 0.6$          | $62.9 \pm 1.0$          | $58.6 \pm 0.7$          |
> | RUNG-$\ell_1$ | $\mathbf{79.7 \pm 0.6}$ | $\mathbf{76.4 \pm 0.6}$ | $68.1 \pm 0.6$          | $63.8 \pm 0.5$          | $60.1 \pm 0.9$          |
> | RUNG (ours)   | $78.5 \pm 0.5$          | $75.5 \pm 0.3$          | $\mathbf{71.5 \pm 0.4}$ | $\mathbf{67.1 \pm 1.6}$ | $\mathbf{64.6 \pm 1.3}$ |
>
> *Prediction accuracy in percentage, global adaptive poisoning on Citeseer:*
> | Budget        | 5\%                     | 10\%                    | 20\%                    | 30\%                    | 40\%                    |
> | ------------- | ----------------------- | ----------------------- | ----------------------- | ----------------------- | ----------------------- |
> | GCN           | $65.5 \pm 1.1$          | $59.8 \pm 1.0$          | $51.0 \pm 1.0$          | $44.0 \pm 1.2$          | $37.9 \pm 1.0$          |
> | APPNP         | $64.2 \pm 1.8$          | $58.1 \pm 2.6$          | $49.8 \pm 2.5$          | $43.4 \pm 2.3$          | $40.6 \pm 2.7$          |
> | SoftMedian    | $67.1 \pm 1.0$          | $63.8 \pm 1.0$          | $58.5 \pm 1.1$          | $54.3 \pm 1.9$          | $51.2 \pm 2.4$          |
> | RUNG-$\ell_1$ | $68.9 \pm 1.0$          | $65.9 \pm 1.1$          | $61.0 \pm 1.0$          | $57.2 \pm 1.1$          | $53.9 \pm 1.3$          |
> | RUNG (ours)   | $\mathbf{72.4 \pm 0.9}$ | $\mathbf{72.1 \pm 1.2}$ | $\mathbf{71.3 \pm 1.4}$ | $\mathbf{70.8 \pm 1.3}$ | $\mathbf{69.7 \pm 1.4}$ |
>
> The poisoning attack follows the setting in [1]. As can be seen from the poisoning results, the bias accumulation effect of $\ell_1$ models (SoftMedian and RUNG-$\ell_1$) also emerges. Our RUNG, however, alleviates this problem and shows a substantial improvement over the baselines.
>
> *Prediction accuracy in percentage, Targeted Injection Attack on Citeseer:*
> | Budget        | Clean   | Attacked |
> | ------------- | ------- | -------- |
> | GCN           | $75.4$  | $28.14$  |
> | RUNG-$\ell_1$ | $75.01$ | $39.22 $ |
> | RUNG (ours)   | $75.65$ | $51.13$  |
>
> The injection attack was conducted following the settings in [2] to evaluate the robustness of different methods. We set up the budget on the number of injected nodes as 100 and the budget on degree as 200. The results show that our RUNG significantly outperforms the baseline models.
>
> We will incorporate these experiments in the final version of the paper.
>
>
> [1] Mujkanovic et al., "Are Defenses for Graph Neural Networks Robust?", NIPS'22
>
> [2] Zou et al. “TDGIA:Effective Injection Attacks on Graph Neural Networks.”, KDD'21
>
> [3] Zheng et al. "Graph robustness benchmark: Benchmarking the adversarial robustness of graph machine learning", NIPS'21
>
> [4] Chen et al., "Understanding and improving graph injection attack by promoting unnoticeability", ICLR'22
>
> [5] Zügner et al., “Adversarial attacks on graph neural networks via meta learning,”, ICLR'19.
>
> [6] Zügner et al., “Adversarial attacks on neural networks for graph data,”, KDD'18.
>
>
>
>
> ### Q5
>
> > *5. Implementation code was not provided.*
>
> We would like to kindly point out that the code is not mandatory at the submission stage. However, we will be happy to make our code available upon the acceptance of our work.

---

> > ### Comment · Reviewer_Vg8s · 2023-11-23
> >
> > The scope of the experiments on injection attacks is limited. I recommend that the authors undertake more extensive experiments as outlined in [4] to better demonstrate the robustness of their proposed models. Additionally, these injection attacks could also be applied in the context of evasion white-box attacks. From the TDGIA results, it's noticeable that the accuracy of RUNG doesn't surpass that of other robust GNN models, although comparisons with other baselines are lacking in the response.
> >
> > Given these reasons, I see no justification to change my initial evaluation and will maintain my current score.
> >
> > [4] Chen et al., "Understanding and improving graph injection attack by promoting unnoticeability", ICLR'22

---

> ### Author Response · Authors · 2023-11-22
> **Official Response to Reviewer Vg8s (4/4)**
>
> ### Q6
>
> > How to calculate the derivative in the $W$ of Equation (7)?
>
> Thank you for asking. We have revised Equation 8 (numbered Equation 7 in the initial submission) for enhanced clarity. For the detailed proof of Equation 8, we put it in Appendix B due to the space limit. We are happy clarify the explicit formulation of $W_{ij}^{(0)}$ and also a sketch of its derivation here:
>
> (1) We acquired the graph smoothing objective $\sum_{(i,j)\in\mathcal{E}}\rho(\lVert f_i-f_j\rVert_2)+\lambda \sum_k \lVert f_i-f_k^{(0)}\rVert_2$, and we proposed QN-IRLS to construct in each iteration a new quadratic upper bound of the original objective.
>
> (2) $W^{(k)}$ is the coefficient of the quadratic bound in the $k$-th iteration and it is proved to be $W_{ij}^{(k)} = \frac{\partial \rho(y)}{\partial y^2}|_{y=\lVert f_i^{(k)}-f_j^{(k)}\rVert_2}$ (note that we are taking the first order derivative w.r.t. $y^2$ instead of the second order derivative).
>
> (3) In RUNG, $\rho(y)$ takes the MCP penalty $\rho_\gamma(y)$. Therefore, $\rho(y)=\rho_\gamma(y) = 1_{y\le \gamma}(y - \frac{y^2}{2\gamma} )+ 1_{y > \gamma} \frac{\gamma}{2}$ and thus $W_{ij}^{(k)} = \max(0, \frac{1}{2\lVert f_i^{(k)} - f_j^{(k)}\rVert_2} - \frac{1}{2\gamma})$.
>
>
> (We omitted the degree normalization of $f_i$ and $f_j$ here for simplicity.)
>
>
> ### Q7
>
> > The hyperparameter of $\gamma$ and $\lambda$ appears crucial in the model. Could you provide an ablation study to demonstrate the impact of varying these values? It would be insightful to understand how sensitive the performance of RUNG is to the selection of $\lambda$ and $\gamma$.
>
> Thank you for the question. We followed your suggestions to conduct a study on the accuracy of RUNG with different combinations of $\gamma$ and $\lambda$ under different attack intensities in Appendix G.1.
>
> In summary, $\lambda$ is the regularization intensity in front of $\sum_{i\in\mathcal{V}}\lVert f_i-f_i^{(0)}\rVert$ in graph smoothing and its optimal value is stable across different attack budgets, while $\gamma$ controls the tradeoff between clean accuracy and robustness of RUNG.
>
> The tuning of $\lambda$ is similar to other graph-smoothing based models [1][2][3]. Moreover, its optimal value is stable across budgets, and we can simply tune it according to the clean scenario and fix it as the optimal value $0.11$. Therefore, $\lambda$ is omitted from the result below. Now, we show the accuracy of RUNG under different attack budgets.
>
> *Global adaptive evasion attack on Cora. Prediction accuracy in percentage:*
> | $\downarrow\gamma$ \ Budget $\rightarrow$ | $5\%$            | $10\%$           | $20\%$           | $40\%$           |
> | ----------------------------------------- | ---------------- | ---------------- | ---------------- | ---------------- |
> | $0.5$                                     | $ 73.1 \pm  0.8$ | $ 72.8 \pm  0.7$ | $ 72.2 \pm  0.6$ | $ 71.7 \pm  0.6$ |
> | $1$                                       | $ 74.8 \pm  0.5$ | $ 73.5 \pm  0.4$ | $ 71.6 \pm  0.5$ | $ 69.4 \pm  0.8$ |
> | $3$                                       | $ 79.1 \pm  0.4$ | $ 75.7 \pm  0.3$ | $ 71.4 \pm  0.5$ | $ 65.4 \pm  1.6$ |
> | $5$                                       | $ 79.4 \pm  0.3$ | $ 75.9 \pm  0.2$ | $ 71.0 \pm  0.8$ | $ 64.0 \pm  2.3$ |
>
>
> The result above verifies that RUNG's performance is not very sensitive to the value of $\gamma$ and it varies in a simple way: under small budgets, a larger $\gamma$ is better, while when the budget is larger, a smaller $\gamma$ is more favorable. $\gamma$ can thus be easily tuned.
>
> We would like to emphasize that in our experiments, we keep $\gamma$ and $\lambda$ the same under different budgets for the sake of fairness. However, if the attack budget is known in advance, RUNG can be easily tuned to achieve even better performance than reported, which is also a significant advantage of our model.
>
>
> [1] Gasteiger et al., "Predict then propagate: Graph neural networks meet personalized pagerank." ICLR'18
> [2] Liu et al., "Elastic graph neural networks.", ICML'21.
> [3] Yang, Yongyi, et al., "Graph Neural Networks Inspired by Classical Iterative Algorithms." , ICML'21
>
>
> To conclude, we believe we have fully addressed all of your concerns. Please kindly let us know about any further concerns you have.

---

> ### Author Response · Authors · 2023-11-23
>
> Dear reviewer,
>
> Thank you for your comments and we would like to make some clarifications.
>
> > Yes, this aligns with my concerns. How exactly is the adaptive attack tailored to your model? The description in Section E suggests it's a standard PGD applicable to any framework. What aspects are specifically designed for your model? Additionally, how does your adaptive attack relate to Section 4 in [2], particularly in addressing potential gradient obfuscation issues?
>
> We did address the gradient issue. In our formulation, the reweighted adjacency matrix $W\odot A$ is adopted, where $\frac{\partial\rho_\gamma(y)}{\partial y^2}$ is non-differentiable at $y=\gamma$ and the gradient is zero for $y\ge\gamma$. We therefore adopted the soft version of $\frac{\partial \rho_\gamma(y)}{\partial y^2}$ which utilizes the soft clipping $\beta\log(1 + e^{\frac{1}{\beta} (\frac{1}{2y} - \frac{1}{2\gamma})})$ where $\beta$ is the temperature.
>
> A similar soft relaxation was adopted for GNNGuard (with edge masking) in our experiments and we found the adaptive attack based on soft-relaxation effective. Our RUNG, though, still proves to be robust under the adaptive attack as is shown in the experiment results. We will add the details in the revision.
>
> > I challenge this claim. Could you specify which sentence in [2] leads to this conclusion? You have only applied a topology attack. However, a stronger attack strategy would encompass both feature and topology attacks together.
>
> We referred to this statement in Appendix I. "Comparison of success of attack approaches":
>
> *In general, we can say that PGD is the dominating attack for global evasion.*
>
> We understand that you probably would like to see the validation of RUNG under more attack scenarios, but we believe the bias effect of $\ell_1$ models as well as the unbiasedness of RUNG should generalize to different scenarios such as poisoning, injection, etc. We have provided some preliminary results in the rebuttal, providing evidence that RUNG is robust across various attack settings. More comprehensive experiments will be added.

---

### Official Review · Reviewer_Sy27 · 2023-10-31

**Soundness:** 3 good
**Presentation:** 3 good
**Contribution:** 4 excellent
**Rating:** 5
**Confidence:** 3

**Summary:**

The authors propose a unifying perspective on three "successful" graph defenses against adversarial structure perturbations. They argue that all models are instances of the so-called ElasticGNN. Moreover, the authors propose a practical optimization algorithm for the devised non-smooth objective that can approximate an L1 estimator. The resulting GNN-layer is called RUNG. The authors provide empirical evidence regarding the efficacy of their method - especially for large perturbation budgets.

**Strengths:**

1. Unified perspective on effective and established defenses
2. New optimization approach for an ElasticGNN derivative
3. Method shows strong empirical performance, especially for large perturbation budgets
4. The authors evaluate their defense using adaptive attacks and study the transferability of the perturbations between models

**Weaknesses:**

1. Empirical evaluation only using Cora ML and Citeseer is insufficient. Consider larger graphs like ogbn-arxiv as well.
1. Computational complexity and cost are neither discussed nor evaluated.
1. The authors should provide an ablation study on how the design choices affect the performance.
1. The authors should show that their method breaks. For example, complement Figure 2 with a setting where the method fails.
1. The authors state "The simulation in Figure 2 verifies that our proposed estimator (η(x) := ργ(∥x∥2)) recovers the true mean regardless of the increasing outlier ratio." Which is somewhat misleading since the asymptotically optimal breakdown point of a location estimator is 50% (under certain assumptions, e.g., without constraining the value range of the estimation).

Minor:
1. It would be interesting to see how the perturbations of RUNG transfer to the other defenses as well
1. Section 2.2 could benefit from some notes on the composition of multiple GNN layers
1. The dimension-wise median, is differentiable almost everywhere (similarly to sorting). There are only differentiability issues if the "center element" changes.

**Questions:**

1. Can the authors elaborate more on how the model behaves if the perturbation strength is very high? I.e., does the model then effectively become an MLP?
1. How does RUNG defy the breakdown point?
1. 10 layers of "graph smoothing" seems to be a lot! How does RUNG perform with fewer steps (e.g. as low as 2-3 as commonly used on Cora/Citeseer)?

I am willing to raise the score if the questions are being resolved and the empirical evaluation is improved (see above).

---

> ### Author Response · Authors · 2023-11-22
> **Official Response to Reviewer Sy27 (1/5)**
>
> Dear reviewer, thank you so much for your inspiring comments and discussions. We are glad you recognized our $\ell_1$ unified view of a family of compelling robust GNNs, the universal applicability of our QN-IRLS algorithm, the strong empirical evidence of the robustness of our RUNG, and the validity of the adaptive attack of our work. We are happy to fully address all of your concerns as follows.
>
> We have updated our manuscript, and the changes are highlighted in blue. For newly added appendix sections, the section title is set to blue.
>
>
> ### Q1
>
> > *1. Empirical evaluation only using Cora ML and Citeseer is insufficient. Consider larger graphs like ogbn-arxiv as well.*
>
> Thank you for the suggestion. Following your suggestion, we conducted new experiments on ogbn-arxiv.
>
> *Global evasion attack on ogbn-arxiv:*
> | Budget        | Clean           | 1\%                   | 5\%                   | 10\%                  |
> | ------------- | --------------- | --------------------- | --------------------- | --------------------- |
> | GCN           | $\mathbf{71.9}$ | $63.1\pm0.4$          | $48.9\pm 3.2$         | $41.8\pm 0.5$         |
> | RUNG-$\ell_1$ | $71.6$          | $\mathbf{65.5\pm0.4}$ | $55.0\pm 1.1$         | $49.6\pm1.1$          |
> | RUNG (ours)   | $70.2$          | $65.2\pm0.2$          | $\mathbf{64.0\pm0.8}$ | $\mathbf{61.2\pm0.7}$ |
>
> The results verified the superior robustness of RUNG model especially under larger attack budgets.
>
> RUNG's robustness outperforms its $\ell_1$ variant which delivers similar performance as SoftMedian and Elastic GNN due to their shared $\ell_1$ graph smoothing, which is also the strongest baseline as shown in our experiments. We will add other baselines (if scalable) in the final version of our paper.
>
> ### Q2
> > *2. Computational complexity and cost are neither discussed nor evaluated.*
>
> We understand your concern. We have added an analysis of our space and time complexity in Appendix C. Let us address your concern about the practical feasibility of our RUNG here:
>
> (1)  RUNG is scalable with time complexity of $O(k(m + n)d)$ and space complexity $O(m+nd)$, where $m$ is the number of edges, $d$ is the number of features, $n$ is the number of nodes, and $k$ is the number of GNN layers. Therefore, the complexity of our RUNG is comparable to normal GCN (with a constant difference) and it is feasible to implement.
>
> (2)  To further mitigate your concern with empirical evidence, we also provide a comparison between the runtime of GCN and RUNG with the same layer number (3 layers) on ogbn-arxiv. The result verifies the efficiency of RUNG.
>
> *Training speed on ogbn-arxiv:*
> | Model          | GCN         | RUNG        |
> | -------------- | ----------- | ----------- |
> | Training speed | $23.3$ it/s | $10.9$ it/s |

---

> ### Author Response · Authors · 2023-11-22
> **Official Response to Reviewer Sy27 (2/5)**
>
> ### Q3
> > *3. The authors should provide an ablation study on how the design choices affect the performance.*
>
> Thank you for your suggestions. Our ablation studies have presented an insightful analysis of our design choices. We have included a comprehensive ablation study in Appendix G, and we are happy to summarize how the design choices in our RUNG model affect the performance.
>
> (1) Objective perspective (RUNG-L1 vs RUNG): the objective (Equation 3) of our RUNG framework can be flexibly modified by replacing the penalty functions of the feature difference, e.g., $\ell_1$ or MCP penalty. In our main experimental results, it can be observed that RUNG (MCP) outperforms the RUNG-$\ell_1$ significantly, especially when the attack budget is large. Our estimation bias analysis in the ablation study provide a insightful explanation for this phenomenon. Specifically, in Figure 6, the bias induced by the $\ell_1$-based models accumulates at a fast rate with increasing budgets, while our RUNG keeps a nearly zero bias across various budgets. These experiments and ablation studies clearly verify the bias reduction and robustness improvement of our RUNG.
>
> (2) Algorithm perspective (IRLS vs QN-IRLS): Besides the objective, we propose a novel algorithm (QN-IRLS) to derive the iterative solution of our proposed objective as graph neural network architecture. We compare the convergence of the IRLS and our QN-IRLS in the "Convergence" part of the ablation study. In Figure 5, we can observe that QN-IRLS exhibits much better convergence than the IRLS. Actually, it is hard to choose a proper stepsize for IRLS to make it converge stably and fast. Our QN-IRLS can provide faster convergence without the need to select a stepsize as discussed in Section 3.2. We also include new ablation experiment evidence in the following tables to compare the performances for RUNG-IRLS and RUNG-QN-IRLS. As can be observed in the table, our RUNG-QN-IRLS always performs better than RUNG-IRLS, further validating the effectiveness of our algorithm.
>
> *Global evasion attack on Cora:*
> | Budget              | Clean            | 5%                      | 10%                     | 20%                     | 30%                     | 40%                     |
> | ------------------- | ---------------- | ----------------------- | ----------------------- | ----------------------- | ----------------------- | ----------------------- |
> | GCN                 | ${85.0 \pm 0.4}$ | $75.3 \pm 0.5$          | $69.6 \pm 0.5$          | $60.9 \pm 0.7$          | $54.2 \pm 0.6$          | $48.4\pm0.5$            |
> | RUNG-IRLS           | $83.3 \pm 0.7$   | $77.4 \pm 0.6$          | $73.9 \pm 0.7$          | $69.1 \pm 0.6$          | $65.5 \pm 0.8$          | $63.5 \pm 0.9$          |
> | RUNG-QN-IRLS (ours) | $84.6 \pm 0.5$   | $\mathbf{78.9 \pm 0.4}$ | $\mathbf{75.7 \pm 0.2}$ | $\mathbf{71.8 \pm 0.4}$ | $\mathbf{67.8 \pm 1.3}$ | $\mathbf{65.1 \pm 1.2}$ |
>
> *Global evasion attack on Citeseer:*
> | Budget              | Clean            | 5%                      | 10%                     | 20%                     | 30%                     | 40%                     |
> | ------------------- | ---------------- | ----------------------- | ----------------------- | ----------------------- | ----------------------- | ----------------------- |
> | GCN                 | ${74.8 \pm 1.2}$ | $66.1 \pm 1.0$          | $60.9 \pm 0.8$          | $53.0 \pm 1.0$          | $47.0 \pm 0.8$          | $41.2\pm1.1$            |
> | RUNG-IRLS           | $72.5 \pm 1.1$   | $69.3 \pm 1.0$          | $67.3 \pm 1.0$          | $65.2 \pm 1.2$          | $63.8 \pm 1.3$          | $63.4 \pm 1.1$          |
> | RUNG-QN-IRLS (ours) | $74.3 \pm 0.7$   | $\mathbf{71.4 \pm 1.0}$ | $\mathbf{69.8 \pm 1.3}$ | $\mathbf{67.6 \pm 1.2}$ | $\mathbf{66.5 \pm 1.3}$ | $\mathbf{65.3 \pm 1.5}$ |
>
> (3) Hyperparameters: the aforementioned two perspectives show the main contributions of our work, we also include the ablation study  on the hyperparameters in the appendix.  When taking different hyperparameters $\gamma$ and $\lambda$ in the graph smoothing objective, the performance of RUNG will be different. We experimented comprehensively with different combinations of $\gamma$ and $\lambda$ and the results have been added to the Appendix G1. In short, $\lambda$ has controls the regularization intensity in graph smoothing, while $\gamma$ controls the tradeoff between clean accuracy and robustness. $\lambda$ has stable optimal value agnostic to different attack budgets alleviating the tuning efforts, and a smaller $\gamma$ can enhance the robustness of RUNG under stronger attacks.
>
> We believe our our design choices are effectively explained so far. We will incorporate them into our revision.

---

> ### Author Response · Authors · 2023-11-22
> **Official Response to Reviewer Sy27 (3/5)**
>
> ### Q4
> > *4. The authors should show that their method breaks. For example, complement Figure 2 with a setting where the method fails.*
>
> > *5. The authors state "The simulation in Figure 2 verifies that our proposed estimator ($\eta(x) := \rho_\gamma(\lVert x\rVert^2)$) recovers the true mean regardless of the increasing outlier ratio." Which is somewhat misleading since the asymptotically optimal breakdown point of a location estimator is 50\% (under certain assumptions, e.g., without constraining the value range of the estimation).*
>
> Thank you for the comments and suggestions. We believe there exist some misunderstandings which we would like to clarify in (1), and a discussion on the estimators' performance beyond the breakdown point follows in (2). The connection to the graph scenario follows in (3).
>
> (1) Our outlier ratio is calculated as $\frac{m}{n}$ instead of $\frac{m}{m+n}$, where $m$ is the number of outliers and $n$ is the number of clean samples. For example, the number of outliers does not exceed that of the clean samples in our simulation (Figure 2) since "$80\%$ outliers" in our paper is actually $44\%$ of all samples. We have changed the notation in our paper to $\frac{m}{m+n}$ and added the definition for this ratio in our revision to avoid potential confusion.
>
> (2) In our simulation of Section 2.3, the mean estimators do fail when the outlier ratio is above the optimal breakdown point of $50\%$. In our setting, when the number of outliers is larger than clean samples, the outliers become the dominating class and the mean estimator will find the center of this dominating outlier class as is shown in our new experiments. These results are provided in Appendix A.2. Therefore, We have revised our phrasing following your comment to be more precise. Nevertheless, our conclusions on the robustness of different estimators are not affected.
>
> (3) We would like to emphasize that, under a budget larger than the breakdown point ($50\%$ modification ratio of edges), RUNG is still able to achieve reasonably good performance because of the extra feature similarity term $\lVert f_i-f_i^{0}\rVert_2$ in the graph estimator. This will be further elaborated in our response to your questions 1 and 2,
>
>
> ### Q5
> > *1.  It would be interesting to see how the perturbations of RUNG transfer to the other defenses as well.*
>
> Thanks for your suggestion. We have added new experiments of transfer attack among different baselines across various budgets in Appendix F. Additionally, we observed that attacks generated by RUNG are stronger when applied to more robust models like SoftMedian, while not as strong against undefended or weakly defended models. The results indicate that transfer attack performances are similar when the surrogate and victim models share a similar architecture. This again justifies the importance of adaptive attacks in avoiding the false sense of robustness since the attacks on one model might not transfer well to another model.
>
> ### Q6
> > *2.  Section 2.2 could benefit from some notes on the composition of multiple GNN layers.*
>
> Thanks for your suggestion. Section 2.2 focuses on providing a unified view and connection of several existing robust baselines, therefore we mainly describe the similar formulations of the objectives solved by the robust GNNs.  We will add some descriptions of the multi-layer architecture of the mentioned models in our revision following your suggestion.
>
> ### Q7
> > *3. The dimension-wise median, is differentiable almost everywhere (similarly to sorting). There are only differentiability issues if the "center element" changes.*
>
> Yes, the median is indeed well-defined almost everywhere. Thank you for pointing this out and we have fixed the imprecise descriptions in the paper. The reason why SoftMedian utilized the soft sorting relaxation is that the gradient of the median is zero almost everywhere [1] (elements except the "center element" have no impact on the output median) which causes problems in the training.
>
> [1] Prillo, Sebastian, and Julian Martin Eisenschlos. "SoftSort: A Continuous Relaxation for the Argsort Operator", ICML'20

---

> ### Author Response · Authors · 2023-11-22
> **Official Response to Reviewer Sy27 (4/5)**
>
> ### Q8
>
> > *1. Can the authors elaborate more on how the model behaves if the perturbation strength is very high? I.e., does the model then effectively become an MLP?*
>
> > *2. How does RUNG defy the breakdown point?*
>
>
> Thank you for your insightful questions. Based on your questions, we did additional experiments by increasing the budget of the global evasion attack on Cora to $50\%$, $80\%$, and $100\%$ and evaluated the robustness of RUNG.
>
> *Prediction accuracy under global evasion attack on Cora:*
> | Budget     | 50%           | 80%           | 100%          |
> | ---------- | -------------- | -------------- | -------------- |
> | MLP        | $65.0 \pm 1.0$ | $65.0 \pm 1.0$ | $65.0 \pm 1.0$ |
> | GCN        | $43.2 \pm 1.2$ | $31.7 \pm 1.6$ | $25.7 \pm 2.8$ |
> | RUNG       | $62.9 \pm 1.6$ | $56.3 \pm 2.7$ | $53.8 \pm 2.5$ |
> | RUNG-tuned | $68.4 \pm 1.2$ | $66.6 \pm 1.4$ | $65.4 \pm 1.7$ |
>
>
> Observations of the experiment and discussions are summarized as follows:
>
> (1) The result in the table shows that our RUNG ("RUNG" in the table) can achieve a tremendous robustness improvement compared to GCN. However, it performs worse than MLP because in our experiments the hyperparameters are set the same for different attack budgets ("RUNG" in the table takes the hyperparameters suitable for smaller budgets in our initial experiments with a largest budget of $40\%$).
>
> (2) Nevertheless, we can easily decrease the hyperparameter $\gamma$ to reach a better performance ("RUNG-tuned" in the table) if we focus on the large-budget robustness. According to the formulation of our model, we can either take $\lambda\rightarrow\infty$ or $\gamma\rightarrow 0$ to cover MLP, and in our experiments we found that under sufficient attack budgets ($> 100\%$ modification), the optimal RUNG ("RUNG-tuned") performs as if MLP because no graph information can be utilized then.
>
> (3) It is worth noting that, however, even under very large attack budgets greater or equal to the optimal breakdown point of a mean estimator (e.g., $50\%$ and $80\%$ in the table), the optimal RUNG("RUNG-tuned" in table) can still outperform MLP. This also reflects RUNG's capability of learning on a very noisy graph in the worst case.
>
> Similar observations can be made from the local attack setting in Table 1 and Table 3 of our paper, which, once again, verifies the soundness of our robustness evaluation.
>
> The behavior of RUNG under budgets larger than the breakdown point is surprising, especially when compared to the mean estimation problem. Therefore, we now elaborate on how RUNG defies the optimal breakdown point. The reason lies in an additional feature proximity term $ \sum_{i \in \mathcal{V}} \lVert f_i - f_i^{(0)}\rVert_2^2$ of our graph estimator design (RUGE in Section 3.1, Eq. (3)). This proximity (or regularization) term enforces the similarity between aggregated feature $f_i$ and the original node feature $f_i^0$, which does not exist in the mean estimator. Therefore, this regularization can provide the estimator $\min\sum_{(i,j)\in\mathcal{E}}\rho(\lVert f_i-f_j\rVert_2)$ with additional information, enabling it to correctly identify the adversarial edges and achieve superior performance even beyond the theoretical optimal breakdown point.

---

> ### Author Response · Authors · 2023-11-22
> **Official Response to Reviewer Sy27 (5/5)**
>
> ### Q9
> > *3. 10 layers of "graph smoothing" seems to be a lot! How does RUNG perform with fewer steps (e.g. as low as 2-3 as commonly used on Cora/Citeseer)?*
>
> Actually, in the graph smoothing-based models, 10 layers is a very common setting. E.g., both APPNP [1] and Elastic GNNs [2] employ 10 or more iterations.
>
> Nevertheless, RUNG also performs well with fewer steps. We added an ablation study on the number of aggregation layers in Appendix G.2 (Figure 13) and summarized the results below.
>
> *Ablation study on propagation steps on Citeseer*
> | Prop steps\Budget | Clean          | 5%             | 10%            | 20%            | 30%            | 40%            |
> | ----------------- | -------------- | -------------- | -------------- | -------------- | -------------- | -------------- |
> | 2                 | $72.6 \pm 0.9$ | $69.6 \pm 1.2$ | $68.2 \pm 1.1$ | $65.6 \pm 1.5$ | $63.9 \pm 1.8$ | $62.6 \pm 1.7$ |
> | 3                 | $73.5 \pm 1.0$ | $70.2 \pm 1.0$ | $68.4 \pm 1.3$ | $66.1 \pm 1.5$ | $64.1 \pm 1.7$ | $62.8 \pm 1.9$ |
> | 6                 | $73.6 \pm 0.8$ | $70.7 \pm 1.1$ | $69.0 \pm 1.2$ | $66.6 \pm 1.3$ | $65.1 \pm 1.8$ | $63.8 \pm 1.6$ |
> | 10                | $73.9 \pm 0.8$ | $70.8 \pm 1.0$ | $69.3 \pm 1.2$ | $67.6 \pm 1.3$ | $66.1 \pm 1.3$ | $64.8 \pm 1.4$ |
> | 20                | $74.4 \pm 0.7$ | $71.6 \pm 1.1$ | $70.1 \pm 1.2$ | $67.8 \pm 1.3$ | $66.7 \pm 1.3$ | $65.5 \pm 1.4$ |
>
> *Ablation study on propagation steps on Cora*
>
> | Prop steps\Budget | Clean          | 5%             | 10%            | 20%            | 30%            | 40%            |
> | ----------------- | -------------- | -------------- | -------------- | -------------- | -------------- | -------------- |
> | 2                 | $82.6 \pm 1.0$ | $76.8 \pm 0.9$ | $73.6 \pm 0.5$ | $68.7 \pm 0.6$ | $65.2 \pm 0.6$ | $62.1 \pm 0.5$ |
> | 3                 | $83.7 \pm 0.6$ | $77.8 \pm 0.6$ | $74.4 \pm 0.2$ | $69.7 \pm 0.2$ | $66.0 \pm 0.3$ | $63.2 \pm 0.4$ |
> | 6                 | $84.4 \pm 0.6$ | $78.6 \pm 0.5$ | $75.3 \pm 0.4$ | $71.0 \pm 0.3$ | $67.4 \pm 0.5$ | $64.5 \pm 0.8$ |
> | 10                | $84.6 \pm 0.5$ | $79.1 \pm 0.4$ | $75.6 \pm 0.2$ | $71.6 \pm 0.6$ | $68.0 \pm 0.6$ | $65.1 \pm 0.8$ |
> | 20                | $84.7 \pm 0.6$ | $79.0 \pm 0.4$ | $75.7 \pm 0.2$ | $71.4 \pm 0.5$ | $67.6 \pm 1.4$ | $65.4 \pm 1.5$ |
>
> The results show that the performance of RUNG monotonically increases as the layer number increases by approaching the precise solution of our robust unbiased graph estimator. However, we can take much fewer layers (e.g. 3 layers) with a minor performance decrease. This may be attributed to the good convergence performance of our QN-IRLS algorithm, which is also an advantage of our method.
>
>
> [1] Gasteiger et al., "Predict then propagate: Graph neural networks meet personalized pagerank." ICLR'18
>
> [2] Liu et al., "Elastic graph neural networks.", ICML'21.
>
> In conclusion, we believe we have fully addressed all of your concerns. Please kindly let us know if you have any further concerns.

---

> ### Comment · Reviewer_Sy27 · 2023-11-23
> **Inductive vs. transductive learning**
>
> I thank the authors for their elaborate and exhaustive response.
>
> To clarify my understanding, did you use the inductive or transductive split on arXiv? I apologize for not bringing up the inductive vs. transductive discussion earlier. In "transductive evasion" - depending on the specifics of the training etc. - there is the risk of leaking the clean graph during training. In other words, a model like an MLP might remember the predictions on the clean graph. However, this setting is not necessarily of practical relevance. Thus, it would be great to see how the method performs in an inductive setting (i.e., test nodes are not being used during training).

---

### Official Review · Reviewer_HR1P · 2023-11-01

**Soundness:** 2 fair
**Presentation:** 3 good
**Contribution:** 2 fair
**Rating:** 5
**Confidence:** 4

**Summary:**

The paper provides some analysis of the robustness of various GNNs and present a unified perspective to understand their strengths and limitations.  The paper identifies an issue with estimation bias in $\ell_1$-based robust graph smoothing and proposes a robust and unbiased graph signal estimator to address this bias. The Quasi-Newton IRLS algorithm is introduced, which can be integrated into GNNs as feature aggregation layers.

**Strengths:**

1. The paper provides some analysis of the robustness of various GNNs, offering valuable insights into their performance under adversarial attacks.

2.  The introduction of the Quasi-Newton IRLS algorithm, which can be integrated into GNNs as feature aggregation layers, is both innovative and practical.

**Weaknesses:**

1. The paper posits that $\ell_1$-based robust estimation introduces an estimation bias, which purportedly deteriorates model performance, especially when an attacker adds heterophilic edges. This foundational claim lacks clarity and remains unproven. The assertion that the unveiled estimation bias explains the significant performance degradation under large attack budgets is inadequately substantiated.

In the numerical simulations, "clean samples" and "outlier samples" are generated using a Gaussian distribution. This illustration raises concerns.

- The designation of red dots as outliers is perplexing. There's no discernible reason to not categorize the red dots as clean samples. If we were to accept this alternate categorization, the entire example would be questionable.  Furthermore, when the third plot indicates that 80\% of the data are outliers, the persistence of the estimation at the center of the blue dots raises concerns. Such behavior might be influenced by factors not elaborated upon in the paper. Consequently, it's premature to commend this as a positive outcome of the proposed method.


- The proposed method offers only a rudimentary approximation of one step in GNN aggregation (7). Even if the example were valid, it doesn't convincingly demonstrate robustness in GNNs. Given that $W^k$ fundamentally varies for each $k$, there's no assurance that the single-step gradient descent aligns with the theorem.

2. The computation of $W$ in eq(7) remains ambiguous. How exactly is it derived?

3. The statement "which not only provides clear interpretability but also covers many classic GNNs as special cases" is misleading. The term "covers" is inappropriate. The paper doesn't truly encompass classic GNNs as special cases but rather offers an approximate perspective on them through the lens of Graph Signal processing.

4. The paper's limitations are not adequately addressed. Like other defense papers aiming to prune the influence of heterophilic edges, this method likely has restricted applicability, perhaps being best suited for datasets that inherently assume homophily.

5. A glaring omission is the lack of reported computational costs. This oversight leaves readers questioning the practical feasibility of the proposed approach.

6. The experimental section is inadequate. The range of attacks considered is also severely limited. It is imperative to incorporate evaluations against poison and evasion attacks, as well as both white-box and black-box scenarios, and to consider both injection and modification types.

**Questions:**

1. Can you address and provide clarity on the concerns highlighted in the weaknesses section?
2. The citation style throughout the manuscript is inconsistent. Can you rectify this?

---

> ### Author Response · Authors · 2023-11-22
> **Official Response to Reviewer HR1P (1/6)**
>
> Dear reviewer HR1P, thank you so much for your valuable comments and your recognition of our contribution to further the understanding of robust GNNs via the lens of $\ell_1$ graph smoothing as well as the significance of our QN-IRLS algorithm. We are happy to fully address all of your concerns as follows.
>
> We have updated our manuscript in the latest submission, and the changes are highlighted in blue. For newly added appendix sections, the section title is set to blue.
>
> ### Q1
>
> > *1. The paper posits that $\ell_1$-based robust estimation introduces an estimation bias, which purportedly deteriorates model performance, especially when an attacker adds heterophilic edges. This foundational claim lacks clarity and remains unproven. The assertion that the unveiled estimation bias explains the significant performance degradation under large attack budgets is inadequately substantiated.*
>
> In our paper, we provide multiple evidence from various perspectives to reveal the estimation bias of $\ell_1$-based estimation. Let's clarify it:
>
> (1) From the theoretical perspective, the extensive literature on high-dimensional statistics [1,2] has proved that $\ell_1$ regularization induces an estimation bias.
>
> (2) From the algorithm perspective, in Section 2.3, we provide the explanation on the $\ell_1$-based estimation problem solver. Specifically, the soft-thresholding operator $S_\lambda(\theta):=\text{sign}(\theta)\max(|\theta|-\lambda, 0) $ induced by the $\ell_1$ regularized problem causes a constant shrinkage for $\theta$ larger than $\lambda$, enforcing the estimator to be biased towards zero with the magnitude $\lambda$.
>
> (3) From the numerical simulation in Section 2.3, we provide an example of mean estimation to verify this estimation bias. As shown in Figure 2, the $\ell_1$ estimator (green) deviates further from the true mean as the ratio of outliers escalates. This can be clearly explained as the effect of the accumulation of estimation bias. In other words, each outlier results in a constant bias, and the bias accumulates with more outliers.
>
> (4) From the performance perspective, $\ell_1$-based GNNs such as SoftMedian, TWIRLS, and RUNG-$\ell_1$ (the $\ell_1$ variant of our model) suffer from significant performance degradation when the attack budget increases.
>
> (5) From our ablation study in Figure 6, we quantify the estimation bias of the aggregated feature $f_i^\star$ on the attacked graph from the feature $f_i$ on the clean graph: $\sum_{i\in \mathcal{V}} \|f_i-f_i^*\|_2^2$. The results demonstrate
> that $\ell_1$-based GNN produces biased estimation under adversarial attacks and the bias indeed scales up with the attack budget. However, our proposed RUNG method exhibits a nearly zero estimation bias under the same attacking budgets.
>
> We believe all of this evidence can convincingly support our claim that $\ell_1$-based robust estimator suffers from the estimation bias, which validates the motivation of our new algorithm design.
>
>
> [1] Tibshirani, Robert. “Regression Shrinkage and Selection via the Lasso.” Journal of the Royal Statistical Society. Series B (Methodological), vol. 58, no. 1, 1996, pp. 267–88.
>
> [2] Zhang, Cun-Hui. “Nearly Unbiased Variable Selection under Minimax Concave Penalty.” The Annals of Statistics, vol. 38, no. 2, Apr. 2010.

---

> ### Author Response · Authors · 2023-11-22
> **Official Response to Reviewer HR1P (2/6)**
>
> ### Q1.2
> > *In the numerical simulations, "clean samples" and "outlier samples" are generated using a Gaussian distribution. This illustration raises concerns.*
>
> > *- The designation of red dots as outliers is perplexing. There's no discernible reason to not categorize the red dots as clean samples. If we were to accept this alternate categorization, the entire example would be questionable. Furthermore, when the third plot indicates that 80\% of the data are outliers, the persistence of the estimation at the center of the blue dots raises concerns. Such behavior might be influenced by factors not elaborated upon in the paper. Consequently, it's premature to commend this as a positive outcome of the proposed method.*
>
>
> Thank you for this thoughtful comment. We believe there exist some misunderstandings. Let's clarify it as follows.
>
> (1) We would like to point out that the outlier ratio in Figure 2 of our initial submission was calculated as $\frac{m}{n}$ instead of $\frac{m}{m+n}$, where $m$ is the number of outliers and $n$ is the number of clean samples. Therefore, the number of outliers does not exceed that of the clean samples in our simulation (Figure 2) since "$80$% outliers" in our paper is actually about $44$% of all samples. We have changed the notation to $\frac{m}{m+n}$ and added the definition for this ratio in our revision to avoid potential confusion.
>
> (2) In our simulation (Figure 2), the mean estimators are computed from the mixture of clean and outlier samples:
> $\min_z\sum_i^{n+m}\rho(\|z-x_i\|)$ where $\rho(y)=1_{y\le \gamma} (y - \frac{y^2}{2\gamma})+1_{y>\gamma}\frac{\gamma}{2}$ and $\gamma$ is the hyperparameter. When the number of outliers is larger than clean samples, we verified that the outliers become the dominating class and the mean estimator will be closer to this dominating class. This aligns well with your intuition. The results of this new experiment can be found in Appendix A.2.
>
> (3) We would like to emphasize that in our final graph estimator design (Section 3.1, Eq. (3)), there is an additional feature proximity term $\sum_{i\in \mathcal{V}} \|f_i - f_i^{(0)}\|_2^2$ that enforces the similarity between aggregated feature $f_i$ and the original node feature $f_i^0$, in contrast to the mean estimator. Therefore, the regularization can reduce the impact of outliers even further, which explains why our RUNG method still performs reasonably well when the attack budget is more than $100\%$ in Section 4 (Table 1, local attack).
>
> ### Q1.3
>
> > *- Regarding the numerical simulations in Figure 2: The proposed method offers only a rudimentary approximation of one step in GNN aggregation (7). Even if the example were valid, it doesn't convincingly demonstrate robustness in GNNs. Given that fundamentally varies for each, there's no assurance that the single-step gradient descent aligns with the theorem.*
>
>
> We believe there are some misunderstandings of numerical simulation in Section 2.3 and the final GNN model in Section 3. We are happy to clarify as follows.
>
> (1) The numerical simulation of mean estimators in Figure 2 (Section 2.3) aims to provide a more intuitive understanding of the bias accumulation effect of $\ell_1$ estimator. As mentioned in Section 2.3, these estimators are computed by the classic iterative Weiszfeld method for many steps until convergence since we do not worry about efficiency in such simulations.
>
> (2) In our final GNN design (Section 3), we need to develop an efficient algorithm (Quasi-Newton IRLS) to build up deep learning layers that are not only efficient but also compatible with back-propagation training. Our RUNG model stacks multiple aggregation layers using Eq. (7) rather than only one-step aggregation. Theorem 2 guarantees that each step of the iteration in Eq. (7) will decrease the objective we design (with a proof in Appendix B).

---

> ### Author Response · Authors · 2023-11-22
> **Official Response to Reviewer HR1P (3/6)**
>
> ### Q2
> > *2. The computation of $W$ in eq(7) remains ambiguous. How exactly is it derived?*
>
> Thank you for asking. We have revised Equation 8 (numbered Equation 7 in the initial submission) for enhanced clarity. For the detailed proof of Equation 8, we put it in Appendix B due to the space limit. We are happy to give the derivation as well as a sketch of proof for Equation 8 here:
>
> (1) We acquired the graph smoothing objective $\sum_{(i,j)\in\mathcal{E}}\rho(\lVert f_i-f_j\rVert_2)+\lambda \sum_k \lVert f_i-f_k^{(0)}\rVert_2$, and we want to find an efficient way to minimize it.
>
> (2) QN-IRLS is proposed and adopted, where in each iteration a new quadratic upper bound of the original objective is constructed.
>
> (3) $W^{(k)}$ is the coefficient of the quadratic bound in the $k$-th iteration and it is proved to be $W_{ij}^{(k)} = \frac{\partial \rho(y)}{\partial y^2}|_{y=\lVert f_i^{(k)}-f_j^{(k)}\rVert_2}$ (note that we are taking the first order derivative w.r.t. $y^2$ instead of the second order derivative).
>
> (4) In RUNG, $\rho(y)=\rho_\gamma(y) = 1_{y\le \gamma}(y - \frac{y^2}{2\gamma} )+ 1_{y > \gamma} \frac{\gamma}{2}$ and thus $W_{ij}^{(k)} = \max(0, \frac{1}{2\lVert f_i^{(k)} - f_j^{(k)}\rVert_2} - \frac{1}{2\gamma})$.
>
> (4) $W^{(k)}$ is then used for weighting the adjacency matrix in the $k$-th layer for aggregation.
>
> (We omitted the degree normalization of $f_i$ and $f_j$ here for simplicity.)
>
>
> ### Q3
> > *3. The statement "which not only provides clear interpretability but also covers many classic GNNs as special cases" is misleading. The term "covers" is inappropriate. The paper doesn't truly encompass classic GNNs as special cases but rather offers an approximate perspective on them through the lens of Graph Signal processing.*
>
> We precisely recover the aggregation of several classic GNNs in Section 3.3 (please refer to the paragraph "Relations with Existing GNNs"). Specifically, when $\rho(y)=y^2$, the graph aggregation layer in classic GNNs such as APPNP ($\lambda>0$) and GCN ($\lambda=0$) can be precisely recovered by Eq. (7) of our RUNG model. When $\rho(y)=y$, Eq. (7) of our RUNG model intrinsically solves the same objective as ElasticGNN, SoftMedian, and TWIRLS since they are inherently $\ell_1$-based GNNs. In this case, our solver is better than but different from their solvers. We will make this statement clear in our revision.
>
> ### Q4
> > *4. The paper's limitations are not adequately addressed. Like other defense papers aiming to prune the influence of heterophilic edges, this method likely has restricted applicability, perhaps being best suited for datasets that inherently assume homophily.*
>
> It is true that our defense is best suited for homophilic graphs, and we will make this point clear in our revision. We would also like to point out that the majority of GNN defenses[1-10] rely on homophily assumption. The defense of GNNs on heterophilic graphs is largely unexplored to the best of our knowledge. The major challenge is the ambiguity between adversarial attacks and heterophic edges. The adversarial robustness in heterophilic graphs will be an interesting future direction.
>
>
> [1] Zhu, Dingyuan, et al. “Robust Graph Convolutional Networks Against Adversarial Attacks.”, KDD'2019
>
> [2] Wu, Huijun, et al. "Adversarial Examples on Graph Data: Deep Insights into Attack and Defense.", IJCAI'19
>
> [3] Jin, Wei, et al., "Graph Structure Learning for Robust Graph Neural Networks.", KDD'20
>
> [4] Entezari, Negin, et al. “All You Need Is Low (Rank): Defending Against Adversarial Attacks on Graphs.”, WSDM'20
>
> [5] Liu, Xiaorui, et al. "Elastic Graph Neural Networks.", ICML'21
>
> [6] Yang, Yongyi, et al. "Graph Neural Networks Inspired by Classical Iterative Algorithms.", ICML'21
>
> [7] Zhang, Xiang, and Marinka Zitnik. "GNNGuard: Defending Graph Neural Networks against Adversarial Attacks.", NIPS'20
>
> [8] Geisler, Simon, et al. "Robustness of Graph Neural Networks at Scale.", NIPS'21
>
> [9] Li, Kuan, et al. “Reliable Representations Make A Stronger Defender: Unsupervised Structure Refinement for Robust GNN.”, KDD'22
>
> [10] Song, Yang, et al. "On the Robustness of Graph Neural Diffusion to Topology Perturbations.", NIPS'22

---

> ### Author Response · Authors · 2023-11-22
> **Official Response to Reviewer HR1P (4/6)**
>
> ### Q5
> > *5. A glaring omission is the lack of reported computational costs. This oversight leaves readers questioning the practical feasibility of the proposed approach.*
>
> Thank you for your comment. We have added an analysis of our space and time complexity in Appendix C. Let us address your concern about the practical feasibility of our RUNG here:
>
> (1)  RUNG is scalable with time complexity of $O(k(m + n)d)$ and space complexity $O(m+nd)$, where $m$ is the number of edges, $d$ is the number of features, $n$ is the number of nodes, and $k$ is the number of GNN layers. Therefore, the complexity of our RUNG is comparable to normal GCN (with a constant difference) and it is feasible to implement.
>
> (2)  To further mitigate your concern in practice, we also provide a comparison between the runtime of GCN and RUNG with the same layer number (3 layers) on ogbn-arxiv with an RTX-A6000 graphics card. The result verifies the efficiency of RUNG.
>
> *Training speed on ogbn-arxiv:*
> | Model          | GCN         | RUNG        |
> | -------------- | ----------- | ----------- |
> | Training speed | $23.3$ it/s | $10.9$ it/s |
>
>
> ### Q6
> > *6. The experimental section is inadequate. The range of attacks considered is also severely limited. It is imperative to incorporate evaluations against poison and evasion attacks, as well as both white-box and black-box scenarios, and to consider both injection and modification types.*
>
>
> Thank you for your advice. We would like to address your concerns as follows.
>
> (1) First of all, we would like to point out that one notable strength of our work is to avoid the false sense of security as much as we can since the recent work [1] reveals that most existing GNN defenses are not robust although they have been shown to be robust under various settings such as black-box attack and poison attack. Therefore, to obtain a reliable robustness estimation, we mainly use the strongest attack setting we have found so far: white-box adaptive attack in the evasion setting, following the work [1].
>
> (2) In our initial submission, we evaluated the black-box attack (transfer attack) and white-box attack (adaptive attack). Our experiments in Figure 8 show that the black-box (transfer attack) is much weaker than the white-box attack and it induces a severe false sense of robustness.
>
> (3) The effect of injection and modification attacks are similar in terms of producing estimation bias by introducing outliers. The same applies to the poisoning attack and the evasion topology attack. Therefore, a comprehensive evaluation of the modification evasion setting should be able to demonstrate the robustness of RUNG.
>
> (4) To further test the robustness of RUNG in different scenarios, we still provide new experimental evidence of surrogate evasion attacks poisoning attacks, and injection attacks, following your suggestions. The results are shown below:

---

> ### Author Response · Authors · 2023-11-22
> **Official Response to Reviewer HR1P (5/6)**
>
> *Global surrogate evasion attack on Cora, generated on GCN:*
> | Budget        | Clean                   | 5\%                     | 10\%                             | 20\%                    | 30\%                             | 40\%                             |
> | ------------- | ----------------------- | ----------------------- | -------------------------------- | ----------------------- | -------------------------------- | -------------------------------- |
> | GCN           | $85.0 \pm 0.4$          | $76.0 \pm 0.7$          | $70.6 \pm 0.9$                   | $62.1 \pm 1.0$          | $55.4 \pm 1.0$                   | $49.8 \pm 1.1$                   |
> | APPNP         | $\mathbf{86.3 \pm 0.4}$ | $78.0 \pm 1.0$          | $72.7 \pm 1.2$                   | $64.4 \pm 1.8$          | $57.7 \pm 1.5$                   | $53.0 \pm 1.2$                   |
> | SoftMedian    | $85.0 \pm 0.7$          | $81.8 \pm 0.4$          | $79.0 \pm 0.6$                   | $73.0 \pm 1.1$          | $66.3 \pm 1.6$                   | $60.4 \pm 2.4$                   |
> | RUNG-$\ell_1$ | $85.8\pm0.5$            | $82.3 \pm 0.8$          | $79.7 \pm 0.6$                   | $75.1 \pm 0.8$          | $70.5 \pm 0.8$                   | $67.1 \pm 0.9$                   |
> | RUNG (ours)   | $84.6 \pm 0.5$          | $\mathbf{83.7 \pm 0.6}$ | $\mathbf{82.9} \pm \mathbf{0.9}$ | $\mathbf{81.3} \pm 1.3$ | $\mathbf{79.2} \pm \mathbf{1.6}$ | $\mathbf{77.9} \pm \mathbf{2.0}$ |
>
> *Global surrogate evasion attack on Citeseer, generated on GCN:*
> | Budget        | Clean                 | 5\%                     | 10\%                    | 20\%                    | 30\%                    | 40\%                    |
> | ------------- | --------------------- | ----------------------- | ----------------------- | ----------------------- | ----------------------- | ----------------------- |
> | GCN           | $74.8 \pm 1.2$        | $66.7 \pm 1.3$          | $62.1 \pm 1.3$          | $54.5 \pm 1.7$          | $48.7 \pm 1.8$          | $43.4 \pm 2.1$          |
> | APPNP         | $75.3 \pm 1.1$        | $67.7 \pm 1.2$          | $62.9 \pm 1.0$          | $55.6 \pm 1.0$          | $50.3 \pm 1.0$          | $45.8 \pm 1.6$          |
> | SoftMedian    | $74.6 \pm 0.7$        | $68.0 \pm 0.7$          | $64.4 \pm 0.9$          | $59.3 \pm 1.1$          | $55.2 \pm 2.0$          | $51.9 \pm 2.2$          |
> | RUNG-$\ell_1$ | $\mathbf{75.5\pm1.1}$ | $72.0 \pm 1.3$          | $69.3 \pm 1.4$          | $65.1 \pm 1.8$          | $61.8 \pm 2.0$          | $58.7 \pm 2.4$          |
> | RUNG (ours)   | $74.3 \pm 0.7$        | $\mathbf{74.2 \pm 1.0}$ | $\mathbf{74.2 \pm 1.0}$ | $\mathbf{74.3 \pm 1.0}$ | $\mathbf{74.2 \pm 1.1}$ | $\mathbf{74.1 \pm 1.1}$ |
>
> The same observations of $\ell_1$ bias accumulation and the outstanding performance of unbiased RUNG can also be made. Compared to the adaptive settings in Table 1 and Table 4 in the paper, the surrogate attacks are much weaker. This is the reason why we emphasized on the adaptive attack settings which avoids the false sense of robustness.
>
>
>
> *Global adaptive poisoning on Cora:*
> | Budget        | 5\%                     | 10\%                    | 20\%                    | 30\%                    | 40\%                    |
> | ------------- | ----------------------- | ----------------------- | ----------------------- | ----------------------- | ----------------------- |
> | GCN           | $74.9 \pm 0.4$          | $69.7 \pm 0.7$          | $60.7 \pm 0.7$          | $54.0 \pm 1.0$          | $48.7 \pm 1.0$          |
> | APPNP         | $76.3 \pm 0.9$          | $71.1 \pm 1.2$          | $63.0 \pm 1.3$          | $57.1 \pm 0.6$          | $53.2 \pm 1.1$          |
> | SoftMedian    | $79.2 \pm 0.7$          | $75.6 \pm 0.3$          | $67.8 \pm 0.6$          | $62.9 \pm 1.0$          | $58.6 \pm 0.7$          |
> | RUNG-$\ell_1$ | $\mathbf{79.7 \pm 0.6}$ | $\mathbf{76.4 \pm 0.6}$ | $68.1 \pm 0.6$          | $63.8 \pm 0.5$          | $60.1 \pm 0.9$          |
> | RUNG (ours)   | $78.5 \pm 0.5$          | $75.5 \pm 0.3$          | $\mathbf{71.5 \pm 0.4}$ | $\mathbf{67.1 \pm 1.6}$ | $\mathbf{64.6 \pm 1.3}$ |

---

> ### Author Response · Authors · 2023-11-22
> **Official Response to Reviewer HR1P (6/6)**
>
> *Global adaptive poisoning on Citeseer:*
> | Budget        | 5\%                     | 10\%                    | 20\%                    | 30\%                    | 40\%                    |
> | ------------- | ----------------------- | ----------------------- | ----------------------- | ----------------------- | ----------------------- |
> | GCN           | $65.5 \pm 1.1$          | $59.8 \pm 1.0$          | $51.0 \pm 1.0$          | $44.0 \pm 1.2$          | $37.9 \pm 1.0$          |
> | APPNP         | $64.2 \pm 1.8$          | $58.1 \pm 2.6$          | $49.8 \pm 2.5$          | $43.4 \pm 2.3$          | $40.6 \pm 2.7$          |
> | SoftMedian    | $67.1 \pm 1.0$          | $63.8 \pm 1.0$          | $58.5 \pm 1.1$          | $54.3 \pm 1.9$          | $51.2 \pm 2.4$          |
> | RUNG-$\ell_1$ | $68.9 \pm 1.0$          | $65.9 \pm 1.1$          | $61.0 \pm 1.0$          | $57.2 \pm 1.1$          | $53.9 \pm 1.3$          |
> | RUNG (ours)   | $\mathbf{72.4 \pm 0.9}$ | $\mathbf{72.1 \pm 1.2}$ | $\mathbf{71.3 \pm 1.4}$ | $\mathbf{70.8 \pm 1.3}$ | $\mathbf{69.7 \pm 1.4}$ |
>
> As can be seen from the poisoning results, the bias accumulation effect of $\ell_1$ models (SoftMedian and RUNG-$\ell_1$) also emerges. Our RUNG, however, alleviates this problem and shows a substantial improvement over the baselines.
>
> *Targeted injection attack on Citeseer:*
> | Budget        | Clean   | Attacked |
> | ------------- | ------- | -------- |
> | GCN           | $75.4$  | $28.14$  |
> | RUNG-$\ell_1$ | $75.01$ | $39.22 $ |
> | RUNG (ours)   | $75.65$ | $51.13$  |
>
> The injection attack was conducted following the settings in [2] to evaluate the robustness of different methods. We set up the budget on the number of injected nodes as 100 and the budget on degree as 200. The results show that our RUNG significantly outperforms the baseline models.
>
> We will incorporate these experiments in the final version of the paper.
>
>
> [1] Mujkanovic, Felix, et al., "Are Defenses for Graph Neural Networks Robust?", NIPS'22
>
> [2] Zou, Xu, et al. “TDGIA:Effective Injection Attacks on Graph Neural Networks.”, KDD'21
>
>
> > *1. Can you address and provide clarity on the concerns highlighted in the weaknesses section?*
>
> We believe we have fully addressed all of your concerns listed above with our clarifications, new empirical evidence, and the revision to the submission.
>
> > *2. The citation style throughout the manuscript is inconsistent. Can you rectify this?*
>
> ICLR2024 has a special format requirement for the citation style. Specifically, when the citation is used as the subject or object, we use "citet". When the citation is not a part of the sentence structure, we use "citep". Therefore, there will be two different styles.
>
>
> To conclude, we believe we have fully addressed all of your concerns. Please kindly let us know if you have any further concerns.

---

> > ### Comment · Reviewer_HR1P · 2023-11-23
> >
> > I appreciate the authors for the detailed experiments added and have thoroughly read their responses, as well as the comments and responses to other reviewers. However, these only partially alleviate my concerns, prompting me to retain my current score. I recommend that the authors revise and restructure the paper for a more rigorous presentation and consider resubmitting it.

---

> > > ### Author Response · Authors · 2023-11-23
> > > **Our updated paper**
> > >
> > > Dear Reviewer,
> > >
> > > Thanks for your prompt feedback. We are glad to know that our response helps solve your concerns. Note that we have revised and restructured the paper. Could you please kindly check our updated paper? Moreover, we really appreciate it if you could let us know your remaining concerns. This can make our discussion more informative. Thank you.
> > >
> > > Best regards,
> > >
> > > All authors

---

### Meta-Review · Area_Chair_RWqh · 2023-12-18

**Metareview:**

This paper aims to establish robust graph neural networks from the perspective of unbiased unrolled updates. First, they find that existing robust GCNs are vulnerable to adaptive local evasion topological attacks, while some exceptions, SoftMedian, TWIRLS, and ElasticGNN, also encounter a similar catastrophic performance degradation as the attack budget scales up. They then analyze these models from a $\ell_1$-based perspective and discuss their estimation bias due to thresholding, as verified by numerical simulation. They proceed to build a robust and unbiased GNN. Inspired by methods in high-dimensional statistics, they propose a RUGE estimator using Minimax Concave Penalty (MCP) to alleviate the bias. To solve this non-smooth and non-convex objective, they propose a quasi-Newton IRLS algorithm with convergence guarantees. Experiments on benchmark datasets show improved robustness of the proposed model.

Although the reviewers generally appreciate the unified view, they are less convinced by the rationales and effectiveness of the proposed approach. The authors have provided a thorough response to address the concerns of the reviewers from various aspects. Nevertheless, after the discussion, some concerns still remain. For example, Reviewer HR1P still requires improvements on the presentation. E.g., the motivation and clarity of the discussion on estimation bias, still remain (Reviewer Vg8s and Reviewer YmW7 share similar concerns). Reviewer Sy27 raises some last-minute concerns on the performance of inductive settings, which are left not fully resolved. Reviewer Vg8s also raises concerns about adaptive attack and limited injection attacks. Given these shared and consistent concerns on the clarity of presentation and the completeness of evaluation, I believe this paper still needs a major revision until it meets the bar for ICLR. The authors are encouraged to take these suggestions into consideration and resubmit it to the next venues.

**Justification For Why Not Higher Score:**

The reviewers consistently believe that this paper lacks good motivation and clarity for presenting the proposed approach.

**Justification For Why Not Lower Score:**

N/A

---

### Decision · Program_Chairs · 2024-01-16

Reject